



# Ice core evidence for a recent increase in snow accumulation in coastal Dronning Maud Land, East Antarctica

Morgane Philippe[1], Jean-Louis Tison[1], Karen Fjøsne[1], Bryn Hubbard[2], Helle A. Kjær[3], Jan T. M. Lenaerts[4], Simon G. Sheldon[3], Kevin De Bondt[5], Philippe Claeys[5], Frank Pattyn[1]

[1]Laboratoire de Glaciologie, Département des Géosciences, Environnement et Société, Université Libre de Bruxelles, BE-1050 Brussels, Belgium
[2]Centre for Glaciology, Department of Geography and Earth Sciences, Aberystwyth University SY23 3DB, United Kingdom
[3]Centre for ice and climate, Niels Bohr Institute, University of Copenhagen, Juliane Maries Vej 30, 2100, Copenhagen, Denmark
[4]Institute for Marine and Atmospheric research Utrecht, Utrecht University, Princetonplein 5, 3584 CC Utrecht, Netherlands
[5]Department of Analytical Environmental and Geo-Chemistry, Vrije Universiteit Brussel, Pleinlaan 2, BE-1050 Brussels, Belgium

*Correspondence to*: M. Philippe (mophilip@ulb.ac.be)

**Abstract.** Ice cores provide temporal records of snow accumulation, a crucial component of Antarctic mass balance. Coastal areas are particularly under-represented in such records, despite their relatively high and sensitive accumulation rates. Here we present records from a 120 m ice core drilled on Derwael Ice Rise, coastal Dronning Maud Land (DML), East Antarctica in 2012. We date the ice core bottom back to 1745 ± 2 AD. $\delta^{18}$O

and δD stratigraphy is supplemented by discontinuous major ion profiles, and verified independently by electrical conductivity measurements (ECM) to detect volcanic horizons. The resulting annual layer history is combined with the core density profile to calculate accumulation history, corrected for the influence of ice deformation. The mean long-term accumulation is 0.425 ± 0.035 m water equivalent (w.e.) a$^{-1}$ (average corrected value). Reconstructed annual accumulation rates show an increase from 1955 onward to a mean value of 0.61

±0.02 m w.e. a$^{-1}$ between 1955 and 2012. This trend is compared to other reported accumulation data in Antarctica, generally showing a high spatial variability. Output of the fully coupled Community Earth System Model demonstrates that sea ice and atmospheric patterns largely explain the accumulation variability. This is the first record from a coastal ice core in East Antarctica showing a steady increase during the 20[th] and 21[st] centuries, thereby supporting modelling predictions.



## 1 Introduction

In a changing climate, it is important to know the Surface Mass Balance (SMB) of Earth's ice sheets as it is an essential component of their total mass balance, directly affecting sea level (Rignot et al., 2011). The average rate of Antarctic contribution to sea level rise is estimated to have increased from 0.08 [–0.10 to 0.27] mm a$^{-1}$ for

1992–2001 to 0.40 [0.20 to 0.61] mm a$^{-1}$ for 2002–2011 mainly due to rising ice discharge from coastal West Antarctica (Vaughan et al., 2013).

This increase in ice loss could be partly balanced by a warming-related increase in precipitation in East Antarctica (e.g Polvani et al., 2011; Krinner et al., 2007). There is consistent evidence that past Antarctic snow accumulation rates were positively correlated with past air temperature, as recently shown by Frieler et al. (2015)

using ice core data and modelling. Similarly, satellite radar and laser altimetry suggest mass gain in East Antarctica (Shepherd et al., 2012), in particular in DML, which has experienced several high-accumulation years since 2009 (Boening et al., 2012; Lenaerts et al., 2013). However, although recent regional atmospheric climate models indicate higher accumulation along the coastal sectors than in previous estimates, they show no long-term trend in the total accumulation over the continent during the past few decades (Monaghan et al., 2006; van

den Broeke et al., 2006; Bromwich et al., 2011; Lenaerts et al., 2012).

Ice cores provide temporal records of snow accumulation. These are essential to calibrate internal reflection horizons in radio-echo sounding records (e.g. Fujita et al., 2011; Kingslake et al., 2014), to force ice sheet flow and dating models (e.g. Parennin et al., 2007) and to evaluate regional climate models (e.g. Lenaerts et al., 2014). However, records of accumulation are still scarce relative to the size of Antarctica. Whilst the majority lack a

significant trend in snow accumulation over the last century (e.g. Nishio et al., 2002), some do show an increase (e.g. Karlof et al., 2005), and others show a decrease (e.g. Kaczmarska et al., 2004). Frezzotti et al. (2013) compiled surface accumulation records for the whole of Antarctica and Altnau et al. (2015) for Dronning Maud Land (DML). Both authors concluded that the trends are insignificant.

However, there is still a clear need for data from the coastal areas of East Antarctica (ISMASS Committee, 2004;

van de Berg et al., 2006; Magand et al, 2007), where very few studies have focused on ice cores, and few of those have spanned more than 20 years. Coastal regions allow higher temporal resolution as accumulation rates generally decrease with both altitude and distance from the coast (Frezzotti et al., 2005). Ice rises are ideal locations for paleoclimate studies (Matsuoka et al., 2015) as they are undisturbed by up-stream topography, since lateral flow is almost negligible. Melt events are also likely to be much less frequent than on ice shelves

(Hubbard et al., 2013).



In this paper we report on continuous ice $\delta^{18}O$ and $\delta D$ measurements (5-10 cm resolution) along a 120 m core drilled on Derwael Ice Rise (70°14'44.88'' S, 26°20'5.64''E), in coastal DML. This record is complemented by major ions profiles to improve the resolution of the seasonal cycles wherever necessary. Dating is checked independently using volcanic horizons detected from continuous electrical conductivity measurements (ECM) along the core (Hammer et al., 1994). After correcting for dynamic vertical thinning, we derive annual accumulation, and average accumulation and trends over the last 267 years, i.e. across the Anthropocene transition. These are compared to other reported trends in Antarctica including DML over the last 20 and 50 years.

## 2 Field site and methods

### 2.1 Field site

The study site is located in coastal DML, East Antarctica. A 120 m ice core was drilled in 2012 on the divide of Derwael Ice Rise, named IC12 after the project name IceCon (70°14'44.88''S, 26°20'5.64''E Figure 1), which is 486 m thick and has a local ice flow (Drews et al., 2015). Due to its coastal location, the accumulation rate is high and allows dating by seasonal peak counting. Only a few very thin melt layers are present. A continuous density profile was obtained by calibrating optical televiewer (OPTV; Hubbard et al. 2008) luminosity records in the borehole with discontinuous gravimetric measurements (Hubbard et al., 2013).

### 2.2 Ice coring

The IC12 ice core was drilled with an Eclipse electromechanical ice corer in a dry borehole. The mean length of the core sections recovered after each run was 0.77 and the standard deviation 0.40 m. Immediately after drilling, temperature (Testo 720 probe, inserted in a 4 mm diameter hole drilled to the centre of the core, precision ±0.1 °C) and length were measured on each core section, which was then wrapped in a PVC bag and stored directly in a refrigerated container at -25 °C, and kept at this temperature until analysis. The core sections were then split lengthwise in two, in a cold room at -20 °C. One half of the core section was used for ECM measurements and then kept as archive, while the other half was sectioned for continuous stable isotope sampling and discontinuous major ion analysis.

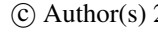



### 2.3 Annual layer counting and dating

#### 2.3.1 Water stable isotopes and major ions

Half of each core section was resampled as a central bar of 30 mm square section with a clean band saw. The outer part of the half-core was melted and stored in 4 ml bottles for $\delta^{18}O$ and $\delta D$ measurements, completely

filled to prevent contact with air. For major ions measurements, the inner bar was then placed in a Teflon holder and further decontaminated by removing ~2 mm from each face under a class-100 laminar flow hood, using a methanol-cleaned microtome blade. Each 5 cm-long decontaminated section was then covered with a clean PE storage bottle, and the sample cut loose from the bar by hitting it perpendicularly to the bar axis. Blank ice samples prepared from milliQ water were processed before every new core section and analysed for

contamination.

Dating was achieved by annual layer counting identified from the stratigraphy of the $\delta^{18}O$ and $\delta D$ isotopic composition of $H_2O$ measured (10 cm resolution in the top 80 m and 5 cm resolution below) with a PICARRO L 2130-i Cavity Ring Down Spectrometer (CRDS) (precision, $\sigma = 0.05$ ‰ for $\delta^{18}O$ and 0.3 ‰ for $\delta D$). The annual layer was identified by the $\delta^{18}O$ summer maximum value.

For sections of unclear isotopic seasonality, major ion analysis ($Na^+$, $SO_4^{--}$, $NO_3^-$, $Cl^-$) was performed with a Dionex-ICS5000 liquid chromatograph. The system has a standard deviation of 2 ppb for $Na^+$ and $SO_4^{--}$, 7 ppb for $NO_3^-$, and 8 ppb for $Cl^-$. Non sea-salt sulfate was calculated as $nssSO_4=[SO_4^{--}]_{tot} -0.052*[Cl^-]$, following Mulvaney et al. (1992) and represents all $SO_4^{--}$ not of a marine aerosol origin. The ratio $R_{Na^+/SO4^{--}}$ was also calculated as an indicator of seasonal $SO_4^{--}$ production.

#### 2.3.2 ECM and volcanic horizons

ECM measurements were made in a cold room at -18°C at the Centre for Ice and Climate, Niels Bohr Institute, University of Copenhagen with a modified version of the Copenhagen ECM described by Hammer (1980). Direct current (1250 V) was applied at the surface of the freshly cut ice and electrical conductivity was measured at a 1 mm resolution. The DC electrical conductivity of the ice, once corrected for temperature, depends

principally on its impurity content located at the crystal boundaries ($SO_4^{--}$, $NO_3^-$, $Cl^-$, etc.) (Hammer, 1980; Hammer et al., 1994). This content varies seasonally and shows longer term maxima associated with sulfate production from volcanic eruptions. ECM can therefore be used both as a relative and an absolute dating tool.

ECM data were smoothed with a 301 point wide first-order Savitsky-Golay filter (Savitsky and Golay, 1964). As measurements were principally made in firn, we multiplied the signal by the ratio of the ice density to firn




density following Kjær (2014). Finally, data were normalized by substracting the mean and dividing by the standard deviation following Karlof et al. (2000). We selected potential volcanic peaks as those above the 2σ threshold, following standard practice (e.g. Kaczmarska et al., 2004).

### 2.4. Corrections

Snow burial not only involves density changes along the vertical, but also involves lateral deformation of the underlying ice. Failure to take the latter process into account would provide an underestimation of reconstructed initial annual layer thickness, and therefore of the accumulation rate, especially within the oldest part of the record. Commonly, three different models are used to represent vertical strain rate evolution with depth: (i) a power-law model (Lliboutry, 1979), (ii) a piece-wise linear model (Dansgaard and Johnsen, 1969) and (iii) a

fully linear model (Nye, 1963).

(i) The power-law model requires measurements of the borehole horizontal displacement, which are unfortunately not available. (ii) The piece-wise model assumes a constant vertical strain rate between the surface and a given depth, which in our case is below the zone of interest since the ice core is drilled in the first quarter of the total ice rise thickness (486 m, Drews et al., 2015), and then a quadratic decrease to zero at the ice-bedrock

interface. The constant strain-rate in the upper part of the ice sheet can be inferred from the slope of water equivalent (w.e.) annual layer thickness versus depth, also in m w.e., assuming a constant long term snow accumulation (equal to annual layer thickness at the surface, Roberts et al., 2014). (iii) Finally, the Nye model corrects the layer thickness $L$ by assuming ice is incompressible, with a linear decrease from a constant annual layer thickness at the surface to zero at the ice bedrock interface (which implies a constant total ice thickness). In

that case, $L_z = L_s (z/H)$, where $H$ is the total ice thickness in m w.e., and subscripts $s$ and $z$ represent the values at the surface and at a height $z$ (in m w.e.) above the bed.

The last two corrections were applied separately and are compared in the results section.

### 2.5 Community Earth System Model (CESM)

To interpret our ice core derived accumulation record and relate it to the large-scale atmospheric and ocean

conditions, we use outputs of the Community Earth System Model (CESM). CESM is a global, fully coupled, CMIP6-generation climate model with an approximate horizontal resolution of 1 degree, and has recently been shown to realistically simulate present-day Antarctic climate and SMB (Lenaerts et al., in press). We use the historical time series of CESM (156 years, 1850-2005) that overlaps with most of the ice core record, and group the 16 single (~10%) years with the highest accumulation and lowest accumulation in that time series. We take



the mean accumulation of the ice covered CESM grid points of the coastal region around the ice core (20-30 degrees East, 69-72 degrees South) as a representative value. For the grouped years of high and low accumulation we take the anomalies (relative to the 1850-2005 mean) in near-surface temperature, sea-ice fraction and surface pressure as parameters to describe the regional ocean and atmosphere conditions corresponding to these extreme years.

## 3 Results

### 3.1 Dating

#### 3.1.1 Relative dating (seasonal peak counting)

Figure 2 illustrates how the high-resolution stable isotopes ($\delta^{18}$O, $\delta$D), smoothed ECM, chemical species and their ratios are used in combination to decipher annual layer boundaries. All of these physico-chemical variables generally show a clear seasonality. The summer peak in water stable isotopes is obvious in most cases. Major ions such as nssSO$_4$, SO$_4^{--}$/Na$^+$, NO$_3^-$ generally help to distinguish ambiguous peaks in the isotopic record. SO$_4^{--}$ is one of the oxidation products of Dimethyl Sulfide (DMS), a degradation product of DMSP which is synthesized by sea ice microorganisms (sympagic) as an antifreeze and osmotic regulator (e.g. Levasseur, 2013). Both nssSO$_4$ and R$_{Na^+/SO4^{--}}$ vary seasonally and are also strong indicators of volcanic eruptions. NO$_3^-$ also shows a seasonal signal but the processes controlling its seasonality are not yet well understood (Wolff et al., 2008). For ECM, there is also a regular seasonal signal, but only to a depth of 80 m. The different age-depth profiles resulting from this counting procedure are presented in Figure 3. No ambiguity in layer counting is detectable above 62.38 m depth (i.e. 1933 A.D.). Between 249 and 269 annual cycles are identified between the reference surface (2012 A.D.) and the bottom of the core, which is accordingly preliminarily dated to 1754 ±10A.D. before absolute dating.

#### 3.1.2 Absolute dating

In order to further improve our annual layer estimates and the depth-age relationship, we have used the ECM signal (which is mainly inherited from the SO$_4^{--}$ profile) to detect volcanic eruptions using a threshold from the background signal of 2σ (Figure 4). The best depth-age match (corresponding to the closest age match at the base of the core) was obtained with the "oldest estimate", for which 12 peaks out of 33 could be assigned to known volcanic eruptions and one more from the chemistry alone (Krakatau - 1883). Following this absolute dating recalibration, the bottom of the core is dated to 1745. The year of deposition of each volcanic peak



allowed us to reduce the uncertainty of the depth-age relationship in the IC 12 core to ±2 years. This is the precision usually associated with volcanic horizons, due to the time lapse between eruption and deposition (see sources in Table 1). The characteristics of these peaks are summarized in Table 1. The 1815 Tambora eruption has a clearly identifiable peak (Figure 4), which is expected from its high Volcanic Explosivity Index of 7 (Table 1) and its signal is detected up to two years after its eruption (e.g. Traufetter et al., 2004). Some eruptions, such as the 1762 Planchon-Peteroa eruption (assigned as unknown in Sigl et al., 2012) are recorded in both hemispheres (Sigl et al., 2012).

## 3.2 Snow accumulation rate history

Combining the annual layer thickness data set with the continuous IC12 density profile (published in Hubbard et al., 2013), we reconstructed the accumulation rate history at the summit of Derwael Ice Rise from 1745 to 2012. The cumulative thickness in w.e. is 91.8 m (Figure 5). Without correction for layer thinning, the mean annual layer thickness is 0.34 ± 0.003 m w.e., the lowest annual accumulation is 0.14 ± 0.05 m w.e. in 1834 and the highest is 1.05 ± 0.05 m w.e. in 1989 (Figure 6).

We applied two corrections: the piece-wise linear model (Dansgaard and Johnsen, 1969) and the fully linear model (Nye, 1963) (see Section 4.2) to investigate the influence of ice deformation on layer thickness, both techniques assuming a constant accumulation rate and a steady state. The piece-wise model approach cannot therefore be applied to the whole data set, since plotting annual layer thickness against depth in m w.e. reveals two trends with different slopes (Figure 5), suggesting an increase in accumulation rates. The transition occurs at ~ 49 m w.e., corresponding to 1900 A.D. Hypothesizing that, if accumulation rates have increased under the intensification of the hydrological cycle in response to the industrial revolution, we can consider the pre-1900 A.D. slope (0.003 a$^{-1}$, Figure 5) as representative of the rate of thinning associated with the constant long-term 'pre-industrial' rate of surface accumulation. We therefore used this strain rate value to correct annual layer thicknesses when applying the Dansgaard-Johnsen model.

Figure 6a shows the reconstructed history of annual accumulation rates at IC12 from 1745 to 2012, with associated error bars without ice deformation and with the two different ice-deformation models. As interannual variability is high, 11 years running means are also shown (thick lines in Figure 6a). As expected, the accumulation rate without ice deformation (blue lines in Figure 6a) is underestimated in the oldest part of the ice core as compared to the other two reconstructions taking ice deformation into account. The uncorrected curve shows a constant increase in accumulation, with multiple-step increases at ~ 1902, 1955 and 1994 A.D. The constant increase in accumulation rates before 1902 attenuates with the correction based on the Nye approach for



taking deformation into account (green lines in Figure 6a and 6b) and becomes insignificant with the Dansgaard-Johnsen model (D-J, black lines in Figure 6a and 6b). However, all curves show a clear increasing trend in accumulation rates since the early 20th century.

Accumulation rates calculated on the basis of deformation corrections (Nye and D-J) and averaged over various
periods framed by volcanic horizons (e.g. Kaczmarska et al., 2004, Sigl et al. 2012; bold years in Table 1) are shown in Figure 6b and summarized in Table 2. The long term annual accumulation, starting from the oldest volcanic layer identified: 1768 to 2012, is between 0.39 and 0.46 m w.e. a$^{-1}$ depending on the correction applied (Table 2). The recent (1955-2012) accumulation rate is between 0.60 and 0.63 m w.e. a$^{-1}$ with, as expected, less impact from the different deformation corrections. The sharpest increase occurs between the periods 1902-1955
and 1955-1992 (36% to 45% increase). With a 31 years running mean, the rate of accumulation change between 1902 and 1992 is 0.21 m w.e. a$^{-1}$ (data not shown).

Table 3 shows the detailed annual accumulation rates for the last 10 years for both corrections. The highest accumulation of the last 10 years occurred in 2009 and 2011, which belong to the 3% and 1% highest accumulation years of the whole record, respectively.

**3.3 Relation to atmospheric and sea ice patterns**

Figure 7 shows a summary of the output from the CESM as described in Section 2.5. In anomalously high-accumulation years (top panel), the sea ice coverage is very low (20-40 fewer days with sea-ice cover) in the Southern Ocean northeast of the ice core location, which is the prevalent source region of the atmospheric flow (Lenaerts et al., 2013). This is associated with higher near-surface temperatures (1-3 K), and a strengthening of
the low climatological low-pressure system (>1 hPa lower surface pressure), located offshore the ice core location (Lenaerts et al., 2013). In low-accumulation years (bottom panel), we see a reverse, albeit less strong, signal, with higher sea ice fraction, lower temperatures and higher core pressure of the low pressure system.

**4 Discussion**

**4.1 Spatial and temporal variability**

Our results show an increase in accumulation on the Derwael Ice Rise in coastal DML from 1955 onward. This confirms the studies that show a current increase in precipitation in coastal East Antarctica on the basis of satellite data and regional climate models (Davis et al., 2005, Lenaerts et al., 2012). Using a new glacial isostatic adjustment model, King et al. (2012) estimated that a 60 ±13 Gt a$^{-1}$ mass increase of the East Antarctic Ice Sheet



during the most recent period was concentrated along coastal regions, particularly in DML. However, until now, no change had been detected in ice cores from the area. Our study is the first in situ validation of a climate-related increase in coastal Antarctica precipitations which is expected to occur mainly in the peripheral areas at surface elevations below 2250 m (Krinner et al., 2007; Genthon et al., 2009).

However, not all of Antarctica would be expected to have the same accumulation trend. Figure 1 and Table A1 summarize results on accumulation trends from previous studies based on ice cores, extended with a few studies based on stake networks and radar. The colour of the site position on Figure 1 refers to the accumulation change at that site. The reference period refers to the last ~200 years, the recent period to the last ~50 years and the most recent period to the last ~20 years. The exact periods are given in Table A1.

Although the ISMASS Committee (2004) pointed out the importance of analysing coastal records, only 25 of the temporal records found in the literature concern ice cores drilled at the coast, and only 16 of them are located in DML. Only two of those records cover a period longer than 20 years: S100 (Kaczmarska et al., 2004) and B04 (Schlosser and Oerter, 2002). They both show a small decreasing trend (Figure 1).

Most studies (69% of those comparing the last ~50 years with the last ~200 years) lack a significant trend (< 
10% change). When we consider only the studies comparing the last 20 years to the last 200 years, the percentage lacking significant trend falls from 69% to 46%, for all Antarctica, but the trends revealed are both positive and negative. For example, Isaksson et al. (1996) found < 3% change at EPICA drilling site (Amundsenisen, DML) between 1865-1965 and 1966-1991. No trend was found on most inland and coastal sites (e.g.B31, S20) in DML, for the second part of the 20$^{th}$ century (Isaksson et al., 1999; Oerter et al. 1999, 2000;
Hofstede et al., 2004) or for the recent period (Fernandoy et al., 2010).

A few studies (9% for the larger period and 18% for the shorter, more recent period) show a decrease of more than 10%. This is the case for several inland sites in DML (e.g. Anschutz et al., 2011), but also coastal sites in this region (Kaczmarska et al., 2004: S100; Isaksson & Melvold, 2002: Site H; Isaksson et al., 1999: S20; Isaksson et al., 1996: Site E; Isaksson et al., 1999: Site M).

Twenty-one percent of the studies record an increase of > 10% of accumulation rates from the middle of the 20$^{th}$ century, and 36% during the most recent period. In East Antarctica, increasing trends were only recorded at inland sites, e.g. in DML (Moore et al., 1991; Oerter et al., 2000), at South Pole Station (Mosley & Thompson, 1999), Dome C (Frezzotti et al., 2005), and around Dome A (Ren et al., 2010; Ding et al., 2011). Other increasing trends were found on the Antarctic Peninsula in coastal West Antarctica (Thomas et al., 2008;
Aristarain et al., 2004). For some sites, the increase only started during the most recent period (Site M: Karlof et al., 2005). The only other coastal site in East Antarctica potentially showing an increase in snow accumulation



rates is Talos Dome, where Frezzotti et al. (2013) reported a 19% decrease during the period 1966-1996 (compared to 1816-2001), while Stenni et al. (2002) reported an increase by 11% during 1992-1996 (compared to 1816-1996).

A pattern arises when we compare the low accumulation sites to the high accumulation sites (not all coastal),
setting the threshold at 0.3 m w.e. a$^{-1}$, following Frezzotti et al. (2013) (Figure 8). The 11 sites above 0.3 m w.e. a$^{-1}$ show an average increase of accumulation of 33.8% between the last ~50 years and the reference period (last ~200 years), whereas the sites with lower accumulation show no trend (Figure 8a). This increase is more important (75%) if we compare the same reference period with the most recent period (last ~20 years) but this only covers two high accumulation sites, including IC12 (Figure 8b). Comparing the most recent period to the
last ~50 years, the 12 high accumulation sites show an average increase of 10.1% (Figure 8c).

### 4.2 Sources of uncertainties

It is important to keep in mind that the trends, reported in this study (and others) have considerable uncertainties (Rupper et al., 2015). The accuracy of reconstruction of past snow accumulation rates depends on the dating exactness. Volcanic horizons are sometimes difficult to identify in coastal ice cores due to the ECM peaks
associated with the presence of marine components. Also, given our vertical sampling resolution, the location of any single summer peak is only identifiable to a precision of 0.1 m. However, annual layer counting is easier than on inland sites, due to higher accumulation rates. Average accumulation rates on longer periods are preferred, since they are less affected by uncertainties than annual accumulation rates. These average estimates are also useful to reduce the influence of inter-annual variability.
Vertical strain rates are also a potential source of error. A companion paper will be dedicated to a more precise assessment of this factor using repeated borehole optical televiewer stratigraphy. However, the present study, by using a range of available strain rate models, shows that knowing the exact strain rates should not affect our main conclusions. Uncertainties are also influenced by the error on density and small scale variability in densification but these are assumed to be very small. For example, Callens et al. (submitted) used a semi-
empirical model of firn compaction (Arthern et al., 2010) adjusting its parameters to fit the discrete measurements instead of using the best fit in Hubbard et al. (2013). Using the first model changes accumulation values by less than 2% (data not shown). Another source of possible error is the potential migration of the ice divide. Indeed, radar layers show accumulation asymmetry next to the Derwael ice Rise divide; if the divide migrated, it could have affected the change in accumulation. However, recent data indicate that there is a very
low probability that such a migration occurred (Drews et al., 2015). Temporal variability at certain locations can



also be due to the presence of surface undulations up-glacier (e.g. Kaspari et al, 2004), but this is not the case for ice divides.

### 4.3 Causes of spatial and temporal variability

The increasing temporal trend in snow accumulation measured here and in ice cores from other areas and the spatial contrast observed could be the result of variable forcing: thermodynamic (temperature change), dynamic (change in atmospheric circulation) or both.

Increasing temperature increases the capacity of the air to hold vapour, generally enhancing precipitation. Oerter et al., (2000) showed a correlation between temperature and accumulation rates in DML. On longer timescales, using ice cores and models, Frieler et al., (2015) found a correlation between temperature and accumulation rates for the whole Antarctic continent. However, Altnau et al. (2015) found no correlation between snow accumulation and changes in ice $\delta^{18}O$ in coastal cores. They hypothesized that changes in synoptic circulation (cyclone activity) have more influence at the coast than thermodynamics alone. The increased frequency of blocking anticyclone and amplifying Rossby waves leads to the advection of moist air from the warmer middle and low latitudes (Schlosser et al., 2010; Frezzotti et al., 2013). This moisture transport is sometimes concentrated into "atmospheric rivers" of which two recent manifestations, in 2009 and 2011, have led to a positive anomaly in the net mass balance of East Antarctica (Shepherd et al., 2012; Boening et al., 2012) which was also observed in situ, at a local scale, next to the Belgian Princess Elisabeth base (72 °S, 21 °E) (Gorodetskaya et al., 2013; 2014). Such individual precipitation events can represent up to 50% of the annual accumulation (Schlosser et al., 2010; Lenaerts et al., 2013). These two highly variable accumulation events are also observed in our data as two notably higher than average accumulation years (2009 and 2011, Table 3). Our record puts these extreme events in an historical perspective, confirming that they are amongst the 1% to 3% highest accumulation years of the last two centuries, despite the fact that higher accumulation years exist in the recent part of record.

A change in climate modes could also partly explain recent changes in accumulation. The Southern Annular Mode (SAM) has shifted to a more positive phase during the last 50 years (Marshall, 2003). This has led to increasing cyclonic activity but also increasing wind speed and sublimation. Kaspari et al. (2004) also established a link between periods of increased accumulation and sustained El Niño events (negative Southern Oscillation Index (SOI) anomalies) in 1991-95 and 1940-42. We compared our detrended data set with SOI and SAM time series (KNMI, 2015) and found no correlation with either of those two indexes, yielding respective $R^2$ value of 0.0016 and 0.0026. In the detrended dataset, mean accumulation is indeed 5% higher during 1991-95



than the long-term average and 17% higher during 1940-42. However, high accumulation is also recorded during 1973-75 (19% higher than average) while that period is characterized by positive SOI values. Therefore, climate modes seem to have little influence on inter-annual variability of accumulation rates at IC12.

Small scale spatial variability in cyclonic activity and atmospheric rivers could explain why our results are
different from others in the same region. Orography can greatly affect spatial variability in snow accumulation (Lenaerts et al., 2013). Highest snowfall and highest trends in predicted snowfall are expected in the escarpment zone, due to orographic uplift (Genthon et al., 2009). The main factor generating spatial variability, however, is commonly the wind; wind ablation represents one of the largest sources of uncertainty in modelling SMB. For example, in the escarpment area of DML, low and medium precipitation amounts can be entirely removed by the
wind, while high precipitation events lead to net accumulation (Gorodetskaya et al., 2015). An enhanced wind speed coupled with an increase in accumulation could only increase SMB where the wind speed is low, while decreasing SMB in the windier areas (90% of the Antarctic surface (Frezzotti et al., 2004)). Frezzotti et al. (2013) suggested that snow accumulation has increased at low altitude sites and on the highest ridges due to more frequent anticyclone blocking events, but has decreased at intermediate altitudes due to stronger wind
ablation in the escarpment areas. In DML however, Altnau et al. (2015) reported an accumulation increase on the plateau (coupled to an increase in $\delta^{18}$O) and a decrease on coastal sites, which they associated with a change in circulation patterns. Around Dome A, Ding et al. (2011) also reported an increase in accumulation rate in the inland area and a recent decrease towards the coast. Their explanation is that air masses may transfer moisture inland more easily due to climate warming.

A more recent study using a fully coupled climate model (Lenaerts et al., in press) suggests that DML is the region most susceptible to an increase in snowfall in a present and future warmer climate. The snowfall increase in the coastal regions is particularly attributed to loss of sea ice cover in the Southern Atlantic Ocean, which in turn enhances atmospheric moisture uptake by evaporation. This is further illustrated in Figure 7, which shows that extremely high accumulation years are associated with low sea ice cover. The longer exposure of open water
leads to higher near-surface temperatures and enhances evaporation and moisture availability for ice sheet precipitation (Lenaerts et al., in press). Additionally, the low-pressure system, located offshore the ice core location (Lenaerts et al., 2013) is strengthened and invigorates meridional heat and moisture transport towards the ice sheet. The opposite is true for extremely low accumulation years.



## 5 Conclusions

A 120 m ice core was drilled on the divide of Derwael ice rise, and dated back to 1745 ±2 A.D. using $\delta^{18}$O, $\delta$D, major ions where necessary, and volcanic horizons identified from ECM data. The mean accumulation at this site is 0.425 ± 0.035 m w.e. a$^{-1}$ after corrections for densification and dynamic layer thinning. An increasing

trend in accumulation rate is observed from 1955 onwards, as expected from climate models. The trends in accumulation observed in other records all over Antarctica are spatially highly variable. In coastal East Antarctica, our study is the only to show an increase in accumulation during the 20$^{th}$ and 21$^{st}$ centuries. Many studies point to a difference in the behaviour of coastal and inland sites, due to a combination of thermodynamics and dynamic processes. Our results of the CESM suggest that accumulation variability is largely explained by

sea ice cover and atmospheric patterns. More studies are still clearly needed at other coastal sites in East Antarctica to determine how representative this result is.

Long time series are scarce in coastal East Antarctica. The divide of Derwael Ice Rise is a suitable drilling site for deep drilling. It has a high accumulation rate, and appropriate ice conditions (few thin ice layers) for paleoclimate reconstruction. With a 486 m ice thickness, drilling to the bedrock could reveal at least 2000 years

of a reliable climate record with high resolution, a priority target of the International Partnership in Ice Core Science (IPICS, Steig et al., 2005).

### Data Availability

Age-depth data and uncorrected accumulation rates are available online (doi:10.1594/PANGAEA.857574).

### Acknowledgements

This paper forms a contribution to the Belgian Research Programme on the Antarctic (Belgian Federal Science Policy Office), Project SD/SA/06A Constraining ice mass changes in Antarctica (IceCon). The authors wish to thank the International Polar Foundation for logistic support in the field. MP is partly funded by a grant from Fonds David et Alice Van Buuren. JTML is funded by Utrecht University through its strategic theme Sustainability, sub-theme Water, Climate & Ecosystems, and the programme of the Netherlands Earth System

Science Centre (NESSC), financially supported by the Ministry of Education, Culture and Science (OCW). Ph. C. thanks the Hercules Foundation (www.herculesstichting.be/) for financing the upgrade of the stable isotope laboratory. The research leading to these results has received funding from the European Research Council under





the European Community's Seventh Framework Programme (FP7/2007-2013) / ERC grant agreement 610055 as part of the Ice2Ice project. The authors also thank Irina Gorodetskaya for her helpful comments.

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





**Tables**

Table 1. Characteristics of the 12 volcanic peaks found in the IC12 core, and used to constrain the depth-age relationship to an uncertainty of ± 2 year. Bold years were used as reference for average accumulation calculations by period in Figure 6. Ref.: references: 1) Traufetter et al., 2004 and references therein ; 2) Kaczmarska et al., 2004 ; 3) Nishio et al., 2002 ; 4) Stenni et al., 2002 ; 5) Kohno and Fuji, 2002 ; 6) Zhang et al., 2002 ; 7) Moore et al., 1991 ; 8) Langway et al., 1994. *identified from ion chromatography.

| Probable source volcano | Year of eruption | Year of deposition | VEI | Depth (m) | Difference between assigned age and year of deposition | Ref. |
|---|---|---|---|---|---|---|
| Unknown | | 2009 | | 4.822 | | |
| Unknown | | 1995 | | 20.01 | | |
| Pinatubo | 1991 | **1992 ±1** | 6 | 23.095 | 0 | 1 |
| El Chichon | 1982 | 1982 ±1 | 4 | 33.63 | -2 | 1 |
| Unknown | | 1976 | | 36.42 | | |
| Unknown | | 1973 | | 38.58 | | |
| Unknown | | 1966 | | 44.08 | | |
| Agung | 1963 | 1964 ±1 | 4 | 45.95 | -1 | 1 |
| Unknown | | 1961 | | 47.15 | | |
| Carran-Los Venados | 1955 | **1955 ±1** | 4 | 50.79 | 0 | 2, 3 |
| Unknown | | 1945 | | 56.37 | | |
| Unknown | | 1940 | | 59.24 | | |
| Unknown | | 1936 | | 61.445 | | |
| Cerro Azul | 1932 | 1932 ±1 | 5 | 62.92 | 0 | 1 |
| Unknown | | 1930 | | 63.81 | | |
| Unknown | | 1922 | | 67.26 | | |
| Unknown | | 1918 | | 69.05 | | |
| Unknown | | 1916 | | 69.82 | | |
| Unknown | | 1912 | | 71.745 | | |
| Unknown | | 1908 | | 73.49 | | |
| Santa Maria | 1902 | **1902 ±1** | 5 | 75.03 | 1 | 2, 4, 5 |
| Unknown | | 1892 | | 78.84 | | |
| Krakatau | 1883 | **1884 ±1** | 6 | 82.237* | 0 | 1 |
| Unknown | | 1844 | | 94.98 | | |
| Coseguina | 1835 | 1835 ±1 | 5 | 97.34 | 0 | 1 |
| Galunggung | 1822 | 1822 ±1 | 5 | 101.3 | -1 | 2, 5, 6 |
| Tambora | 1815 | **1816 ±1** | 7 | 102.4 | 2 | 1 |
| Unknown | 1809 ± 2 | 1809 ±3 | ? | 104 | 2 | 1 |
| Cotopaxi | 1768 | **1768 ±1** | 4 | 115.3 | -1 | 2, 7, 8 |
| Planchon-Peteroa | 1762 | 1762 ±1 | 4 | 116.2 | 1 | 1 |
| Unknown | | 1759 | | 117.4 | | |
| Unknown | | 1750 | | 119.2 | | |
| Unknown | | 1747 | | 119.9 | | |





Table 2. Average accumulation rates at IC12 for various time periods framed by volcanic horizons. The first year of each period is included, not the second (ex: 1768-2012: includes 1768, not 2012). Nye: correction for a linear decrease of annual layer thickness with depth. D-J: Corrected using a strain rate of 0.003 $a^{-1}$ which is the slope of the annual layer thickness (in m w.e.) vs. depth relationship before 1900.

| Period | Accumulation (m w. e. $a^{-1}$) (Nye) | | Accumulation (m w. e. $a^{-1}$) (D-J) | |
|---|---|---|---|---|
| 1768 - 2012 | 0.39 | ± 0.003 | 0.46 | ± 0.004 |
| 1992 - 2012 | 0.64 | ± 0.07 | 0.65 | ± 0.07 |
| 1955 - 1992 | 0.58 | ± 0.03 | 0.62 | ± 0.03 |
| 1902 - 1955 | 0.40 | ± 0.02 | 0.46 | ± 0.02 |
| 1884 - 1902 | 0.36 | ± 0.04 | 0.43 | ± 0.05 |
| 1816 - 1884 | 0.31 | ± 0.01 | 0.39 | ± 0.01 |
| 1768 - 1816 | 0.27 | ± 0.01 | 0.37 | ± 0.01 |
| 1768 - 1955 | 0.33 | ± 0.004 | 0.41 | ± 0.005 |
| 1955 - 2012 | 0.60 | ± 0.02 | 0.63 | ± 0.02 |



Table 3. Accumulation for the last 10 years from IC12 ice core. *See Table 2 legend and text for explanation **not a full year

| Year (A.D.) | Accumulation (m w.e. a$^{-1}$) (Nye*) | Accumulation (m w.e. a$^{-1}$) (D-J*) |
|---|---|---|
| 2012** | 0.482 | 0.482 |
| 2011 | 0.959 | 0.960 |
| 2010 | 0.632 | 0.634 |
| 2009 | 0.811 | 0.815 |
| 2008 | 0.642 | 0.646 |
| 2007 | 0.690 | 0.695 |
| 2006 | 0.653 | 0.658 |
| 2005 | 0.673 | 0.679 |
| 2004 | 0.658 | 0.666 |
| 2003 | 0.613 | 0.621 |





~1960-present vs ~1816-present

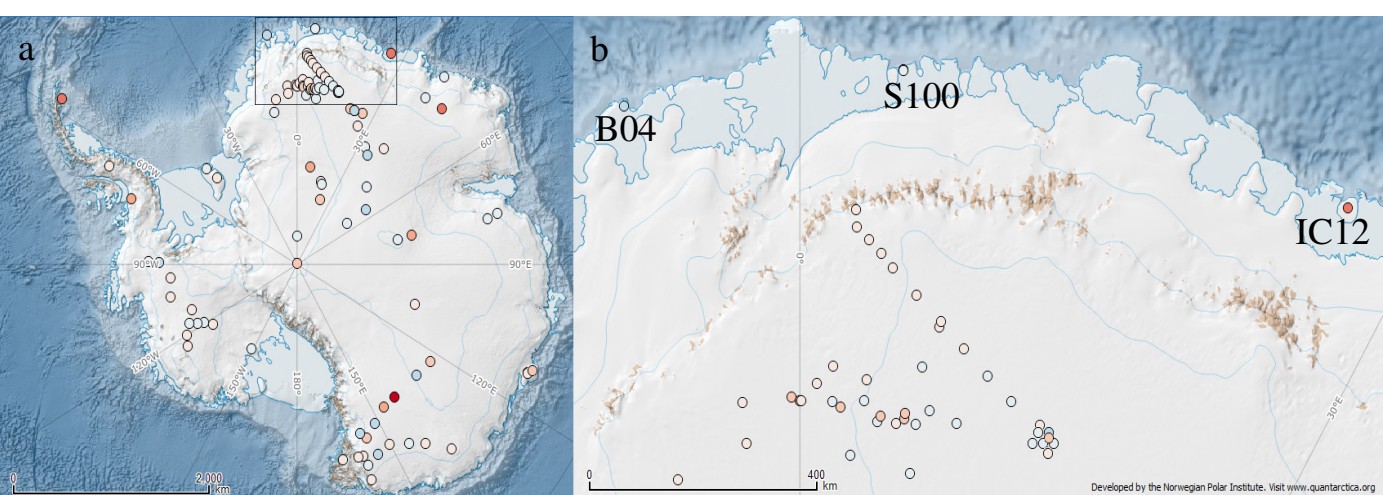

~1990-present vs ~1816 present

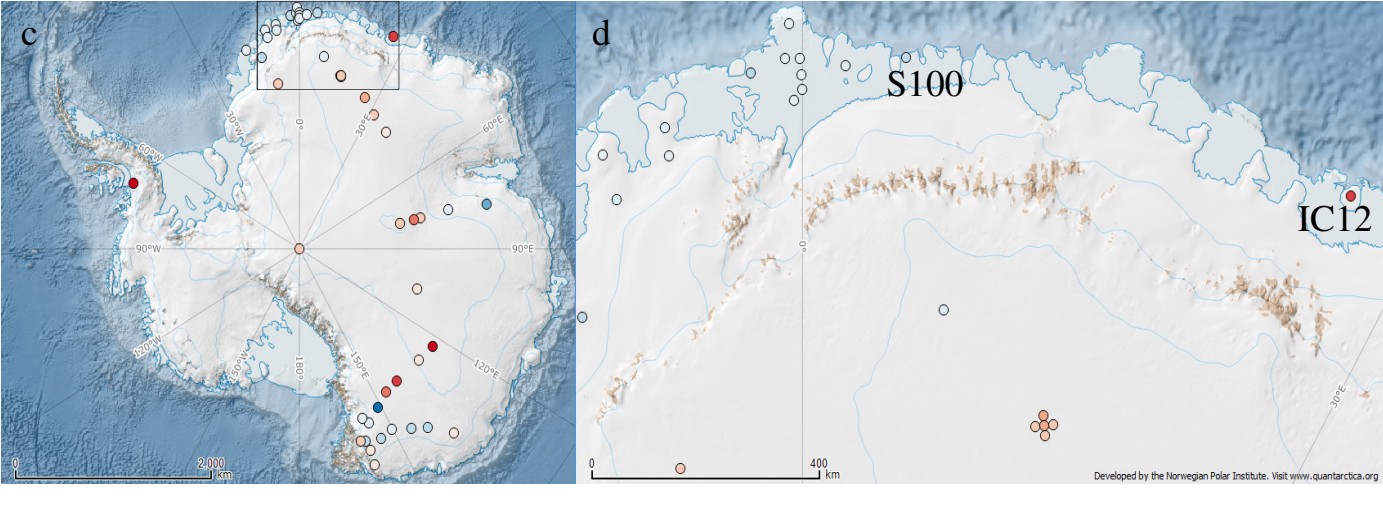

Change in accumulation (%)

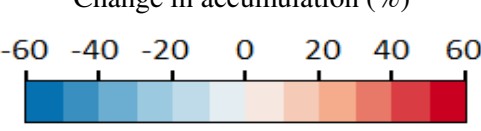

Figure 1: Location of IC12 and other ice cores referred to in the discussion. Change in Accumulation between ~1960-present average compared to ~1816-present average (a-b) and ~1990-present compared to 1816-present (c-d), see Table A1 for exact periods. Panels (b) and (d) are zoomed of the framed zone in panels (a) and (c).



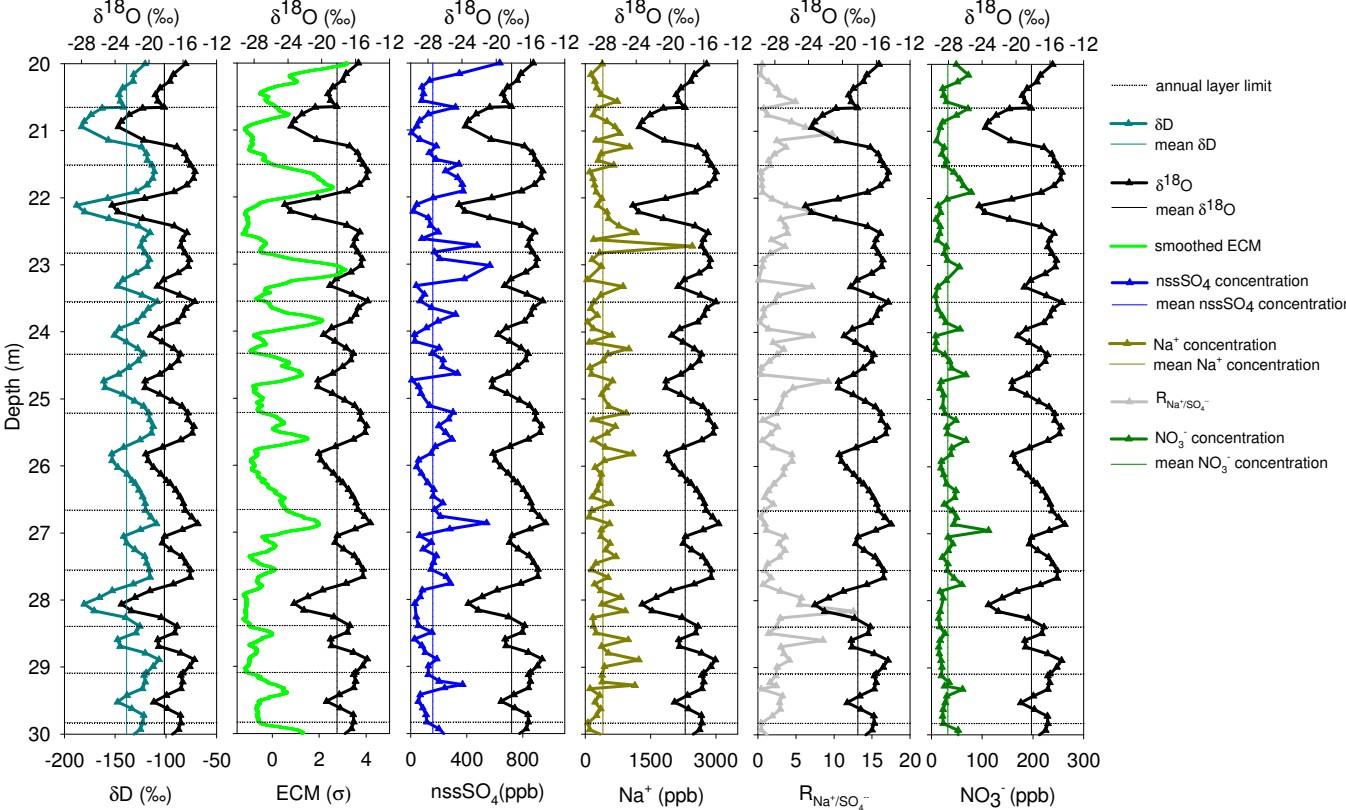

Figure 2. Variations in stable isotopes ($\delta^{18}O$, $\delta D$), smoothed ECM (running mean, 0.1 m), chemical species and their ratios used to constrain annual layer thickness in an example 10 m long section (20 - 30 m depth) of the IC12 ice core. Dashed horizontal lines indicate the annual layer limit (middle of the summer $\delta^{18}O$ peak).



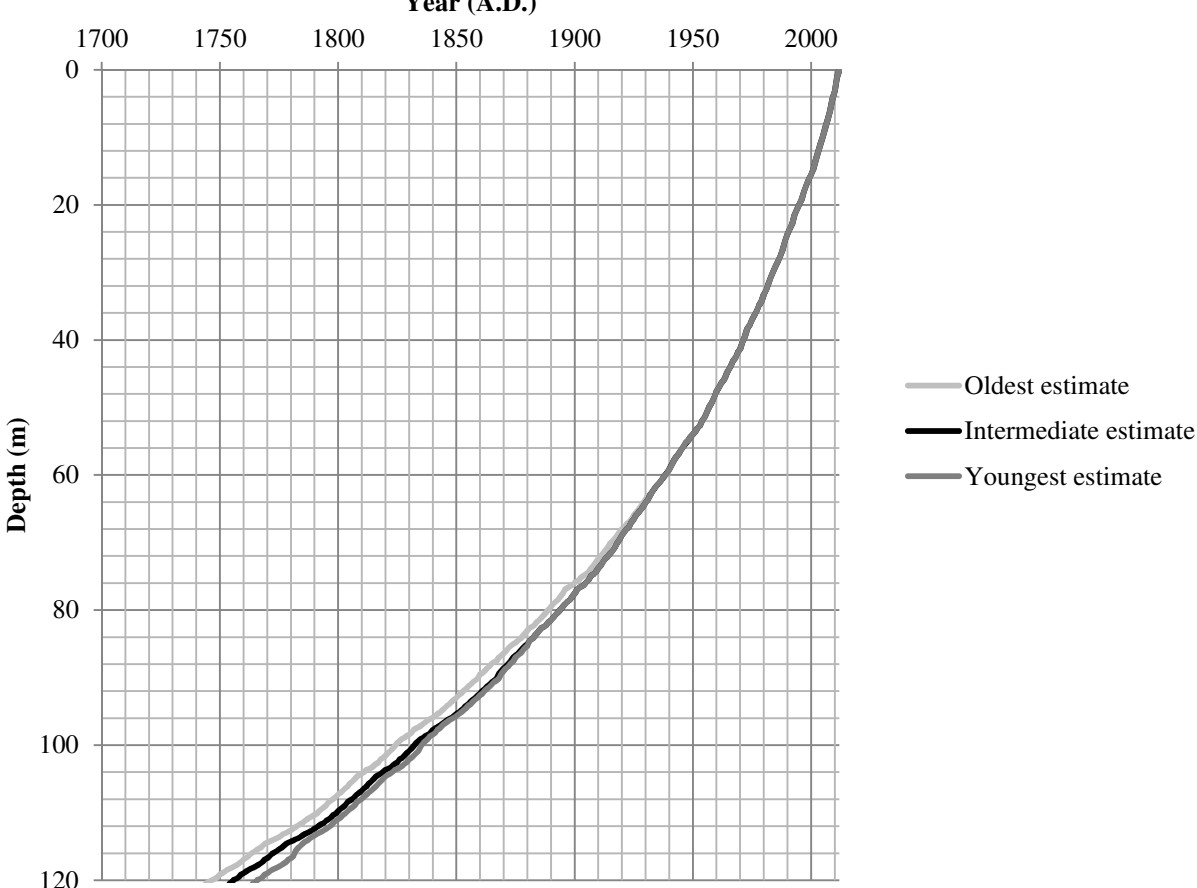

Figure 3. Age-depth relationships reconstructed from the relative dating process. Note that the approach results in no uncertainty above 62.38 m depth (year 1933). At 120 m depth, the uncertainty is ± 10 years.



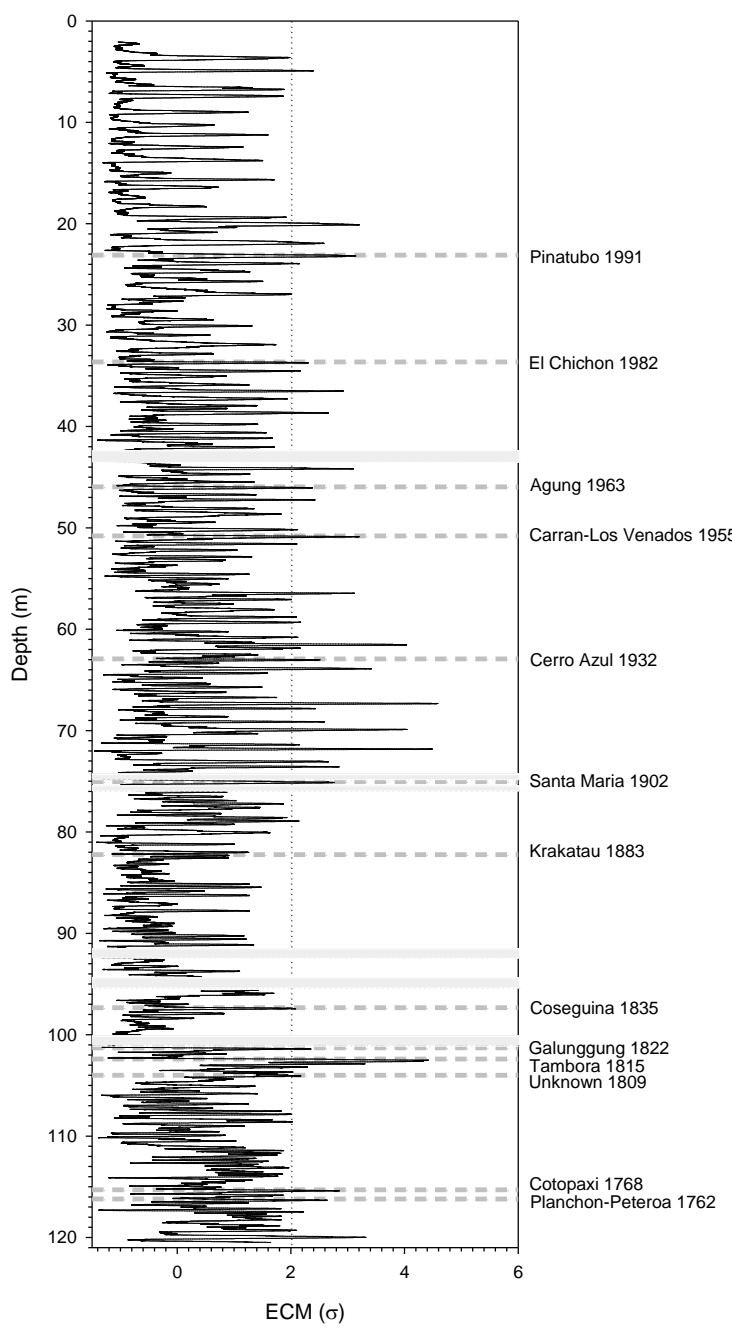

Figure 4. Continuous record of ECM (except for 6 measurement gaps shown as grey bands). Normalized conductivity (black line) is expressed as multiple of standard deviation (σ). The 2σ threshold is shown as a dotted vertical line, and identified volcanic peaks as dashed grey horizontal lines.





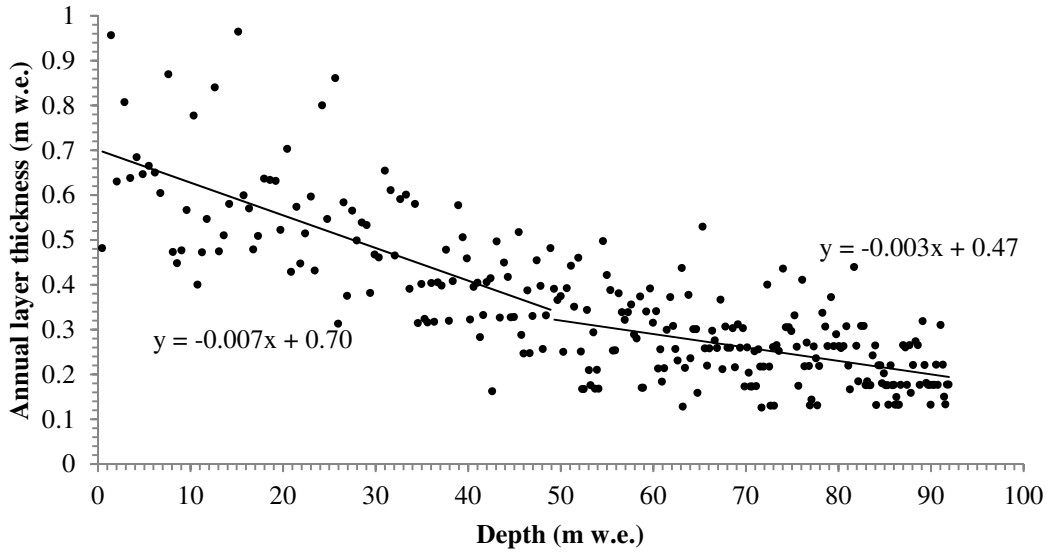

Figure 5. Annual layer thickness plotted against depth. The record is divided into two age/depth ranges, before and after 1900/49 m, for which best-fit straight lines are presented. We use the hypothesis that no temporal drift in annual accumulation existed prior to 1900 (see text for details).



Figure 6. Accumulation rates at IC12. (a) Annual (thin lines with error bars) and average (11 years running mean, thick lines) accumulation rates. The blue lines show uncorrected annual layer thickness in m w.e. The red diamonds highlight years 2009 and 2011 discussed in the text (a-b) Corrected annual layer thicknesses are shown by green lines for the Nye approach and black lines for the Dansgaard and Johnsen approach (see text for details). (b) Dotted horizontal lines represent long-term accumulation (mean plus standard deviation and mean minus standard deviation) for various time periods bounded by specific volcanic eruption events (indicated by vertical lines and bold years).





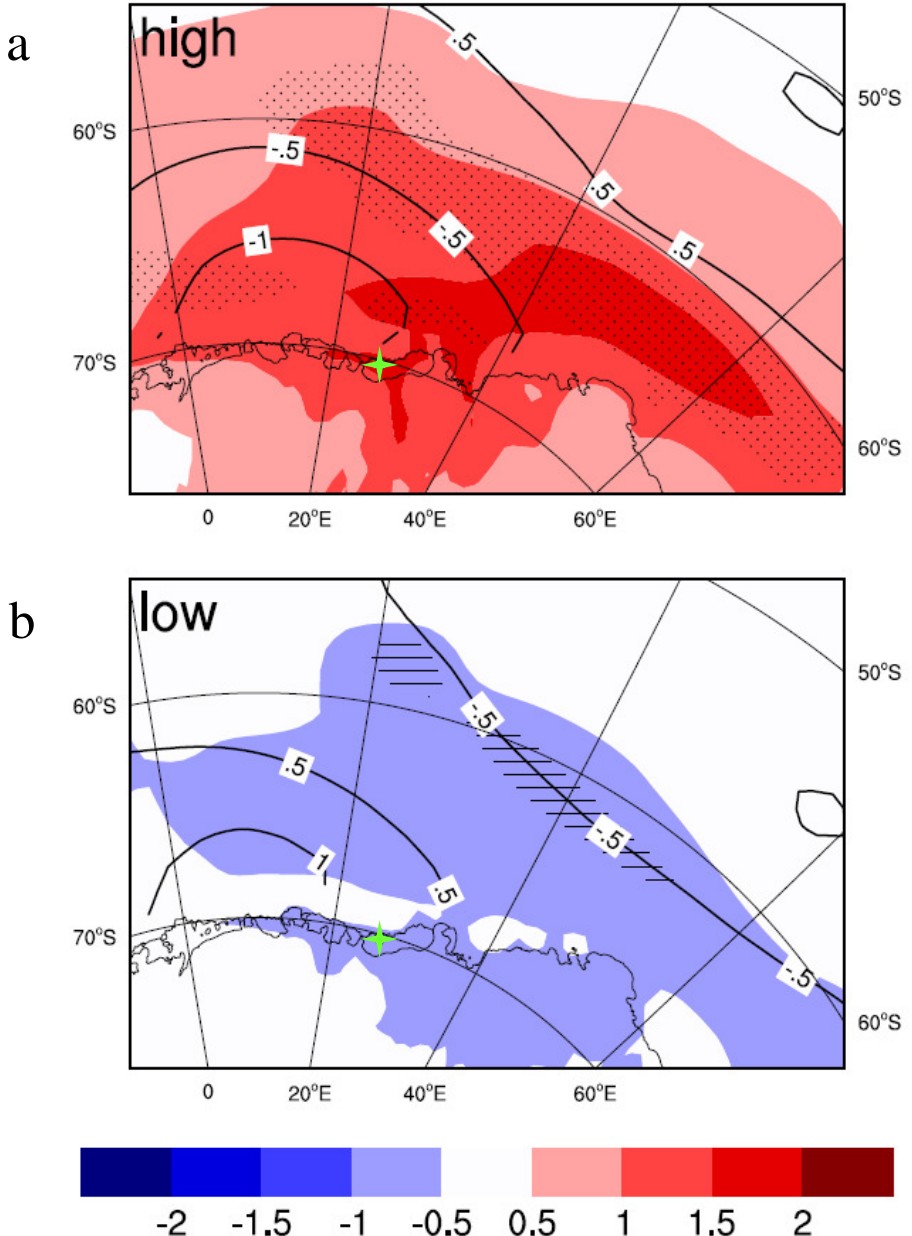

Figure 7. Large-scale atmospheric, ocean and sea-ice anomalies in (a) high-accumulation (10% highest) and (b) low-accumulation (10% lowest) years in the CESM historical time series (1850-2005). The colours show the annual mean near-surface temperature anomaly (in °C), the lines show the surface pressure anomaly (in hPa), and the stippled/hatched areas show the anomaly in sea-ice coverage (stippled areas are areas with >20 days less sea ice cover than the mean, hatched areas show areas with >20 days more sea ice than the mean). The green star shows the location of the ice core.





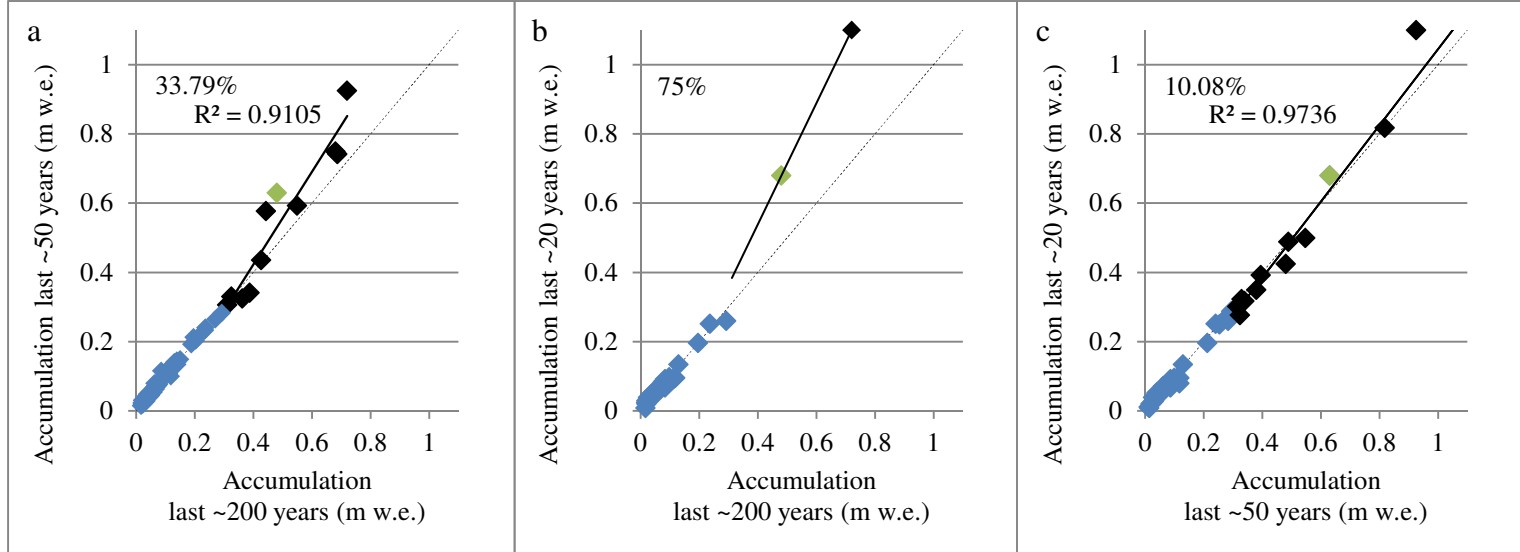

Figure 8. Comparison of SMB during (a) the last ~200 years and the last ~50 years (b) the last ~200 years and the last ~20 years, and (c) the last ~50 years and the last ~20 years. See Table A1 for exact periods. Sites above 0.3 m w.e. a$^{-1}$ are shown in black, except our study site, IC12 shown in green. Sites below 0.3 m w.e. a$^{-1}$ are shown in blue. The black lines show a linear regression through high accumulation sites. Increases in % between the periods compared are shown on the graph with R$^2$ value when relevant. The 1:1 slope (0% change) is shown as a dotted line.





## Appendix A

Table A1. Sites information and snow accumulation values *no significant trend during the 20[th] century **short record: only recent periods are compared ***when only a stacked accumulation change is given, accumulation from individual ice cores are inferred from the stacked record as if it was the same trend for all ice cores. Ref : reference period. Numbers in italic are inferred from the trend given in the referenced paper

| Site name | Latitude | Longitude | Elevation (m a.s.l.) | Reference period | Accumulation ($10^{-3}$ m w.e.) (kg m² a$^{-1}$) | Recent period | Accumulation ($10^{-3}$ m w.e.) (kg m² a$^{-1}$) | Most recent period | Accumulation ($10^{-3}$ m w.e.) (kg m² a$^{-1}$) | % change (50a - ref) | % change (20a - ref) except** | Method | Study |
|---|---|---|---|---|---|---|---|---|---|---|---|---|---|
| Siple Dome | -81.6530 | -148.9980 | 620 | 1890-1994 | 120 | 1922-1991 | 118 | | | -1.67% | | Ice core | Kaspari et al., 2004 |
| ITASE00-5 | -77.6830 | -123.9950 | 1828 | 1716-2000 | 140 | 1922-1991 | 141 | | | 0.71% | | Ice core | Kaspari et al., 2004 |
| ITAE99-1 | -80.6200 | -122.6300 | 1350 | 1724-1998 | 139 | 1922-1991 | 146 | | | 5.04% | | Ice core | Kaspari et al., 2004 |
| ITASE00-4 | -78.0830 | -120.0800 | 1697 | 1799-2000 | 189 | 1922-1991 | 193 | | | 2.12% | | Ice core | Kaspari et al., 2004 |
| RIDS C | -80.0100 | -119.4300 | 1530 | 1903-1995 | 112 | 1970-1995 | 108.35 | | | -3.26% | | Ice core | Kaspari et al., 2004 |
| RIDS B | -79.4600 | -118.0500 | 1603 | 1922-1995 | 150 | 1970-1995 | 149.37 | | | -0.42% | | Ice core | Kaspari et al., 2004 |
| RIDS A | -78.7300 | -116.3300 | 1740 | 1831-1995 | 235 | 1922-1991 | 234 | | | -0.43% | | Ice core | Kaspari et al., 2004 |
| ITASE00-1 | -79.3830 | -111.2390 | 1791 | 1653-2001 | 220 | 1922-1991 | 222 | | | 0.91% | | Ice core | Kaspari et al., 2004 |
| ITASE01-2 | -77.8430 | -102.9100 | 1353 | 1890-2001 | 427 | 1922-1991 | 436 | | | 2.11% | | Ice core | Kaspari et al., 2004 |
| ITASE01-3 | -78.1200 | -95.6460 | 1633 | 1859-2001 | 325 | 1922-1991 | 331 | | | 1.85% | | Ice core | Kaspari et al., 2004 |
| ITASE01-5 | -77.0590 | -89.1370 | 1246 | 1780-2001 | 388 | 1922-1991 | 342 | | | -11.86% | | Ice core | Kaspari et al., 2004 |
| ITASE01-6 | -76.0970 | -89.0170 | 1232 | ** | | 1978-1990 | 395 | 1978-1999 | 392.6 | -0.61% | | Ice core | Kaspari et al., 2004 |
| Gomez | -73.5900 | -70.3600 | 1400 | 1855-2006 | 720 | 1970s-2006 | 925 | 1997-2006 | 1100 | 28.47% | 53% | Ice core | Thomas et al., 2008 |
| Dyer Plateau | -70.6700 | -64.8900 | 2002 | 1790-1989 | 549 | 1969-1989 | 593 | | | 8.00% | | Ice core | Raymond et al., 1996 |
| James Ross Island | -64.2200 | -57.6800 | 1640 | 1847-1980 | 443 | 1964-1990 | 578 | | | 30.47% | | Ice core | Aristarain et al., 2004 |
| R1 | -78.3075 | -46.2728 | 718 | 1816-1998 | 204 ±7 | * | 204 | | | 0.00% | | Ice core | Mulvaney et al., 2002 |
| Berkner B25 | -79.5700 | -45.7200 | 890 | 1816-1956 | 131 | 1965-1994 | 141 | | | 7.63% | | Ice core | Ruth et al., 2004 |
| A | -72.6500 | -16.6333 | 60 | ** | | 1975-1989 | 380 | 1980-1989 | 350 | | -8% | Ice core | Isaksson & Melvold, 2002 |
| E | -73.6000 | -12.4333 | 700 | ** | | 1932-1991 | 324 | 1980-1991 | 277 | | -15% | Ice core | Isaksson & Melvold, 2002; Isaksson et al., 1996 |
| B39 | -71.4100 | -9.9000 | 655 | ** | | 1935-2007 | 818 | 1987-2007 | 818 | | 0.00% | Ice core | Fernandoy et al., 2010 |
| FB0704 | -72.0600 | -9.5600 | 760 | ** | | 1962-2007 | 489 | 1987-2007 | 489 | | 0.00% | Ice core | Fernandoy et al., 2010 |
| BAS-depot | -77.0333 | -9.5000 | 2176 | 1816-1997 | 71 | 1965-1997 | 71 | | | 0.00% | | Ice core | Hofstede et al., 2004 |
| B04 | -70.6200 | -8.3700 | 35 | 1892-1981 | 362 ±95 | 1960-1980 | 325 | | | -10.22% | | Ice core | Schlosser & Oerter, 2002 |
| CV | -76.0000 | -8.0500 | 2400 | 1816-1997 | 62 | 1965-1997 | 68 ±2 | 1992-1997 | 70 | 9.68% | 13% | Ice core | Karlof et al., 2005 |
| B38 | -71.1600 | -6.7000 | 690 | ** | | 1960-2007 | 1257 | 1987-2007 | 1257 | | 0.00% | Ice core | Fernandoy et al., 2010 |
| FB0702 | -71.5700 | -6.6700 | 539 | ** | | 1959-2007 | 547 | 1987-2007 | 500 | | -9% | Ice core | Fernandoy et al., 2010 |
| FB9816 | -75.0000 | -3.5037 | 2740 | 1800-1997 | 47 ±17 | 1950-1997 | 51.5*** | | | 9.57% | | Ice core | Oerter et al., 2000 |
| B31 | -75.5800 | -3.4300 | 2669 | 1816-1997 | 58.4 | 1966-1989 | 59.8 | | | 2.40% | | Ice core | Oerter et al., 2000 |
| H | -70.5000 | -2.4500 | 53 | ** | | 1953-1994 | 480 | 1980-1993 | 425 | | -11% | Ice core | Isaksson & Melvold, 2002 |
| NUS08-2 | -87.8500 | -1.8000 | 2583 | 1815-2007/8 | 67.4 ±2.6 | 1963-2007/8 | 63.4 ±4.2 | | | -5.93% | | Ice core | Anschutz et al., 2011 |
| S32 | -70.3100 | -0.8000 | 53 | ** | | 1995-2009 | 339 ±36 | | 318 | | -6% | Ice core | Schlosser et al., 2014 |
| G3 | -69.8230 | -0.6120 | 57 | ** | | 1993-2009 | 295 ±29 | | 288 | | -2% | Ice core | Schlosser et al., 2014 |
| FB9815 | -74.9492 | -0.5055 | 2840 | 1801-1997 | 59 ±24 | 1950-1997 | 65*** | | | 10.17% | | Ice core | Oerter et al., 2000 |
| G4 | -70.9020 | -0.4020 | 60 | ** | | 1983-2009 | 330 ±21 | | 323 | | -2% | Ice core | Schlosser et al., 2014 |
| M2 | -70.3160 | -0.1090 | 73 | ** | | 1981-2009 | 315 ±22 | | 302 | | -4% | Ice core | Schlosser et al., 2014 |
| G5 | -70.5450 | -0.0410 | 82 | ** | | 1983-2009 | 298 ±21 | | 290 | | -3% | Ice core | Schlosser et al., 2014 |
| K | -70.7500 | 0.0000 | 53 | ** | | 1954-1996 | 254 | 1980-1996 | 250 | | 0% | Ice core | Isaksson & Melvold, 2002 |
| SPS | -90.0000 | 0.0000 | 2850 | 1816-1956 | 76.5 | 1965-1994 | 84.8 ±3.3 | 1992-1997 | 84.5 ±8.9 | 10.85% | 10% | Ice core and poles | Mosley & Thompson, 1999 |
| B32 | -75.0023 | 0.0070 | 2882 | 1816-1997 | 63 | 1966-1997 | 80 | | | 26.98% | | Ice core | |
| EPICA DML | -75.0020 | 0.0680 | 2774 | 1915-2008 | 73 | 1964-2008 | 73.1 ±1.7 | | | 0.14% | | Firn core and radar | Fujita et al., 2011 |
| FB9808 | -74.7507 | 0.9998 | 2860 | 1801-1997 | 68 ±22 | 1950-1997 | 74.5*** | | | 9.56% | | Ice core | Oerter et al., 2000 |
| FB9809 | -74.4992 | 1.9608 | 2843 | 1801-1997 | 89 ±29 | 1950-1997 | 97.5*** | | | 9.55% | | Ice core | Oerter et al., 2000 |
| EPICA (Amundsenisen) | -75.0000 | 2.0000 | 2900 | 1865-1965 | 78 | 1966-1991 | 76 | | | -2.56% | | Ice core | Isaksson et al., 1996 |
| G8 | -70.4100 | 2.0100 | 58 | ** | | 1991-2009 | 282 ±26 | | 273 | | -3% | Ice core | Schlosser et al., 2014 |
| FB9814 | -75.0837 | 2.5017 | 2970 | 1801-1997 | 64 ±21 | 1950-1997 | 71*** | | | 10.94% | | Ice core | Oerter et al., 2000 |
| C | -72.2583 | 2.8911 | 2400 | 1955-1996 | 119 | 1965-1996 | 123 | | | 3.36% | | Ice core | Isaksson et al., 1999 |
| D | -72.5083 | 3.0000 | 2610 | 1955-1996 | 112 | 1965-1996 | 116 | | | 3.57% | | Ice core | Isaksson et al., 1999 |
| DML08 | -75.7528 | 3.2828 | 2971 | 1919-96 | 60 ±19 | * | 60 | | | 0.00% | | Ice core | Oerter et al., 1999 |
| E | -72.6750 | 3.6628 | 2751 | 1955-1996 | 55 | 1965-1996 | 59 | | | 7.27% | | Ice core | Isaksson et al., 1999 |
| DML02 | -74.9683 | 3.9185 | 3027 | 1919-95 | 59 ±14 | * | 59 | | | 0.00% | | Ice core | Oerter et al., 1999 |
| FB9810 | -74.6672 | 4.0017 | 2980 | 1801-1997 | 86 ±29 | 1950-1997 | 94.5*** | | | 9.88% | | Ice core | Oerter et al., 2000 |
| F | -72.8583 | 4.3514 | 2840 | 1955-1996 | 23 | 1965-1996 | 24 | | | 4.35% | | Ice core | Isaksson et al., 1999 |
| S100 | -70.2333 | 4.8000 | 48 | 1816-2000 | 292 | 1956-2000 | 284 | 1991-2000 | 260 ±80 | -2.74% | -11% | Ice core | Kaczmarska et al., 2004 |
| S20 | -70.2417 | 4.8111 | 63 | 1955-1996 | 271 | 1965-1996 | 265 | | | -2.21% | | Ice core | Isaksson et al., 1999 |
| FB0601 | -75.2470 | 4.8440 | 3090 | 1915-2008 | 52 | 1964-2008 | 51.6 ±1.2 | | | -0.77% | | Firn core and radar | Fujita et al., 2011 |
| FB9813 | -75.1673 | 5.0033 | 3100 | 1816-1997 | 48 | 1950-1997 | 53*** | | | 10.42% | | Ice core | Oerter et al., 2000 |
| G | -73.0417 | 5.0442 | 2929 | 1955-1996 | 28 | 1965-1996 | 30 | | | 7.14% | | Ice core | Isaksson et al., 1999 |
| FB9804 | -75.2503 | 6.0000 | 2630 | 1801-1997 | 50 ±16 | 1950-1997 | 55*** | | | 10.00% | | Ice core | Oerter et al., 2000 |
| H | -73.3917 | 6.4606 | 3074 | 1955-1996 | 44 | 1965-1996 | 46 | | | 4.55% | | Ice core | Isaksson et al., 1999 |





| Site name | Latitude | Longitude | Elevation (m a.s.l.) | Reference period | Accumulation (10⁻³ m w.e.) (kg m² a⁻¹) | | Recent period | Accumulation (10⁻³ m w.e.) (kg m² a⁻¹) | | Most recent period | Accumulation (10⁻³ m w.e.) (kg m² a⁻¹) | | % change (50a - ref) | % change (20a - ref) except** | Method | Study |
|---|---|---|---|---|---|---|---|---|---|---|---|---|---|---|---|---|
| B33 | -75.1670 | 6.4985 | 3160 | 1816-1997 | 45.9 | | 1966-1989 | 55 | | | | | 19.83% | | Ice core | Oerter et al., 2000, Sommer et al., 2000 |
| FB9811 | -75.0840 | 6.5000 | 3160 | 1801-1997 | 58 | ±16 | 1950-1997 | 64*** | | | | | 10.34% | | Ice core | Oerter et al., 2000 |
| DML09 | -75.9333 | 7.2130 | 3156 | 1897-1996 | 45 | ±12 | * | 45 | | | | | 0.00% | | Ice core | Oerter et al., 1999 |
| DML10 | -75.2167 | 7.2130 | 3364 | 1900-96 | 47 | ±11 | * | 47 | | | | | 0.00% | | Ice core | Oerter et al., 1999 |
| DML04 | -74.3990 | 7.2175 | 3179 | 1905-1996 | 53 | ±15 | * | 53 | | | | | 0.00% | | Ice core | Oerter et al., 1999 |
| I | -73.8008 | 7.9406 | 3174 | 1955-1996 | 52 | | 1965-1996 | 53 | | | | | 1.92% | | Ice core | Isaksson et al., 1999 |
| NUS07-1 | 74.7200 | 7.9800 | 3174 | 1815-2007/8 | 52 | ±2 | 1963–2007/08 | 55.9 | ±3.9 | | | | 7.50% | | Ice core | Anschutz et al., 2009 |
| Site I | -73.7167 | 7.9833 | 3174 | 1815-2007 | 52 | ±1.3 | 1963-2007 | 56 | ±4.7 | 1991-2007 | 52 | | 7.69% | 0% | Ice core | Anschutz et al., 2009 |
| DML06 | -75.0007 | 8.0053 | 3246 | 1899-1996 | 50 | ±14 | * | 50 | | | | | 0.00% | | Ice core | Oerter et al., 1999 |
| NUS08-6 | -81.7000 | 8.5700 | 2447 | 1815-2007/8 | 39.2 | ±1.5 | 1963-2007/8 | 49.2 | ±3.4 | | | | 25.51% | | Ice core | Anschutz et al., 2011 |
| J | -74.0417 | 9.4917 | 3268 | 1955-1996 | 44 | | 1965-1996 | 45 | ±4 | | | | 2.27% | | Ice core | Isaksson et al., 1999 |
| FB0603 | -75.1170 | 9.7240 | 3300 | 1915-2008 | 41 | | 1964-2008 | 38 | ±0.9 | | | | -7.32% | | Firn core and radar | Fujita et al., 2011 |
| K | -74.3583 | 11.1036 | 3341 | 1955-1996 | 45 | | 1965-1996 | 41 | | | | | -8.89% | | Ice core | Isaksson et al., 1999 |
| L | -74.6417 | 12.7908 | 3406 | 1955-1996 | 45 | | 1965-1996 | 41 | | | | | -8.89% | | Ice core | Isaksson et al., 1999 |
| A28 | -74.8617 | 14.7420 | 3466 | 1915-2008 | 44 | | 1964-2008 | 44.5 | ±1 | | | | 1.14% | | Firn core and radar | Fujita et al., 2011 |
| MC | -75.0112 | 14.8865 | 3470.4 | 1816-1884 | 40 | | 1955-2000 | 39 | | 1992-2000 | 46 | | -2.50% | 15% | Ice core | Karlof et al., 2005 |
| MD | -74.9706 | 14.9567 | 3470.8 | 1816-1884 | 42 | | 1955-2000 | 40 | | 1992-2000 | 53 | | -4.76% | 26% | Ice core | Karlof et al., 2005 |
| M | -75.0000 | 14.9964 | 3470 | 1816-1884 | 41 | ±0.7 | 1955-2000 | 41 | ±0.5 | 1992-2000 | 50 | ±1.1 | 0.00% | 22% | Ice core | Karlof et al., 2005 |
| M150 | -74.9900 | 15.0000 | 3470 | 1816-1997 | 43 | | 1965-1997 | 48.5 | | | | | 12.79% | | Ice core | Hofstede et al., 2004 |
| M | -74.9917 | 15.0017 | 3453 | 1955-1965 | 51 | | 1965-1996 | 45 | | | | | -11.76% | | Ice core | Isaksson et al., 1999 |
| MB | -75.0294 | 15.0435 | 3470.4 | 1816-1884 | 39 | | 1955-2000 | 42 | | 1992-2000 | 46 | | 7.69% | 18% | Ice core | Karlof et al., 2005 |
| MA | -74.9887 | 15.1134 | 3470.4 | 1816-1884 | 42 | | 1955-2000 | 42 | | 1992-2000 | 48 | ±1.3 | 0.00% | 14% | Ice core | Karlof et al., 2005 |
| NUS08-5 | -82.6300 | 17.8700 | 2544 | 1815-2007/8 | 35 | ±0.8 | 1963-2007/8 | 37.6 | ±2.3 | | | | 7.43% | | Ice core | Anschutz et al., 2011 |
| NUS08-4 | -82.8167 | 18.9000 | 2552 | 1815-2007/8 | 36.7 | ±0.9 | 1963-2007/8 | 36.1 | ±2.1 | | | | -1.63% | | Ice core | Anschutz et al., 2011 |
| NUS08-3 | -84.1300 | 22.0000 | 2625 | 1815-2007/8 | 40.1 | ±1 | 1963-2007/8 | 45.3 | ±3.1 | | | | 12.97% | | Ice core | Anschutz et al., 2011 |
| A35 | -76.0660 | 22.4590 | 3586 | 1915-2008 | 35 | | 1964-2008 | 39.2 | | | | | 12.00% | | Firn core and radar | Fujita et al., 2011 |
| NUS07-2 | -76.0700 | 22.4700 | 3582 | 1815-2007/8 | 33 | ±0.7 | 1963-2007/8 | 28 | ±2 | | | | -15.15% | | Ice core | Anschutz et al., 2011 |
| MP | -75.8880 | 25.8340 | 3661 | 1286-2008 | 33.1 | ±1.0 | 1964-2008 | 38.7 | ±0.9 | 1993-2008 | 41.9 | ±2.8 | 16.92% | 27% | Firn core and radar | Fujita et al., 2011 |
| NUS07-3 | -77.0000 | 26.0500 | 3589 | 1815-2007/8 | 22 | ±0.5 | 1963-2007/8 | 23.7 | ±1.7 | | | | 7.73% | | Ice core | Anschutz et al., 2009 |
| IC12 | -70.2458 | 26.3349 | 450 | 1816-2012 | 480 | ±10 | 1955-2012 | 630 | ±20 | 1992-2012 | 680 | ±70 | 31.25% | 42% | Ice core | This paper |
| DK190 | -76.7940 | 31.9000 | 3741 | 1286-2008 | 28.7 | ±0.9 | | | | 1993-2008 | 34.1 | ±2.3 | | 19% | Firn core and radar | Fujita et al., 2011 |
| NUS07-4 | -78.2167 | 32.8500 | 3595 | 1815-2007/8 | 19 | ±0.5 | 1963-2007/8 | 17.5 | ±1.2 | | | | -7.89% | | Ice core | Anschutz et al., 2009 |
| NUS07-5 | -78.6500 | 35.6300 | 3619 | 1815-2007/8 | 24 | ±0.5 | 1963-2007/8 | 20.1 | ±1.4 | | | | -16.25% | | Ice core | Anschutz et al., 2011 |
| DF | -77.3170 | 39.7030 | 3810 | 1816-2001 | 26.3 | | 1964-2008 | 28.8 | ±0.7 | 1995-2006 | 27.3 | ±0.4 | 9.51% | 4% | Ice core | Igarashi et al., 2011 |
| YM85 | -71.5800 | 40.6300 | 2246 | 1816-2002 | 140 | | 1965-2002 | 135 | | | | | -3.57% | | Ice core | Takahashi et al., 2009 |
| H72 | -69.2047 | 41.0906 | 1214 | 1831-1998 | 311 | | 1973-1998 | 307 | | | | | -1.29% | | Ice core and poles | Nishio et al., 2002 |
| NUS07-6 | -80.7833 | 44.8500 | 3672 | 1815-2007/8 | 22 | | 1902-2007/8 | 21 | | | | | -4.55% | | Ice core | Anschutz et al., 2009 |
| G15 | -71.2000 | 45.9800 | 2544 | 1816-1964 | 86 | | 1964-1984 | 116 | | | | | 34.88% | | Ice core | Moore et al., 1991 |
| NUS07-8 | -84.1833 | 53.5333 | 3452 | 1815-2007/8 | 32 | ±1.2 | 1963-2007/8 | 30 | ±2.1 | | | | -6.25% | | Ice core | Anschutz et al., 2009 |
| NUS07-7 | -82.0700 | 54.5500 | 3725 | 1815-2007/8 | 29.4 | ±0.6 | 1963-2007/8 | 26.1 | ±1.9 | | | | -11.22% | | Ice core | Anschutz et al., 2011 |
| DT217 | -75.7167 | 76.8333 | 2800 | ** | | | 1998-2008 | 12 | ±1.72 | 2005-2008 | 12 | | | 0% | Stake arrays | Ding et al., 2011 |
| DT364 | -78.3333 | 77.0000 | 3380 | ** | | | 1999-2008 | 62 | ±0.14 | 2005-2008 | 72 | | | 16% | Stake arrays | Ding et al., 2011 |
| DT401 | -79.0200 | 77.0000 | 3760 | 1816-1999 | 19 | | 1963-1999 | 24 | | 1999–2005 | 25 | ±16 | 26.32% | 32% | Ice core | Ren et al., 2010; Ding et al., 2011a |
| DT001 | -70.8300 | 77.0700 | 2325 | 1810-1959 | 131 | | 1959-1996 | 131 | | | | | 0.00% | | Ice core | Zhang et al., 2006 |
| Dome A | -80.3667 | 77.3500 | 4093 | ** | | | 2005-2008 | 19 | ±0.25 | 2008-2009 | 21 | | | 11% | Stake arrays | Ding et al., 2011 |
| DomeA | -80.3600 | 77.3600 | 4092 | 1815-1998 | 23 | | 1963-1998 | 23 | | | | | 0.00% | | Ice core | Jiang et al., 2012 |
| LGB65 | -71.8500 | 77.9200 | 1850 | 1815-1996 | 131 | | 1960-1996 | 131 | | | | | 0.00% | | Ice core | Xiao et al., 2004 |
| DT008 | -72.1667 | 77.9333 | 2390 | ** | | | 1998-2008 | 118 | ±0.30 | 2005-2008 | 80 | | | -32% | Stake arrays | Ding et al., 2011 |
| VOSTOK | -78.4500 | 106.8300 | 3488 | 1816-2010 | 20.6 | ±0.3 | 1955-2010 | 21.5 | ±0.5 | 1958-2010 | 20.8 | | 4.37% | 1% | Snow pits and poles | Ekaykin et al., 2004 |
| DSS | -66.7697 | 112.8069 | 1370 | 1816-2000 | 680 | | 1970-2009 | 750 | | | | | 10.29% | | Ice core | Roberts et al., 2015 |
| LAW DOME | -66.7700 | 112.9800 | 1370 | 1816-1966 | 687 | | 1966-2005 | 742 | | | | | 8.01% | | Ice core | Morgan et al., 1991; van Ommen & Morgan, 2010 |
| DomeC | -75.1200 | 123.3100 | 3233 | 1816-1998 | 25.3 | | 1965-1998 | 28.3 | | 1996-1998 | 39 | | 11.86% | 54% | Ice core and poles | Frezzotti et al., 2005 |
| D6 A | -75.4400 | 129.8100 | 3027 | 1816-1998 | 36 | ±1.8 | 1966-1998 | 29 | ±1.4 | 1998-2002 | 39 | | -19.44% | 8% | Ice core and poles | Frezzotti et al., 2005 |
| D66 | -68.9400 | 136.9400 | 2333 | 1966-1864 | 196 | | 1965-2001 | 213 | ±13 | 2001-2003 | 197 | | 8.67% | 1% | Ice core and poles | Magand et al., 2004; Frezzotti et al., 2013 |
| D2 A | -75.6200 | 140.6300 | 2479 | 1816-1998 | 20 | ±1.0 | 1966-1998 | 31 | ±1.6 | 1998-2002 | 30 | | 55.00% | 50% | Ice core and poles | Frezzotti et al., 2005 |
| GV1 | -70.8700 | 141.3800 | 2244 | 1816-2001 | 114 | | 1965-2001 | 117 | ±7 | 2001-2003 | 96 | | 2.63% | -16% | Ice core and poles | Magand et al., 2004; Frezzotti et al., 2013 |
| GV2 | -71.7100 | 145.2600 | 2143 | 1816-2001 | 112 | | 1965-2001 | 112 | ±7 | 2001-2003 | 92 | | 0.00% | -18% | Ice core and poles | Magand et al., 2004; Frezzotti et al., 2013 |
| MdPtA | -75.5300 | 145.8600 | 2454 | 1816-1998 | 36 | ±1.8 | 1966-1998 | 45 | ±2.7 | 1998-2010 | 47 | | 25.00% | 31% | Ice core and poles | Frezzotti et al., 2005 |




| Site name | Latitude | Longitude | Elevation (m a.s.l.) | Reference period | Accumulation ($10^{-3}$ m w.e.) (kg m² a⁻¹) | | Recent period | Accumulation ($10^{-3}$ m w.e.) (kg m² a⁻¹) | | Most recent period | Accumulation ($10^{-3}$ m w.e.) (kg m² a⁻¹) | % change (50a - ref) | % change (20a - ref) except** | Method | Study |
|---|---|---|---|---|---|---|---|---|---|---|---|---|---|---|---|
| GV3 | -72.6300 | 150.1700 | 2137 | 1816-2001 | 81 | | 1965-2001 | 84 | ±5 | 2001-2003 | 73 | 3.70% | -10% | Ice core and poles | Magand et al., 2004;Frezzotti et al., 2013 |
| M2 A | -74.8000 | 151.2700 | 2278 | 1816-1998 | 17 | ±0.8 | 1966-1998 | 15 | ±7.5 | 1998-2002 | 8.5 | -11.76% | -50% | Ice core and poles | Frezzotti et al., 2005 |
| GV4 | -72.3900 | 154.4800 | 2126 | 1816-2001 | 119 | | 1965–2001 | 100 | ±6 | 2001–2003 | 96 | -15.97% | -19% | Ice core and poles | Magand et al., 2004;Frezzotti et al., 2013 |
| 31DPT A | -74.0300 | 155.9600 | 2069 | 1816-1998 | 98 | ±4.9 | 1966-1998 | 112 | ±5.6 | 1998-2002 | 98 | 14.29% | 0% | Ice core and poles | Frezzotti et al., 2005 |
| GPS2A | -74.6400 | 157.5020 | 1804 | 1816-1998 | 60 | ±3.0 | 1966-1998 | 54 | ±2.7 | 1993-2000 | 55 | -10.00% | -8% | Ice core and poles | Frezzotti et al., 2005 |
| GV5 | -71.8900 | 158.5400 | 2184 | 1816-2001 | 129 | | 1965-2001 | 129 | ±7 | 2001-2004 | 135 | 0.00% | 5% | Ice core and poles | Magand et al., 2004;Frezzotti et al., 2007 |
| GV7 | -70.6800 | 158.8600 | 1947 | 1854-2001 | 237 | | 1965-2001 | 241 | ±13 | 2001-2004 | 252 | 1.69% | 6% | Ice core and poles | Magand et al., 2004;Frezzotti et al., 2007 |
| Talos Dome | -72.7700 | 159.0800 | 2316 | 1816-2001 | 83.6 | | 1966-1996 | 86.6 | | 2001-2010 | 68 | 3.59% | -19% | Ice core and poles | Magand et al., 2004;Frezzotti et al., 2007; 2013 |
| Talos Dome | -72.8000 | 159.1000 | 2246 | 1816-1996 | 83.6 | | 1966-1996 | 86.6 | | 1992-1996 | 92.5 | 3.59% | 11% | Ice core | Stenni et al., 2002 |
| Hercules Neve | -73.1000 | 165.4000 | 2960 | 1816-1966 | 118 | | 1966-1992 | 129 | | | | 9.32% | | Ice core | Stenni et al., 1999 |