# Peer review of "Ice core evidence for a 20th century increase in surface mass balance in coastal Dronning Maud Land, East Antarctica"

_The Cryosphere, 2016_

## Referee Comment (RC1) · Anonymous Referee #1 · 23 Mar 2016

Review: Philippe et al., Ice core evidence for a recent increase in snow accumulation in coastal Dronning Maud Land, East Antarctica

This paper reconstructs the accumulation-rate history for a site in coastal East Antarctica using a 120m ice/firn core. The authors find an increase in accumulation rate in the past few decades compared to the previous ∼200 years. The authors suggest the increase is due to recent warming, supporting model projections of increased accumulation in Antarctica as the world warms. Unfortunately, flaws in the layer correction and timescale development raise major questions about the inferred increase in accumulation. This paper should be rejected.

The paper has much potential but was not yet ready to be submitted. In particular, the

underdeveloped timescale and inaccurate correction for layer thinning raise questions about the inferred increase in accumulation. The ice-core records presented here are potentially very interesting, but to become publishable, this paper needs considerably more analysis, discussion, and possibly even more lab measurements.

Major Issues This paper has three major issues: 1) the timescale is not convincing, 2) the description of the layer thinning correction is too small and 3) the climate implications are underdeveloped.

1 - Timescale The development of the timescale is the crux of this study and it is clear that considerable effort has gone into the timescale. The paper makes it clear, without explicitly stating so, that the timescales was quite difficult to develop. I have great sympathy for anyone who develops ice-core timescales. However, the timescale as presented is not convincing for two reasons. First, the isotope sampling is too low to resolve annual layers for much of the core. At a sampling interval of 10cm (above 80m), this yields only 5 or 6 samples per year for much of the timescale given the accumulation rates. You need about twice that to resolve clear annual layers, especially on a proxy such as oxygen isotopes that have relatively noisy seasonal cycles. Statements like "no ambiguity in layer counting is detectable above 62.38 m (i.e. 1933 AD)" are in direct contradiction with the need to perform major ion analysis "for sections of unclear isotopic seasonality" and I can see ambiguity in Figure 2 (near 20 and 29 m depths). It seems odd to me that for a relatively short core, the whole thing wasn't sampled at much finer resolution (water isotope analyses are cheap and don't need much ice) and that aerosol analysis wasn't performed on the full core.

Second, the volcanic matches are not convincing. In Figure 4 it appears that any small peak that rises past the 2sigma level is considered a volcano if it happens to be of the correct age. This may be because the data are normalized before identifying volcanic peaks. Regardless of the normalization issue, using ECM (or sulfur) at coastal sites to identify volcanic events is very difficult because the volcanic signal gets overwhelmed by marine inputs.

The timescale is the crux of this paper. This means considerably more effort needs to be made to describe it convincingly, for both the annual layers and volcanic matches. Items that this paper needs:

1) A clear description of what measurements were made at what depths (i.e. where were aerosols measured and show the ambiguities and how they were interpreted)

2) An analysis of the impact of the low sampling resolution on the ability to resolve annual layers (often, low resolution leads to picking false peaks)

3) A realistic assessment of annual-layer interpretation uncertainty

4) A critical assessment of volcanic matches. I.e. why is Cerro Azul 1932 not one of the bigger, yet unmatched, peaks about a meter above or below. (this same question applies to pretty much every match, except for possibly Tambora).

5) A description of why ECM loses the annual signal yet preserves the volcanic signal

6) Why Krakatau isn't observable in the ECM record and what the distinctive characteristics in the aerosol record are that allow it to be identified. Also a description of why the technique to identify Kratatau wasn't applied for the full core.

It sounds like the only truly identifiable volcanic event was Tambora. The authors need to make use of Tambora, and pattern with the unknown 1809 eruption, to make a strong case that this is indeed properly matched (Figure 4 does not do this). Plot it against high resolution ECM/Sulfur/Sulfate records of this event. If the authors can demonstrate that this is a clearly identifiable match, then it would strongly support their annual layer interpretation.

2- The corrections for flow-induced layer thinning reveal a lack of understanding of how ice flow and are clearly underestimated. In particular, it is disappointing that the authors don't make use of the detailed ice-flow modeling that's been done on Derwael Ice Rise (Drews et al., 2015) to develop the vertical thinning function. It is clear from phase sensitive radar measurements (Kingslake et al., 2014) that the simple approximations

for vertical thinning have trouble replicating the vertical strain pattern under ice divides. The Nye assumption is so obviously not applicable to Derwael Ice Rise, which has a distinctive Raymond Bump, that it should not even be considered. The authors' don't supply the kink height value of the D-J model. Using a kink height of nearly 1, Kingslake et al. could still not match the pattern under Roosevelt Island. Since the authors say the kink height is below the zone of interest, I can infer that they didn't use anything greater than $\sim$0.7. This will lead to an underestimation of the amount of strain experienced by the ice in the core. Thus the older accumulation rates will be underestimated, and it will appear that there has been an increase in modern accumulation rates. The underestimation is likely exacerbated by the preferred thinning rate of 3cm per year for the ice rise (Drews et al., 2015). Getting the thinning right is critical to primary conclusion of this paper.

3- The discussion of atmospheric and sea-ice patterns seems like an after thought. I'm not sure why the authors choose to analyze only anomalously high and low years. I also wonder why the authors don't compare the inferred accumulation rate history to the climate reanalyses. Other cores (e.g. Medley et al., 2013, GRL; Morris et al., 2015, Nature Geosciences) find good correlation of annual accumulation.

Below are a few additional points that caught my attention.

P1, L27-29 – your data do not actually support this because your thinning correction is much too small.

P2, L10 – you should mention timescales. Frieler et al only address glacial-interglacial changes. The most directly comparable ice-core record to yours is from Law Dome, which does not show a consistent relationship between accumulation and temperature in the Holocene.

P3L1 – in the ice-core community, continuous is generally used to mean a melting system where discrete samples are not cut. Using continuous to mean that all of the core has been sampled discreetly is confusing.

P3,L13 – Be specific about what you mean by local ice flow. You should really mention that it's an ice rise with a well developed Raymond Bump that has likely been stable for thousands of years.

P4,L25 – DC-ECM does not depend on the impurity content at the crystal boundaries. It depends on the acidity.

P4,L28 – a ∼30cm smoothing window seems really large to me.

P5,L1-3 – Did you normalize the data before identifying the volcanic peaks? If so, you can no longer reliably identify volcanic events with a threshold because the increased conductance of volcanic events would impact the normalization. Is the 2sigma threshold then for the entire data set. I'm confused, but I think this is a major problem.

P5,L5-22 – This section should be entirely redone. Get a thinning function from the ice-flow modelers working on Derwael Ice Rise.

P6,L17 – explain what changes in the ECM and why

P6,L27 – explain how Krakatau was identified

P10,L11 – why are the uncertainties being presented after the results

P10,L30 – there is a lot complexity in the position of the divide, the Raymond Bump, and the minimum accumulation (which is offset from the divide). There needs to be a much more detailed discussion of whether small (i.e. one ice thickness)

---

## Referee Comment (RC2) · Anonymous Referee #2 · 30 Mar 2016

The paper investigates the surface mass balance (SMB) at Derwael Ice Rise, Fimbulisen, Dronning Maud Land, using an ice core drilled at the top of the ice rise. The authors describe their dating methods in some detail and investigate trends in the SMB. They find a positive trend in SMB in the last century, which they state is expected by climate models. They also try to explain this trend by sea ice, temperature, and surface pressure data from a General Circulation Model (GCM). They report that the presented core is the first coastal core in East Antarctica that shows a steady increase in SMB since the beginning of the 20th century.

The paper is generally interesting and the ice core certainly yields valuable new data from a still quite poorly explored area. The English is mostly correct, even though

sometimes the wording is a bit awkward and some formulations are not clear or ambiguous.

However, there are several important points that need to be reconsidered. I would recommend major revision.

General comments:

I have only basic knowledge in dating ice cores using flow models, so I cannot assess the critics of referee #1 considering this point. The authors do show both the uncorrected data and the correction with the different models, so the reader can assess what they have done. Also, their main conclusion (positive SMB trend in the last 100 years) would still be valid for any calculation of layer thinning that lies between the two methods they use.

However, I share Referee #1's doubts about the details of the dating, particularly the use of volcanic horizons, since the attribution of the ECM peaks in Figure 4 to the different eruptions is not convincing, except for Tambora. Also, the authors do not give details about the layer counting using stable isotopes, to which depth this was possible etc. Nobody expects a perfect dating of an ice core because this hardly ever exists. However, I think the authors should discuss the error possibilities of the dating a bit more and give a more realistic quantitative estimate of the error. Probably, within the error bounds, their main result would hold. But, see above, I cannot assess the details of the used models. The authors state that their findings (increase in SMB in a coastal East Antarctic core) are the first ones that support model predictions. This does not make them discuss how representative their results are. They compare their results with other firn/ice cores, but do not compare the temporal variations of the SMB derived from the core with temporal variations of measured and/or modelled air temperature, sea ice, or surface pressure data). Instead they look at composites for very positive and very negative years, which is, in principal, not a bad thing to do, but I would expect stronger signals here in order to be convincing. The arguments using the output from

the Community Earth System Model are a bit weak. The discussion of the atmospheric dynamics involved is not clear and mixes up conditions at the coast and in the interior of Antarctica. Also, different time scales are mixed together and often it is not clear, which time period is meant when certain trends are reported.

Specific comments:

Title: what does "recent" mean?, and, to be correct, "snow accumulation" should be "surface mass balance".

Abstract: It would be good to re-write the abstract after the main text has been revised.

P2:

l5: increasing ice discharge

l8: What does the Polvani paper have to do with warming- related increase in precip? There are other papers that involve data and modelling and do not find either warming or increase in precipitation in the considered period. Please, make sure that it is clear about which time period you are talking.

l23: "both authors concluded that the trends were insignificant". This is not correct and not exact. Which trends? Altnau et al. found a statistically significant positive trend in SMB for the interior DML.

P3: L10ff: grammar: in your sentence, "which" refers to the project.

L12: a local flow regime

How high is the accumulation rate? It would be good to give this information already here.

P4:

L3: do you mean 30mm x 30mm?

L13: the boundary between annual layers
L21: better: were carried out

P5:

L5: snow burial: better: the compression of the snow under its own weight It would be interesting to see the density profile here, maybe you could add this in a figure. I also miss some information about the depth until which seasonal variations in the isotope ratios can be resolved.

P6:

L3: how reliable are the CESM data for the 19th century, especially sea ice?

L24: better : mainly derived from...

P7:

1ff: see above. The volcanic peaks in Figure 4 seem to be pretty ambigous in most cases.

P8:

L15ff: This is a very short and simplified view. The sea ice argument is not convincing, especially the hatched area of anomalies is fairly small and should not have a large impact on precipitation amounts. A decrease in surface pressure of not much more than 1hPa is not very much, even in a composite, and in that case, lower surface pressure does not necessarily mean higher precipitation. I'll get back to that in the discussion part.

L26: define "current" , please.

P9:

L2: How do you define "climate-related"? What else could it be on this time scale? Could it be that the first in-situ validation of increased precipitation in coastal Antarctica is due to the fact that the drilling location is influenced rather locally? Did you compare

it with temperature proxies? I am not saying it is wrong or right what you state, but you should discuss this.

L8: strange usage of "refer to". Maybe better "represents" or similar.

L13ff. Decreasing trend: I assume you mean "negative trend". Decreasing would mean getting stronger negative with time.

Please, make sure that it is clear, which time period is considered in your respective comparisons.

L10: Stenni et al: 1992-1996: too short a period to consider any trend calculation

P10:

L5. What is the reason for the choice of the threshold? Many coastal stations have SMBs around 0.3. This seems a bit arbitrary.

L9: this is covered by only two high accumulation sites..

L14: dating accuracy

P11:

L4ff: the positive trend in SMB... the result of various forcings

L7: the air does not "hold vapor", a higher temperature means a higher saturation vapor pressure.

L7ff: Paragraph 4.3 is very important, but, unfortunately, it contains quite a few misconceptions (in spite of the fact that one of the co-authors is a meteorologist and expert for polar/Antarctic meteorology) and thus should be re-written:

First of all, there is quite a bit of confusion of coastal and continental conditions. Several papers are quoted, of which some deal with the interior and others with the coastal areas of Antarctica, which, however, have very different precipitation regimes. Amplified Rossby waves are particularly important for precipitation in the interior of the continent,

NOT for the coast. The coastal areas are always under the influence of synoptic activity in the circumpolar trough. The individual events quoted in line 18 can bring up to 50% of the total accumulation in the interior, not at the coast. And also this means the sum of all events, not one single event. 2009 and 2011 were years with such events in the interior, which of course, also bring high precipitation to some coastal areas, but are not necessarily associated with lower surface pressure, on the contrary, the pressure in the coastal areas of Antarctica is usually lower in years like 2010, where a zonal flow was predominant and the interior of the continent got less precipitation than on average.

L25ff: SAM: what was the temporal resolution of your comparison of SAM, SOI and your data? Annual means, monthly values? You should not expect any signal in the annual mean since the SAM index has high intra-annual variations.

P12:

L 4ff: you discuss topographic influences here, but never question that the result for the ice rise might be more locally influenced than climate-related (whatever that means). The topography of an ice rise influences the synoptically caused winds much more than the surrounding ice shelf or the plateau since the ice rise represents a disturbance in the main flow. This is especially surprising since the authors include the Lenaerts et al. J. Glac.2014 paper, which investigates the climate and mass balance on ice rises, in the reference list, but never discuss it in the text.

L19: what do you mean by "these two highly variable accumulation events"?

L20: what is the physical explanation for DML being most susceptible to an increase in snowfall in a warmer climate? So far, a positive trend in Antarctic sea ice has been observed, which according to your findings, should decrease precipitation. (not sure about the regional trends, though, I am no sea ice expert.)

L24ff: see general comment. What is the temporal resolution of the investigation of the

relationship between SAM, SOI and SMB?

L26ff: Low pressure: see above. Usually the pressure in the circumpolar trough is lower (on average) in years with more zonal flow and less meridional heat and moisture exchange (positive SAM index) than in years with amplified Rossby waves.

P13:

L4: positive trend

L12ff: I do agree that the ice rise is a suitable potential drilling site for a longer core. However, you should investigate the representativeness of your results a bit closer and keep this in mind when interpreting a deeper core

References:

The reference list contains quite a few publications that are not quoted in the text. Please, check.

P16: L15: new paragraph: Hofstede...

P20: l25; new paragraph: Schlosser...

P26: the caption of Figure 26 should be rephrased: "Diff. in mean annual SMB between ~1960-present and ~1816 –present (a,b)" (c,d accordingly)

P31: Figure 6: a) b) lables missing

The legend is a bit confusing, since the dotted lines claim to be a mean SMB, only the caption explains that it is mean plus/minus STD. Maybe a single line with some shading for the range of the STD would be show this more clearly. For 1992 to 2012, one would expect that the averages are not very different, given the closeness of the green and the black line. ??

---

## Author Response (AR1)

The development of the timescale is the crux of this study and it is clear that considerable effort has gone into the timescale. The paper makes it clear, without explicitly stating so, that the timescales was quite difficult to develop. I have great sympathy for anyone who develops ice-core timescales. However, the timescale as presented is not convincing for two reasons. First, the isotope sampling is too low to resolve annual layers for much of the core. At a sampling interval of 10cm (above 80m), this yields only 5 or 6 samples per year for much of the timescale given the accumulation rates. You need about twice that to resolve clear annual layers, especially on a proxy such as oxygen isotopes that have relatively noisy seasonal cycles. Statements like "no ambiguity in layer counting is detectable above 62.38 m (i.e. 1933 AD)" are in direct contradiction with the need to perform major ion analysis "for sections of unclear isotopic seasonality" and I can see ambiguity in Figure 2 (near 20 and 29 m depths).

It seems odd to me that for a relatively short core, the whole thing wasn't sampled at much finer resolution (water isotope analyses are cheap and don't need much ice) and that aerosol analysis wasn't performed on the full core.

Second, the volcanic matches are not convincing. In Figure 4 it appears that any small peak that rises past the 2sigma level is considered a volcano if it happens to be of the correct age. This may be because the data are normalized before identifying volcanic peaks. Regardless of the normalization issue, using ECM (or sulfur) at coastal sites to identify volcanic events is very difficult because the volcanic signal gets overwhelmed by marine inputs.

The timescale is the crux of this paper. This means considerably more effort needs to be made to describe it convincingly, for both the annual layers and volcanic matches.

Items that this paper needs:

1) A clear description of what measurements were made at what depths (i.e. where were aerosols measured and show the ambiguities and how they were interpreted)

2) An analysis of the impact of the low sampling resolution on the ability to resolve annual layers (often, low resolution leads to picking false peaks)

3) A realistic assessment of annual-layer interpretation uncertainty

4) A critical assessment of volcanic matches. I.e. why is Cerro Azul 1932 not one of the bigger, yet unmatched, peaks about a meter above or below. (this same question applies to pretty much every match, except for possibly Tambora).

5) A description of why ECM loses the annual signal yet preserves the volcanic signal

6) Why Krakatau isn't observable in the ECM record and what the distinctive characteristics in the aerosol record are that allow it to be identified. Also a description of why the technique to identify Kratatau wasn't applied for the full core.

It sounds like the only truly identifiable volcanic event was Tambora. The authors need to make use of Tambora, and pattern with the unknown 1809 eruption, to make a strong case that this is indeed properly matched (Figure 4 does not do this). Plot it against high resolution ECM/Sulfur/Sulfate records of this event. If the authors can demonstrate that this is a clearly identifiable match, then it would strongly support their annual layer interpretation.

**Author's response:**

*We respond to this comment in three steps. First, we assess the referee's comments concerning annual layer counting. Second, we discuss the volcanic matches. Third, we address each specific item that Referee 1 suggested we consider or implement.*

*First, we point out that the isotope sampling resolution reported in the original manuscript was not 10 cm everywhere above 80 m. To explore this, we have nowe calculated the number of samples per year and report these in Figure R1. Other studies have worked within this range (e.g., Schlosser and Oerter, 2002). We agree that it should be stated more clearly which resolution was used at which depth, and have now added the full isotope profile with a visual indicator*

*of the resolution as two supplementary figures: Fig. S1 and Fig. S2. Ambiguities are now highlighted and discussed. At some depths, (e.g., between ~74 and 77 m) we increased the resolution to 5 cm, but – with an annual layer thickness of several tens of cm, in no case did the higher resolution data actually improve in the identification of annual layers. Therefore, we have not made more isotopic measurements. However, we did measure ECM at high resolution all along the core and this can be used to identify annual layers as well. It is a combination of both methods, supplemented by ionic measurements where available, that gives us confidence in our annual layer counting. For example, the ambiguities observed by Referee 1 "near 20 and 29 m depth" in $\delta^{18}O$ are resolved at 20 m by looking at the synchronous peaks in MSA, nssSO₄ and NO₃⁻, and at 29 m by looking at the synchronous peaks NO₃⁻, ECM and the trough in the Na⁺/SO₄⁼ ratio. However, we do agree with the reviewer that ambiguities remain elsewhere, and this is precisely why we adopted (and have retained) the approach of two age estimates: youngest and oldest. The sentence "no ambiguity in layer counting is detectable above 62.38 m (i.e. 1933 AD)" has been removed. It is now stated that this method has a ±16 year uncertainty at the base of the ice core. This new 'uncertainty' is the result of us considering the potential issues raised by Reviewer 1 and working through the entire record again in a more "conservative" way (described in Methods). Therefore, and despite the fact that the Tambora eruption still confirms the oldest estimate (see below), we added an analysis of the impact of this 16 years dating error on the trends reported in the paper in all figures and tables.*

[Figure]

*Figure R1. Distribution of the number of samples per year for the youngest and oldest estimates*

*Second, we agree with the referee that volcanic matching in a coastal ice core is, although not impossible (e.g. Kaczmarska et al., 2004), very difficult, even though we used ECM in combination with the annual layer counting. Therefore, we have chosen a simpler "conservative approach" along the lines suggested by the referee, i.e. only focusing on the Tambora eruption signature. In the revised manuscript we moved the high resolution ECM profile to a supplementary figure and added into the manuscript proper a figure centered on the depth range corresponding to where Tambora eruption should be visible, according to our two estimates (Fig.5). We also went back to the laboratory and made additional major ion measurements to document the Tambora eruption and show these on the same graph. We were very pleased to discover and report that there is only one peak that crosses the ECM 4 sigma threshold in the expected depth range and that it occurs at a depth corresponding precisely to our oldest age-depth estimate.*

*While these changes do not influence our conclusions, they do improve confidence in them and we thank Reviewer 1 for pointing us in this direction.*

*We now discuss each specific item suggested or requested by Referee 1:*

1) A clear description of what measurements were made at which depths (i.e. where were aerosols measured and show the ambiguities and how they were interpreted)

*This is achieved by the full isotope profile as Fig. S1 and Fig. S2, including a visual indicator of the resolution and an explicit indication of the annual layer boundaries identified according to the two estimates.*

2) An analysis of the impact of the low sampling resolution on the ability to resolve annual layers (often, low resolution leads to picking false peaks)

*This is now done with the youngest estimate, which only interprets the minimum number of annual layers.*

3) A realistic assessment of annual-layer interpretation uncertainty

*This is also addressed by the youngest and oldest estimates.*

4) A critical assessment of volcanic matches. I.e. why is Cerro Azul 1932 not one of the bigger, yet unmatched, peaks about a meter above or below. (this same question applies to pretty much every match, except for possibly Tambora).

*As explained above, in the revised manuscript we have focused solely on identifying the most distinctive peak, that of Tambora. However, for information, the oldest estimate in our revised manuscript would now be 3 years older at the same depth and Cerro Azul does indeed correspond to the peak at 61 m. We do not discuss this in the revised manuscript since it occurs in a section of the core where the mismatch between our older and younger estimates is still reasonably low (±2 years).*

5) A description of why ECM loses the annual signal yet preserves the volcanic signal

*ECM loses the annual signal and the volcanic signal only in some sections of the record, e.g. between 83 and 85 m. This could result from a variety of factors that we do not discuss because ECM seasonality is only used as a back-up where needed and not as a primary source of information.*

6) Why Krakatau isn't observable in the ECM record and what the distinctive characteristics in the aerosol record are that allow it to be identified. Also a description of why the technique to identify Krakatau wasn't applied for the full core.

It sounds like the only truly identifiable volcanic event was Tambora. The authors need to make use of Tambora, and pattern with the unknown 1809 eruption, to make a strong case that this is indeed properly matched (Figure 4 does not do this). Plot it against high resolution ECM/Sulfur/Sulfate records of this event. If the authors can demonstrate that this is a clearly identifiable match, then it would strongly support their annual layer interpretation.

*The characteristics of a volcanic peak are now shown only for Tambora, with $nssSO_4$ and $SO_4^=/Na^+$, that also show a peak, outside the seasonal variations and synchronous with the ECM record. We agree with the referee in believing that the ECM signal is potentially subject to too much influence by marine inputs to act as an unambiguous indicator for many of the other peaks. We thank the reviewer for these observations and the Methods and Discussion sections of the revised manuscript have been changed accordingly.*

*The Tambora peak and the associated ion record can now be seen in Fig. 5. No other peak above or below could be associated with this eruption. We associate it to a clear peak in $SO_4^=/Na^+$ which occurs between two seasonal peaks and corresponds to high $nssSO_4$ value (3.3 times higher than the mean). We believe this new conservative approach is scientifically robust and lends strength to our oldest estimate of the time scale involved.*

**Referee 1 general comment: 2- Description of the layer thinning correction**

The corrections for flow-induced layer thinning reveal a lack of understanding of how ice flow and are clearly underestimated. In particular, it is disappointing that the authors don't make use of the detailed ice-flow modeling that's been done on Derwael Ice Rise (Drews et al., 2015) to develop the vertical thinning function. It is clear from phase sensitive radar measurements (Kingslake et al., 2014) that the simple approximations for vertical thinning have trouble replicating the vertical strain pattern under ice divides.

The Nye assumption is so obviously not applicable to Derwael Ice Rise, which has a distinctive Raymond Bump, that it should not even be considered. The authors don't supply the kink height value of the D-J model. Using a kink height

of nearly 1, Kingslake et al. could still not match the pattern under Roosevelt Island. Since the authors say the kink height is below the zone of interest, I can infer that they didn't use anything greater than ∼0.7. This will lead to an underestimation of the amount of strain experienced by the ice in the core. Thus the older accumulation rates will be underestimated, and it will appear that there has been an increase in modern accumulation rates. The underestimation is likely exacerbated by the preferred thinning rate of 3cm per year for the ice rise (Drews et al., 2015). Getting the thinning right is critical to primary conclusion of this paper.

**Author's response**

*We thank the referee for this remark, which is certainly relevant and important. However, as we will show below, the effect of taking the Raymond effect into account does not alter the main conclusions of the manuscript.*

*First, we removed the Nye time scale approach from the revised manuscript, which is – as rightly pointed out by the referee – much too simplistic to be valid at the ice divide of an ice rise (we actually initially chose to show it to demonstrate the importance of using a more refined and adequate approach). Also as suggested by the reviewer, we took the vertical velocity profile from Drews et al. (JGR, 2015), which takes into account the Raymond effect on the Derwael Ice Rise through a full Stokes approach, as well as a slight amount of thinning (although the thinning is not the main factor to obtain the best fit) and ice anisotropy. This Drews profile indeed yields the best match with radar layers at depth. However, the Drews et al. (2015) strain rate profile used a mean accumulation rate that is somewhat lower that the long-term accumulation rate we obtain from the ice core. In order to determine the long-term accumulation rate we relied on an independent measure of horizontal surface strain measured on the Derwael Ice Rise. From a hexagonal strain network, we calculated horizontal strain rates ($\varepsilon_{xx} + \varepsilon_{yy}$) to be equal to 0.002 $a^{-1}$. Mass conservation then gives a vertical strain rate at the surface of -0.002 $a^{-1}$. The vertical velocity profile was then scaled to match the measured vertical strain rate at the surface. A best fit to the measured radar layers was obtained with a value of a mean accumulation rate of 0.55 m $a^{-1}$ ice equivalent (see Fig. R2 below and Fig. 2 in the revised manuscript).*

*As an alternative approach, we used the Dansgaard-Johnsen model to fit the characteristics at the ice divide, as exhibited by the Raymond effect. Assuming that the horizontal velocity is zero, the vertical velocity is maximum at the surface, where it equals the accumulation rate (with negative sign), and is zero at the bed. Assuming a vertical surface strain rate of -0.002 $a^{-1}$, we can determine the location of the kink point (between constant strain rate above and a strain rate linearly decreasing with depth below) that obeys these conditions (Cuffey and Paterson, 2010). This approach indicates that the kink point lies at 0.9H, where H is the ice thickness. As seen in Fig. 2b, this method yields a vertical strain pattern that is consistent with that of Drews et al. (2015), especially in the first 120 m corresponding to the length of the ice core.*

*Both strain rates (Drews/D-J) were then used to correct the ice equivalent layer thickness for strain thinning. Layer thicknesses were then converted in from ice equivalent to w.e. for easier comparison with other studies.*

*While these results still conform to the previous conclusions of the paper, they are more robust and we thank the reviewer again for raising this issue. Figure 6 of the revised manuscript has been adapted to include this new, more physically sound, approach. We would like to point out that this paper is one of the few that actually investigates the impact of deformation on annual layer thicknesses in such details. We also now include Reinhard Drews as one of the co-authors.*

[Figure]

*Figure R2. Vertical velocity (a) and vertical strain rate (b) profiles, according to the modified Dansgaard-Johnsen model (blue) and the full stokes model (black, Drews et al., 2015).*

**Refereee 1 general comment: 3- Climate implications**

5    The discussion of atmospheric and sea-ice patterns seems like an after thought. I'm not sure why the authors choose to analyze only anomalously high and low years. I also wonder why the authors don't compare the inferred accumulation rate history to the climate reanalyses. Other cores (e.g. Medley et al., 2013, GRL; Morris et al., 2015, Nature Geosciences) find good correlation of annual accumulation.

**Author's response:**

10    *In the revised manuscript, we have framed the discussion of the relation between core-derived SMB and climate parameters better. We now compare the ice-core-derived SMB with P-E estimates from ERA-Interim and RACMO2. The correlation is moderate for both (R²=0.36 and 0.5 for ERA-Interim and RACMO2, respectively), compared to other ice cores in West Antarctica, which indicates that local wind-induced snow redistribution and sublimation are significant contributors to local SMB at the ice core site (Lenaerts et al., 2014). Subtle variations in wind speed and*
15    *direction could lead to strong perturbations of the snow accumulation, especially at a wind-exposed site such as Derwael Ice Rise. However, it is unlikely that the wind has an impact on the temporal trend observed in IC12. Unfortunately, our methods do not allow explicit partitioning of the SMB explained by precipitation vs. wind processes. Therefore, we compared with CESM rather than with the reanalyses data because (1) it yields an SMB and climate time series that overlaps substantially with the ice core record (1850-2012), unlike the reanalyses that only*
20    *covers ~35 years, and (2) the present-day climate and SMB are realistic (Lenaerts et al., 2016). This is now clearly indicated in the text of the revised manuscript.*

*We now address the specific comments made by Referee 1.*

| Referee 1 specific comments | Author's response |
|---|---|
| P1, L27-29 – your data do not actually support this because your thinning correction is much too small. | *We revised the complete strain correction by using the Drews et al. (2015) strain rates and a modified D-J model (discussed in detail above). Both are further constrained by measured surface horizontal strain rates.* *This amendment has not altered our conclusions.* |
| P2, L10 – you should mention timescales. Frieler et al only address glacial-interglacial changes. The most directly comparable ice-core record to yours is from Law Dome, which does not show a consistent | *We added the precision "during glacial-interglacial changes" and took more care at mentioning timescales in the revised manuscript.* |

| | |
|---|---|
| relationship between accumulation and temperature in the Holocene. | |
| P3L1 – in the ice-core community, continuous is generally used to mean a melting system where discrete samples are not cut. Using continuous to mean that all of the core has been sampled discreetly is confusing. | *Amended* |
| P3,L13 – Be specific about what you mean by local ice flow. You should really mention that it's an ice rise with a well developed Raymond Bump that has likely been stable for thousands of years. | *This is now described in Paragraph 2.1* |
| P4,L25 – DC-ECM does not depend on the impurity content at the crystal boundaries. It depends on the acidity. | *Rectified* |
| P4,L28 – a ~30cm smoothing window seems really large to me. | *We also tested with a smaller smoothing window (101 and 201) and we chose 301 points in an attempt to reduce the noise from the marine input. This does not have an impact on the Tambora volcanic horizon we discuss.* |
| P5,L1-3 – Did you normalize the data before identifying the volcanic peaks? If so, you can no longer reliably identify volcanic events with a threshold because the increased conductance of volcanic events would impact the normalization. Is the 2sigma threshold then for the entire data set. I'm confused, but I think this is a major problem. | *We applied the method described in Karlof et al. (2000) and Kaczmarska et al. (2004): "The Savitsky-Golay filter eliminates peaks created due to random noise or short-term chemistry events but preserves peaks expected from volcanic events."*
 *We now use a $4\sigma$ threshold instead of the $2\sigma$.* |
| P5,L5-22 – This section should be entirely redone. Get a thinning function from the ice-flow modelers working on Derwael Ice Rise. | *Amended (discussed in detail above)* |
| P6,L17 – explain what changes in the ECM and why | *We changed the sentence by: "For ECM, there is also a regular seasonal signal, but it becomes very noisy below 80 m, although some seasonal cycles can still be seen for example between 115 and 118 m (Suppl. Fig. 1)"* |
| P6,L27 – explain how Krakatau was identified | *This sentence was removed. See response to general comment nr 1, specific items nr 6.* |
| P10,L11 – why are the uncertainties being presented after the results | *We modified the structure of the discussion and moved the paragraph about uncertainties to the end of the Results section.* |
| P10,L30 – there is a lot complexity in the position of the divide, the Raymond Bump, and the minimum accumulation (which is offset from the divide). There needs to be a much more detailed discussion of whether small (i.e. one ice thickness) | *This comment has been clipped (the last sentence is not finished) but we understand that the reviewer suggests we explain the small-scale variability of the SMB near ice divides in more detail. We have amended sections 2.1 and 3.3 accordingly.* |

**Referee 2 general comments:**

I have only basic knowledge in dating ice cores using flow models, so I cannot assess the critics of referee #1 considering this point. The authors do show both the uncorrected data and the correction with the different models, so the reader can assess what they have done. Also, their main conclusion (positive SMB trend in the last 100 years) would still be valid for any calculation of layer thinning that lies between the two methods they use.

However, I share Referee #1's doubts about the details of the dating, particularly the use of volcanic horizons, since the attribution of the ECM peaks in Figure 4 to the different eruptions is not convincing, except for Tambora. Also, the authors do not give details about the layer counting using stable isotopes, to which depth this was possible etc. Nobody expects a perfect dating of an ice core because this hardly ever exists.

However, I think the authors should discuss the error possibilities of the dating a bit more and give a more realistic quantitative estimate of the error. Probably, within the error bounds, their main result would hold. But, see above, I cannot assess the details of the used models. The authors state that their findings (increase in SMB in a coastal East Antarctic core) are the first ones that support model predictions. This does not make them discuss how representative their results are. They compare their results with other firn/ice cores, but do not compare the temporal variations of the SMB derived from the core with temporal variations of measured and/or modelled air temperature, sea ice, or surface pressure data). Instead they look at composites for very positive and very negative years, which is, in principal, not a bad thing to do, but I would expect stronger signals here in order to be convincing. The arguments using the output from the Community Earth System Model are a bit weak. The discussion of the atmospheric dynamics involved is not clear and mixes up conditions at the coast and in the interior of Antarctica. Also, different time scales are mixed together and often it is not clear, which time period is meant when certain trends are reported.

**Author's response to referee 2's general comments:**

*We decided to follow the advice of the referees and removed the detailed volcanic matching, except for Tambora (described in detail above). We also include an assessment of the impact of the 16 years dating uncertainty in all graphs and tables and in the main text to show that it does not change our conclusions.*

*As outlined in our response to Referee 1, there is a moderate temporal correlation between the SMB from the ice core and the SMB from climate reanalyses, which suggests that wind processes influence local SMB at Derwael Ice Rise. The relationships between precipitation and sea ice, SST and large-scale circulation are analyzed using output from the Community Earth System Model (CESM). CESM was selected for two reasons: (1) it yields an SMB and climate time series that overlaps to a great extent with the ice core record (1850-2012), unlike the reanalyses that only cover ~35 years, and (2) the present-day climate and SMB are realistic (Lenaerts et al., 2016). This is now more clearly indicated in the text.*

*We thank the reviewer for the suggestion on the significance of the signals that are found in low and high accumulation years. We have now compared the anomalies in those years with the temporal standard deviation, and adapted ex-Figure 7 (now Fig. 8) such that signals are only shown where they are larger than one standard deviation. Clearly, the signals exceed the standard deviation for the high anomaly years, but are not significant for the low accumulation years. Therefore, we decided to omit the bottom panel and only show the situation in the high accumulation years.*

| Referee 2 Specific comments | Author's response |
|---|---|
| Title: what does "recent" mean? and, to be correct, "snow accumulation" should be "surface mass balance". | *The title has been changed to: "Ice core evidence for a 20th century increase in surface mass balance in coastal Dronning Maud Land, East Antarctica."* |
| Abstract: It would be good to re-write the abstract after the main text has been revised. | *Agreed and done.* |

| | |
|---|---|
| P2: | |
| l5: increasing ice discharge | *Amended* |
| l8: What does the Polvani paper have to do with warming- related increase in precip? There are other papers that involve data and modelling and do not find either warming or increase in precipitation in the considered period. Please, make sure that it is clear about which time period you are talking. | *We deleted the Polvani reference and added a sentence acknowledging papers that do not find warming, except in West Antarctica. Papers that do not find an increase in SMB were already mentioned.* *We added precisions of the periods considered.* |
| l23: "both authors concluded that the trends were insignificant". This is not correct and not exact. Which trends? Altnau et al. found a statistically significant positive trend in SMB for the interior DML. | *We apologise for the confusion. The sentence has been changed to "Frezzotti et al. (2013) showed no significant SMB changes over most of Antarctica since the 1960s, except for an increase in coastal regions with high SMB and the highest part of the East Antarctic ice divide, and Altnau et al. (2015) found a statistically significant positive trend in SMB for the interior DML."* |
| P3: | |
| L10ff: grammar: in your sentence, "which" refers to the project. | *The sentence has been changed accordingly.* |
| L12: a local flow regime | *Amended* |
| How high is the accumulation rate? It would be good to give this information already here. | *We added this information and chose to use the previously published accumulation rate of 0.50 m w.e. (0.55 m i.e., Drews et al., 2015).* |
| P4: | |
| L3: do you mean 30mm x 30mm? | *Yes, amended.* |
| L13: the boundary between annual layers | *Amended* |
| L21: better: were carried out | *Amended* |
| P5: | |
| L5: snow burial: better: the compression of the snow under its own weight | *Amended* |
| It would be interesting to see the density profile here, maybe you could add this in a figure. I also miss some information about the depth until which seasonal variations in the isotope ratios can be resolved. | *We think that adding the density profile in a figure is not necessary, since it is published in Hubbard et al., 2013. However, if the referee or Editor believes this would improve the quality of the paper, we are ready to do it.* |
| P6: | |
| L3: how reliable are the CESM data for the 19th century, especially sea ice? | *That is a very good question. In fact, we have little to no observational estimates of 19th century sea-ice extent. The CESM simulated sea-ice extent in the observational period is very realistic compared to observations (Lenaerts et al., 2016) and does not show any trend in the Atlantic sector, which gives us confidence that the sea ice is treated realistically.* |
| L24: better: mainly derived from. . . | *Amended* |
| P7: | |
| 1ff: see above. The volcanic peaks in Figure 4 seem to be pretty ambiguous in most cases. | *The correspondence with volcanic peaks has been completely revised (addressed in detail above)* |
| P8: | |
| L15ff: This is a very short and simplified view. The sea ice argument is not convincing, especially the hatched area of anomalies is fairly small and should not have a large impact on precipitation amounts. A decrease in surface pressure of not much more than 1hPa is not very much, even in a composite, and in that case, lower surface pressure does not necessarily mean higher precipitation. I'll get back to that in the discussion part. | *We do not agree entirely with the statement that the anomalies are fairly small. We find a maximum anomaly of sea ice extent of more than 30 days, which is much larger than the inter-annual variability. We agree that the surface pressure anomaly is fairly small; we have revised the text according the reviewers' comments (see below).* |

| | |
|---|---|
| L26: define "current", please. | *"current" was replaced by "recent".* |
| P9: | |
| L2: How do you define "climate-related"? What else could it be on this time scale? Could it be that the first in-situ validation of increased precipitation in coastal Antarctica is due to the fact that the drilling location is influenced rather locally? Did you compare it with temperature proxies? I am not saying it is wrong or right what you state, but you should discuss this. | *We removed the term "climate-related".*
 *We now discuss the spatial significance of our results at greater length.* |
| L8: strange usage of "refer to". Maybe better "represents" or similar. | *Amended* |
| L13ff. Decreasing trend: I assume you mean "negative trend". Decreasing would mean getting stronger negative with time. | *Amended* |
| Please, make sure that it is clear, which time period is considered in your respective comparisons. | *We agree that it was not clear and replaced all references to "the recent period" by "the last 50 years" and the "most recent period" by "the last ~20 years".* |
| L10: Stenni et al: 1992-1996: too short a period to consider any trend calculation | *Reference to this has been deleted* |
| P10: | |
| L5. What is the reason for the choice of the threshold? Many coastal stations have SMBs around 0.3. This seems a bit arbitrary. | *This threshold was chosen in order to be consistent with Frezzotti et al. (2013) (no threshold allows isolation of only coastal stations)..* |
| L9: this is covered by only two high accumulation sites.. | *Amended* |
| L14: dating accuracy | *Amended* |
| P11: | |
| L4ff: the positive trend in SMB. . . the result of various forcings | *Amended* |
| L7: the air does not "hold vapor", a higher temperature means a higher saturation vapor pressure. | *Amended* |
| L7ff: Paragraph 4.3 is very important, but, unfortunately, it contains quite a few misconceptions (in spite of the fact that one of the co-authors is a meteorologist and expert for polar/Antarctic meteorology) and thus should be re-written:
 First of all, there is quite a bit of confusion of coastal and continental conditions. Several papers are quoted, of which some deal with the interior and others with the coastal areas of Antarctica, which, however, have very different precipitation regimes. Amplified Rossby waves are particularly important for precipitation in the interior of the continent, NOT for the coast. The coastal areas are always under the influence of synoptic activity in the circumpolar trough. The individual events quoted in line 18 can bring up to 50% of the total accumulation in the interior, not at the coast. And also this means the sum of all events, not one single event. 2009 and 2011 were years with such events in the interior, which of course, also bring high precipitation to some coastal areas, but are not necessarily associated with lower surface pressure, on the contrary, the pressure in the coastal areas of Antarctica is usually lower in years like 2010, where a zonal flow was predominant and the interior of the continent got less precipitation than on average. | *We agree with the reviewer that this part should be more concisely written, and that we should discriminate better between coastal and interior regions. We have revised the text accordingly.* |

| | |
|---|---|
| L25ff: SAM: what was the temporal resolution of your comparison of SAM, SOI and your data? Annual means, monthly values? You should not expect any signal in the annual mean since the SAM index has high intra-annual variations. | *This was indeed a comparison of annual mean, but we decided to delete this sentence, since it is not relevant.* |
| P12: | |
| L 4ff: you discuss topographic influences here, but never question that the result for the ice rise might be more locally influenced than climate-related (whatever that means). The topography of an ice rise influences the synoptically caused winds much more than the surrounding ice shelf or the plateau since the ice rise represents a disturbance in the main flow. This is especially surprising since the authors include the Lenaerts et al. J. Glac.2014 paper, which investigates the climate and mass balance on ice rises, in the reference list, but never discuss it in the text. | *We appreciate the reviewers comment, and we agree with it. In the revised manuscript we now include discussion of the local wind effects on the SMB.* |
| L19: what do you mean by "these two highly variable accumulation events"? | *Sentence amended* |
| L20: what is the physical explanation for DML being most susceptible to an increase in snowfall in a warmer climate? So far, a positive trend in Antarctic sea ice has been observed, which according to your findings, should decrease precipitation. (not sure about the regional trends, though, I am no sea ice expert.) | *Lenaerts et al. (2016) attributed future increase in DML snowfall partly to increasing temperature and partly to a simulated future decrease in sea ice extent. The observational record does not show any significant changes in sea-ice in the Southern Ocean region around 30-70 °E (e.g. Bintanja et al., 2013).*

*However, although global sea ice area does appear to be increasing slightly in the Southern Ocean, several studies show that it this general expansion hides strong regional differences. Indeed, Stammerjohn et al. (2009) showed that the Princess Ragnhild coast area and, more generally, the Southern Ocean to the East of it, show a recent slight reduction of the sea ice season duration. This is part of a circum-antarctic bipolar pattern similar to the SAM spatial distribution.* |
| L24ff: see general comment. What is the temporal resolution of the investigation of the relationship between SAM, SOI and SMB? | *This comment is not linked to P.12, L24.*
*Anyway, we removed the investigation of the correlation between SAM, SOI and our observed SMB data from the revised manuscript..* |
| L26ff: Low pressure: see above. Usually the pressure in the circumpolar trough is lower (on average) in years with more zonal flow and less meridional heat and moisture exchange (positive SAM index) than in years with amplified Rossby waves. | *That is correct, and we apologize for the misinterpretation. Since the anomalies in surface pressure are smaller than the standard deviation, we decided to omit these from the Figure and revised text.* |
| P13: | |
| L4: positive trend | *Amended* |
| L12ff: I do agree that the ice rise is a suitable potential drilling site for a longer core. However, you should investigate the representativeness of your results a bit closer and keep this in mind when interpreting a deeper core | *The discussion has been amended accordingly.* |
| References: The reference list contains quite a few publications that are not quoted in the text. Please, check. | *Thank you, we checked the reference list and removed the errors. There are still a few references that are not quoted in the text. This is because they are referred to in Table A1, and therefore, used in Figure 1.* |

| | These are: Anschutz et al., 2009; Ekaykin et al., 2004; Frezzotti et al., 2007; Igarashi et al., 2011 ; Jiang et al., 2012; Morgan et al., 1991 ; Mulvaney et al., 2002 ; Roberts et al., 2015; Ruth et al., 2004 ; Schlosser et al., 2014; Sommer et al., 2000; Stenni et al., 1999; Takahashi et al., 2009; van Ommen and Morgan, 2010; Xiao et al., 2004; Zhang et al., 2006. |
|---|---|
| P16: L15: new paragraph: Hofstede. . . | Amended |
| P20: l25; new paragraph: Schlosser. . . | Amended |
| P26: the caption of Figure 26 should be rephrased: "Diff. in mean annual SMB between ~1960-present and ~1816 –present (a,b)" (c,d accordingly) | Amended |
| P31: Figure 6: a) b) labels missing | Amended |
| The legend is a bit confusing, since the dotted lines claim to be a mean SMB, only the caption explains that it is mean plus/minus STD. Maybe a single line with some shading for the range of the STD would be show this more clearly. For 1992 to 2012, one would expect that the averages are not very different, given the closeness of the green and the black line? | The Figure has now changed completely (discussed above). Since most volcanic horizons are not used as reference markers anymore, Figure 7 now illustrates the rate of change between fixed periods of 20 and 50 years. |

**References in response**

Bintanja, R., van Oldenborgh, G. J., Drijfhout, S. S., Wouters, B., & Katsman, C. A.: Important role for ocean warming and increased ice-shelf melt in Antarctic sea-ice expansion. Nature Geosci., 6(5), 376–379. doi:10.1038/ngeo1767, 2013.

Lenaerts, J. T. M., Brown, J., Van Den Broeke, M. R., Matsuoka, K., Drews, R., Callens, D., ... and  Van Lipzig, N. P. M.: High variability of climate and surface mass balance induced by Antarctic ice rises. J. Glaciol., 60(224), 1101– 1110. doi:10.3189/2014JoG14J040, 2014.

Stammerjohn, S.E., Martinson, D.G. Smith, R.C.,Yuan, X., and Rind, D.: Trends in Antarctic annual sea ice retreat and advance and their relation to El Niño–southern oscillation and southern annular mode variability, J. Geophys. Res., 113, p. C03S90 http://dx.doi.org/10.1029/2007JC004269, 2008.

[revised manuscript text omitted]
 aAAccumulation rates for three different periods (chosen for easier comparison to with previous studies) starting from theare calculated on the basis of deformation corrections (Nye and D-J) and averaged over various periods framedaveraged over three different periods between by Tambora eruptionand Cerro Azul volcanic horizons and the surface (1816–2011), the last 50 years compared to the previous full period of timeprevious (i.e., 1962–2011 cfvs. 18166–1961), and the last 20 years compared to the previous full period of time previous (i.e. 1992–2011 cf.vs 1816–1992), for the youngest and oldest estimates and average between both (Table 1(e.g. Kaczmarska et al., 2004, Sigl et al. 2012; bold years in Table 1) are shown in Figure 6b and summarized in Table 2). The long term annual accumulation, starting from the oldest volcanic layer identified:From 1768 1816 to 20112, the average accumulation rate is between 0.39 and 0.460.49 ± 0.02 m w.e. a$^{-1}$ depending on the correction applied. For the last 50 years (Table 2). The recent (19551962–20111)2), the accumulation rate is between 0.60 61and 0.63 ± 0.01 m w.e. a$^{-1}$ with, as expected, less impact from the different deformation corrections, representing a. The sharpest increase occurs between the periods 1902-1955 and 1955-

[revised manuscript text omitted]

20   migration occurred as the DIR has been stable for at least the last thousands of years (Drews et al., 2015; Callens et al., 2016). Temporal variability of accumulation rates at certain locations can also be due to the presence of surface undulations up-glacier (e.g. Kaspari et al, 2004), but this effect is minimised at ice divides.

Average accumulation rates on longer time periods are therefore more robust than reconstructed annual accumulation rates because they are less affected by uncertainties. These average estimates are also useful to reduce

25   the influence of inter-annual variability.

**3.43 Relation to atmospheric and sea ice patternsComparison with climate models**

Figure 8Figure 7 compares the trend in our IC12 SMB record with outputs from two atmospheric models: ERA-

30   Interim reanalysis (Dee et al., 2009) and the CESM model. ERA-Interim shows no trend in the Interestingly, in the

relatively short overlapping period (1979—2012) it covers, , the ice core derived SMB correlates  moderately to ERA-Interim  and RACMO2 (Lenaerts et al., 2014), yielding R0.4, respectively ).  For a longer overlapping period, we used the output of the CESM model, although it is a freely evolving model that does not allow a direct comparison with measured data. The average SMB at Derwael in CESM (closest grid point) is too low (0.295 ± 0.061 m a$^{-1}$) because the orographic precipitation effect is not well simulated .   CESM does reproduce (much of) the observed trend. Subtle small-scale variations in wind speed and direction, typically not resolved by reanalyses or regional climate models, might disrupt the inter-annual variability of SMB, although we assume that it does not influence the positive SMB trend found in the ice core record. Unfortunately, our method does not allow for an explicit partitioning of the SMB explained by precipitation vs. wind processes. Instead, we focus on the drivers of precipitation at the ice core site using the output of CESM (Fig. 8), and we discuss it in Sect. 4.1.~~9). In anomalously high-accumulation years (top panel), , the sea ice coverage is significantly lower than average (20-40 fewer days with sea-ice cover) in the Southern Ocean northeast of the ice core location, which is the prevalent source region of the atmospheric flow (Lenaerts et al., 2013). This is associated with significantly higher near-surface temperatures (1-3 K). In low-accumulation years (not shown), we see a reverse, but less pronounced (not significant) signal, with higher sea ice fraction (10-20 days), and slightly lower temperatures and the oceanic source region of precipitation.~~

Figure 7 shows a summary of the output from the CESM as described in Section 2.5. In anomalously high-accumulation years (top panel), the sea ice coverage is very low (20-40 fewer days with sea-ice cover) in the

Southern Ocean northeast of the ice core location, which is the prevalent source region of the atmospheric flow (Lenaerts et al., 2013). This is associated with higher near-surface temperatures (1-3 K), and a strengthening of the low climatological low-pressure system (>1 hPa lower surface pressure), located offshore the ice core location (Lenaerts et al., 2013). In low-accumulation years (bottom panel), we see a reverse, albeit less strong, signal, with
5 higher sea ice fraction, lower temperatures and higher core pressure of the low pressure system.

**4 Discussion**

**4.1 Regional-scaleSmall-scale variability**

Figure 9. Large-scale atmospheric, ocean and sea-ice anomalies in high-accumulation (10% highest) years in the
10 CESM historical time series (1850-2005). The colours show the annual mean near-surface temperature anomaly (in °C), and the hatched areas show the anomaly in sea-ice coverage ( >20 days less sea ice cover than the mean). The green area shows the location of the ice core.

Small scale spatial variability in cyclonic activity and atmospheric rivers could both explain why our results are
15 different from others in the same region, and why they correlate only moderately to the climate reanalyses (ERA-Interim and RACMO2). Orography can greatly affect spatial variability in SMBsnow accumulation (Lenaerts et al., 20143). Local wind phenomena are important factors of interannual variability. Indeed, the lowpoorer correlation with ERA-Interim and RACMO2 in our study, as compared to ice cores collected on West Antarctica (Medley et al., 2013; MorrisThomas et al., 2015) is presumably explained by the strong influencempact of local
20 wind-induced snow redistribution and sublimation on the SMB on the wind-exposed ridge of the Derwael ice riseDIR (Lenaerts et al., 2014).
However, Callens et al. (2016) showed that thise spatial pattern has been constant for the last thousands of years. Therefore, ourthe observed trend of increasing annual accumulation is highly unlikely to be explained by a different orographic precipitation pattern caused by a change in local wind direction or strength, which would cause a
25 different orographic precipitation pattern. This argument, along with the existing correlations with ERA-Interim and RACMO2, suggests that thisese trends isare not only representativelimited of the climate on to the DIR the Roi Baudouin ice shelf but that they areit is alsorepresentative of at least the Roi Baudouin areaice shelf, surrounding the DIR typical of a wider area.

[revised manuscript text omitted]
 combination of the wind spatial variability and the local nature of the atmospheric phenomenon potentially involved can explain the spatially contrasting trends observed.

A more recent study using a fully coupled climate model (Lenaerts et al., 2016) suggests that DML is the region most susceptible to an increase in snowfall in a present and future warmer climate. The snowfall increase in the coastal regions is particularly attributed to loss of sea ice cover in the Southern Atlantic Ocean, which in turn enhances atmospheric moisture uptake by evaporation. This is further illustrated in Fig. 8, which suggests that extremely high accumulation years are associated with low sea ice cover. The longer exposure of open water leads to higher near-surface temperatures and enhances evaporation and moisture availability for ice sheet precipitation (Lenaerts et al., 2016).

Small scale spatial variability in cyclonic activity and atmospheric rivers could explain why our results are different from others in the same region. Orography can greatly affect spatial variability in snow accumulation (Lenaerts et al., 2013). Highest snowfall and highest trends in predicted snowfall are expected in the escarpment zone, due to orographic uplift (Genthon et al., 2009). The main factor generating spatial variability, however, is commonly the wind; wind ablation represents one of the largest sources of uncertainty in modelling SMB. For example, in the escarpment area of DML, low and medium precipitation amounts can be entirely removed by the wind, while high precipitation events lead to net accumulation (Gorodetskaya et al., 2015). An enhanced wind speed coupled with an increase in accumulation could only increase SMB where the wind speed is low, while decreasing SMB in the windier areas (90% of the Antarctic surface (Frezzotti et al., 2004)). Frezzotti et al. (2013) suggested that snow accumulation has increased at low altitude sites and on the highest ridges due to more frequent anticyclone blocking events, but has decreased at intermediate altitudes due to stronger wind ablation in the escarpment areas. In DML however, Altnau et al. (2015) reported an accumulation increase on the plateau (coupled to an increase in $\delta^{18}O$) and a decrease on coastal sites, which they associated with a change in circulation patterns. Around Dome A, Ding et al. (2011) also reported an increase in accumulation rate in the inland area and a recent decrease towards the coast. Their explanation is that air masses may transfer moisture inland more easily due to climate warming.

A more recent study using a fully coupled climate model (Lenaerts et al., in press) suggests that DML is the region most susceptible to an increase in snowfall in a present and future warmer climate. The snowfall increase in the coastal regions is particularly attributed to loss of sea ice cover in the Southern Atlantic Ocean, which in turn enhances atmospheric moisture uptake by evaporation. This is further illustrated in Figure 7, which shows that extremely high accumulation years are associated with low sea ice cover. The longer exposure of open water leads to higher near-surface temperatures and enhances evaporation and moisture availability for ice sheet precipitation (Lenaerts et al., in press). Additionally, the low pressure system, located offshore the ice core location (Lenaerts et al., 2013) is strengthened and invigorates meridional heat and moisture transport towards the ice sheet. The opposite is true for extremely low accumulation years.

**5 Conclusions**

A 120 m ice core was drilled on the divide of the DIRerwael ice rise, and dated back to 1759 ± 16 A.D. 1745 ±2 A.D. using $\delta^{18}O$, $\delta D$, major ionsmajor ion where necessary, and volcanic horizons identified from ECM data. Three volcanic indicators allowed the identification of Tambora 182015 eruption, which constrained the dating of the bottom of the ice core to 1743 ± 2 A.Dto the oldest estimate.. THowever, 
[revised manuscript text omitted]

**Figurese caption**

~1960-~present -vs ~1816-~present

[Figure]

~1990-~present -vs ~1816-~present

[Figure]

Change in accumulation (%)

[Figure]

-60  -40  -20   0   20  40  60

Figure S1. Full vertical profile of water stable isotopes with a grey and black band on the left indicating sections of 10 cm and 5 cm resolution, respectively (a); major ion (b–f), normalized ECM conductivity expressed as multiple of standard deviation (σ) (light grey: 1 mm resolution, dark grey: 0.05 m running mean). The 4σ threshold is shown as a dotted vertical line, and identified volcanic peaks as dashed grey horizontal lines (g); annual layer

[revised manuscript text omitted]

**Supplementary materials**

[Figure]

Fig. S1. Full vertical profile of water stable isotopes with a grey and black band on the left indicating sections of 10 cm and 5 cm resolution, respectively (a); major ion (b–f), normalized ECM conductivity expressed as multiple of standard deviation (σ) (light grey: 1 mm resolution, dark grey: 0.05 m running mean). The 4σ threshold is shown as a dotted vertical line, and identified volcanic peaks as dashed grey horizontal lines (g); annual layer boundaries in the youngest (Green) and the oldest (Blue) estimates. Each colour transition indicates a boundary (h).

Fig. S2. Full vertical profile, as in Fig. S1 but split in 17 sections for more visibility.

[Figure]

[Figure]

[Figure]

[Figure]

[Figure]

[Figure]

[Figure]

[Figure]

[Figure]

[Figure]

[Figure]

[Figure]

[Figure]

[Figure]

[Figure]

[Figure]

[Figure]

---

## Referee Report (RR1)

Philippe et al. TCD

The authors put a lot of work into the revised version and the manuscript has been greatly improved. However, some points remain, which I will address in the following.

First a remark: the font size of the manuscript pdf file is a bit of an imposition, it was completely unnecessary to keep the formatting info on the right side. Especially with all the marked changes it was very hard to read. I suggest to avoid that in the future.

Comments to reply to specific comments (line numbers refer to the original manuscript:

P2, l23: you still do not mention that Altnau et al. found a negative trend for SMB at the coast at this point. Why?

P6, L3: your reply should be discussed in the text.

P10, L5: I still think that an SMB of 0.3 is difficult to use as a threshold since many sites at the coast have SMBs just around 0.3. so slightly above or below 0.3 would not mean a systematic difference here. (I am not sure why Massimo Frezzotti chose it in the first place.)

P12, L20: Please, discuss this in the text, too. The high SMB in 2009 and 2011 found by Lenaerts was mainly due to the atmosheric circulation patterns durings those years. Those patterns have a much stronger influence on SMB than a couple of days longer or shorter sea ice coverage. (and keep in mind that sea ice extent refers to 15% sea ice concentration, so plenty of open water available for evaporation.)

From now on, the line numbers refer to the revised version:

P2

Ll9: what does it mean that this is the only record that supports model results?? Are all the other cores not representative and the DIR core is the only representative one? This is not self-evident.

L20: this is not consistent evidence: e.g. Fudge et al. found that SMB and temperature are not always positively correlated.

Fudge, T. J., B. R. Markle, K. M. Cuffey, C. Buizert, K. C. Taylor, E. J. Steig, E. D.
Waddington, H. Conway, and M. Koutnik (2016), Variable relationship between
accumulation and temperature in West Antarctica for the past 31,000 years,
Geophys. Res. Lett., 43, 3795–3803, doi:10.1002/2016GL068356.

P3

L 18: see above, negative trend in SMB in coastal cores in Altnau et al.

L20: I would not call this "very few", there are quite a few investigations of DML cores from German, Scandinavian and Indian expeditions.

L21: higher than the interior (a comparative needs something to compare to)

P4

L15: ice rises are too small to "block" atmospheric circulation. (this would mean that the air flows AROUND the ice rise rather than over it. Blocking is a clearly defined term in meteorology.

L18-22: good!

P9

L3: this is not correct, there is ERA20C (ECMWF) meanwhile, which covers the entire 20[th] century.

P12

L7: delete "is"

P17

L4. Better: source region of atmospheric moisture for DIR.

Fig. 8 should be described as part of the results section.

L2-8: sea ice is only one factor. The same factor that causes high accumulation might influence the sea ice extent without changes in sea ice being the reason for the accumulation deviations. I still do not find this paragraph very convincing. E.g. high sea ice extent related to lower air temperatures might be caused by a generally more zonal atmospheric circulation pattern, which at the same time could be the reason for low accumulation due to lack of meridional moisture transport (as Lenaerts et al. showed for 2009 and 2011. These things should be discussed in the text.

L18: see above. Does it not make you think that no other coastal core in DML shows an increase in SMB?

L27: see above, please explain the physical reason for the choice of the threshold.

L31: if we compared

P20

L1-7: see above, maybe quote Fudge et al. here, too.

L8ff: this is still not clear. Atmospheric rivers don't occur for the whole year, just for certain events. Precipitation at the coast is usually event-type, but the events occur during the whole year, whereas in the interior those events happen not very often, but are related to amplified Rossby waves.

L25: 2009 and 2011

P21

L8: wind is certainly a very important factor, but e.g. the interannual variability in the years 2009-2011 was definitely not mainly caused by the wind. Be careful with general statements like this. Of course, in years with fairly "average" flow patterns, the wind is the main factor, that is correct.

L21: why should there be a decrease at the coast then?

L24ff: see comment on P1

Parts of the discussion actually belong into the results section.

p23

l16ff: see above

---

## Referee Report (RR2)

Review of Revised Phillippe et al., 2016 manuscript:

The revised manuscript by Phillippe et al. has notable improvements, particularly in regards to the ice-flow modeling and reconstruction of the accumulation rate history. The collaboration with other researchers doing ice-dynamics work has led to a robust and convincing correction of strain thinning.

Unfortunately, the development of the timescale remains flawed. The attempt to identify Tambora is unconvincing and undermines the entire development of the timescale. A critique of the timescale development is below.

Overall, the authors need to admit that they:
- cannot reliably identify any volcanic events and thus have no age control beyond annual layer counting
- did not sample the core at high enough resolution and may thus be overcounting, predominantly in the deep core
- and thus the interpretation of an increase in recent accumulation is tentative and more work is needed to answer this interesting question definitively.

This work can become published, but only after giving up on identifying volcanic peaks and giving an honest and detailed assessment of the annual layer interpretation and the likelihood of overcounting in the deeper part of the core. The type of questions that needs to be answer is: What would the record look like if you subtracted 1 year in every 20 below 40 m depth? Or 1 year in every 10? And then discuss whether your annual layer count is reliable at that level. One approach could be to use automated techniques (such as Mai Winstrup's straticounter: https://github.com/maiwinstrup/StratiCounter) on the different data sets to get a sense of the uncertainty. My guess is that the real uncertainty is closer to 6% than 0.6%.

The Timescale:
The paper hinges upon accurate interpretation of the depth-age relationship. The main conclusion is that the accumulation rate has increased in recent decades compared to the previous couple of centuries. This interpretation relies both the timescale and the corrections for density and ice-flow induced thinning. The authors have greatly improved the corrections, which can now be both understood and trusted. However, the timescale remains both poorly described and untrustworthy.

The magnitude of the inferred accumulation rate depends directly upon the thickness of the identified annual layers. Thus, supporting the interpretation of increased recent accumulation requires showing that the annual cycles are properly identified. The authors attempt to do this in two ways: 1) the presentation of data with distinct seasonal cycles to convince the reader that the seasonal cycles at all depths of the core are unambiguous (or at least nearly unambiguous) and 2) identify horizons (such as volcanic events) that can be tied to other cores (or other paleoclimate records) that confirm the annual cycle interpretation.

-Tambora
I recommended focusing on Tambora in my previous review but am deeply frustrated by the revised manuscript. It is worth discussing here why Tambora is so distinctive in Antarctic ice core records. Tambora is indeed the largest event in the past few hundred years. But part of the reason it is so distinctive is that is in preceded by the second largest event of the past few hundred years, yielding

distinctive double peak. I've attached a figure of Tambora and the unknown events as shown by Sigl et al. (2013) from the WAIS Divide ice core. It is worth noting that both events have durations of 3 years.

[Figure]

The purported Tambora in IC12 lacks all resemblance to this event found elsewhere.

1) the authors make no attempt to identify the preceding event (commonly know as the unknown 1809 event).
2) The sulfate peak associated with Tambora is not even the largest in the figure. The authors do not explain why the ECM is so anomalously high while the source of acidity, H2SO4, does not result in high SO4 levels.
3) The figure starts at 101 m, conveniently hiding the fact there is no data from 100 to 101 m – something that I do not believe is discussed in the text.
4) The lack of chemistry measurements of the full core means there is no ability to reliably identify and compare SO4 peaks along the core
5) The ECM data is heavily filtered, indicating it has major quality-control issues and reducing any confidence that it can reliably detect volcanic events. Further, the filtering methods are unclear with the techniques used reference (Karlof et al. 2000) being more complicated than what is described in this text (just the Savitsky-Golay filter and normalization) – I'm still not sure what was done to the ECM data presented here. Further again, the ECM data is corrected for density based on borehole optical televiewing that is likely not that well depth-referenced (the methods are not described except for an unprovided *in review* manuscript) and regardless, is not appropriate for ~10 cm scale variations anyway, as explicitly stated in the given reference, Hubbard et al., 2013. This technique likely just introduced a bunch of noise at annual-to-volcanic frequencies into an already noisy record.
6) The authors claim that the ECM peak occurs in wintertime, despite being in an Na trough with a clear So4/Na peak (and hence SO4). Though the water isotopes are indeed unusual, there is a shoulder which may indicate a lack resolution to identify peaks. This seems more likely to be a thin year than volcano, let alone Tambora. Further, the logic of a wintertime peak is faulty. Tambora is a multi-year event such that the peak should be highest in summertime when there is both volcanic deposition and ocean-derived deposition. I guess the authors could argue that the coastal characteristics of deposition could truncate the duration of Tambora - but they would need to do that and be convincing with climate model output and observation al data.

My biggest issue with the "identification" of Tambora is that it is so far from convincing I simply have no trust in the authors development of any part of the timescale. This is particularly important because the

timescale was clearly developed iteratively; the chemistry measurements were only made in areas of uncertain annual layering indicating that the annual layers (of d18O and ECM) were interpreted prior to the chemistry measurements clarifying the annual layer iterpretation. It is likely that the volcanic matching was also done before, and thus the annual layer interpretation may have (consciously or subconsciously) been interpreted to get the right age at Tambora. While this sort of issue is not uncommon producing annual timescales, the lack of awareness of this issue in the manuscript is troubling especially given the propensity to pick too many years in undersampled data.

-Annual layer interpretation
Evaluating annual interpretation in publications has to be based largely on trust since it is difficult to present that data and interpretation in a manageable way. The authors do a good job of presenting the data with the addition the supplementary figures. However, the authors fail to convince me of their uncertainty, and hence the underlying timescale. This is in part because of the unconvincingly attempt to identify Tambora, but also because the measurements are just not of sufficient quality and continuity to get 0.6% accuracy. Some of the issues:
-   The stable isotopes, the primary parameter used to identify annual layers, are of insufficient resolution much of the time. The histogram in Figure R1 shows that the mode of the number of data points per annual layer is 4, with a significant number of layers identified with 3, 2, and even 1 data point. 4 data points is not enough for annual layer interpretation, less alone fewer. The authors may say that the other data sets define these layers, so I address that below:
-   The ECM data is heavily filtered with a 301-point window and unclear other techniques (see above). The authors do not describe how this impacts the annual interpretation, which is a major shortcoming. But what I see of the ECM data suggests to me that it is of questionable reliability for interpreting annual layers, with an uncertainty of 10% not less than 1%.
-   The chemistry data is sporadic with samples sizes that appear larger than for the stable isotopes (it's hard to see in Figure S2), such that the same criticisms of the stable isotope record apply to the chemistry records.
For instance, between 103 and 104 meters, there is a sequence of 3 thin years which is interpreted exactly the same in the oldest and youngest scenarios (1837-1835 or 1813-11). Yet the Na/So4 looks like there are only two years. In both cases, the sampling frequency is too low to be sure peaks and troughs are being resolved. This type of interpretation may be finding an extra year, biasing the annual layer thickness low, and underestimating the accumulation rate. While individual interpretations can always be nit-picked, the real concern here is that the authors do not even acknowledge the uncertainty.

---

## Editor Decision (ED1)

Page/line numbers are referred to the editorial tracking version.

P1L20: SMN -> SMB. Also, sensitivity of what?

P1L22: "120-m-long"

P1L23: DML is not used in the abstract.

P1L26: The bottom of the ice core is dated as 1759 +/- 16 AD, so that the ice core includes the climate proxies in the past 240 years, not only in the 20th century as it is said in the manuscript title and at the end of the abstract (20th and 21st centuries).

P1L32: What does "in at least the last 50 years" mean? I think that the authors want to say "Reconstructed SMB increases with time in the last 50 years by 30%" or "Reconstructed SMB increases with time, and this trend becomes even clearer in the past 50 years".

P2L11: see my comment above about "20th and 21st centuries."

P2L18: remove "coastal". Ice discharge always happens at the coast.

P2L21: balanced -> compensated?

P3L20-24: Please rewrite these new sentences. In particular, I have no idea what "with high SMB" means at the line 21.

P4L2: "120-m-long"

P4L8: remove ", including DML,"

P4L14-15: "preliminary ice core analysis" refers an earlier work of what is reported in this manuscript, I believe. So, it is not appropriate to cite that result in this way. Is it possible to show stake-measured SMB instead (as it is independent of the ice core work reported here)?

P4L26: Change to "Radar stratigraphy shows that locally maximum SMB happens about 4 km upwind of the ice core site"? And consider adding a figure showing the ice core site together with layer-depth SMB and surface elevations in contours (similar to Fig. 4 of Drews et al., 2015).

P7L5-10: Because Kjær et al. is still under review, please include sentences that describe the magnitude of this correction. I assume that this method removes long-term trends but not short-term variations so that determining annual cycles in the ECM record is not sensitively affected by this correction. Also, it is hard to match depths of the ice core and borehole (Hubbard et al., 2013) precisely (Reviewer pointed out this issue but the authors did not respond clearly).

P7L28: change to "and rheological anisotropy of the ice. The strain rates are insensitive to the surface thinning and the strain rates remain the same even if the surface elevation is kept uniform in the model" or such. Also, be more specific which Drews's model result is used here. I think you used "A(n=3), dH = 100, chi = 0.03 m/a, layer-depth SMB, (anisotropic rheology)" in Fig. 11 of Drews et al.

P7L29: change to "Separately, we used GPS data to derive the horizontal strain on the surface."

P7L30: change "0.002 a-1" to "2 x 10-3 a-1"

P8L1: What does "scaled" mean here? Do you mean "The vertical strain rates derived by Drews et al. (2015) is xx so we increased (or decreased?) Drews's vertical strain rate by xx uniformly"?? Even with this change, it is unclear what "best fit" means (if Drews's profile is simply shifted, not shape of the depth function is changed).

P8L7: Change "alternatively" to "The other method we used to derive the vertical thinning rates is …."

P8L12-13: Figure 2b shows that Drews et al. and DJ model show distinct e_zz over the ice-core depths. They are different by ~13%. Is it significant for your discussion, i.e. do you need to develop the historical SMB records each for Drews's strain rate and for DJ strain rate?

P8L15: do you mean "We used Drews's and DJ's strain rates to compensate dynamic thinning in the annual layer thickness in order to estimate past SMB."?

P9L11: Please add a sentence to describe how this model is used in this paper.

P9L26: What do you mean by "trend"? Is it east-west trend, temporal trend??

P10L7: change to "These properties change smoothly over a few very thin ice layers (white dots in Fig. 3) so we assume that they are not disturbed by surface melting".

P10L22: change to "the reference surface (November 2012 AD)"

P10L23: change to "correspondingly dated to 1775 AD and 1743, respectively, or 1758 +/- 16 AD." (the mean of 1775 and 1743 is 1759, not 1758, but I assume that this difference is related to the timing of the drilling in 2012).

P10L28: Do you want to say "Hereafter, we examine volcanic signals in ECM signals as possible age controls to more precisely develop the depth-age scale bounded by the oldest and youngest cases."

P11L9: Be careful to say "threshold". If I understand correctly, the authors want to say "the preliminary depth-age scale developed with layer counting shows that the largest ECM peak beyond 4 sigma presents at 1815 so we interpreted it as the Tambra eruption. The secondary peak associated with the Tambora peak is found as well (unknown source, 1809) but its ECM peak reaches only 2 sigma. This ECM peak is lower than those found in most ice cores [ref] but still in a range of previously reported values [ref]. We found 13 other ECM peaks beyond 2 sigma, which can be potentially matched with known volcanic events. Nevertheless, there are many other ECM peaks beyond 2 sigma as well. So, we conclude …."

P11L17: "absolute", not "relative"? I believe that the authors say that the absolute dating using volcanic eruptions remain uncertain.

P11L19: The response letter says that the authors prefer the oldest estimate. If it is the case, develop the argument here further and say something like "We believe that the oldest depth-age scale is more realistic than the youngest estimate because of matching with the Tambora eruption, though it is not really convincing. Therefore, we use …"

P12L6: Add thinning rate corrections.

P12L8: The authors said that they cannot conclude whether the young or old depth-age scales are better, but the age of the ice-core bottom shown here (1744) is probably tied to the oldest estimate (but if so, it would be 1743 not 1744). Also, Why is the youngest age in the core changed to 201 from November 2012?

P12L9: move the sentence "without correction for layer thinning" above so that you report the layer thickness first, and then derived SMB. Also, show the range of annual layer thickness (max, min, mean), instead of just reporting the mean value.

P12L12: I got confused. The paragraph immediately above reports the derived SMB, so I assume that the thinning corrections are already made. Please reorganize paragraphs in Section 3.2 to clearly demostrate the logical flow.

P12L14: Is it really Section 4.2 (Discussion)? If so, it's better to say something like "we discuss this point further in Section 4.2."

P12L28-30: Remove the sentence about dynamic thinning; it is obvious and rather confusing. You just say here that the layer has been thinned, not thickened.

P13: see comments to Table 1. Consider moving these paragraphs about averaged SMB values to a discussion session where you compare these values to previous studies.

P14L4: add "~240 years" after "the whole period"

P14L6: change to SMB.

P14L8: "bounded", instead of "determined"? The real SMB is expected to be somewhere between the oldest and youngest estimates.

P14L11: Here you explain the error bars in Figure 6, but the explanation is too brief to give a comprehensive idea what they are. If I understand correctly, the authors assume that the summer peak can be shifted up to 5 cm to both sides. In other words, if the annual layer is A cm thick, the thickest possible layer can be A + 10 cm (5 cm widen to both sides), and the thinnest possible layer can be A - 10 cm. Then you applied the thinning factor to estimate the uncertainty of the estimated SMB value. Do I understand correctly? However, if this is the case, the error bar is shorter when the SMB value is small, and it is longer otherwise. I cannot such feature in Figure 6.

P14L11-13: I cannot understand this argument. Consistent features between isotopes and ions support an hypothesis that both represent seasonal changes. However, because both were sampled by 5 cm or 10 cm, both depth profiles may overlook an annual cycle that appears less than 5 cm thick. Uncorrected layer thickness (orange curve in Fig. 6a) shows that it is unlikely to miss such thin annual layers, but similarity of isotope and ion profiles cannot be the evidence for this argument.

P14L22: Provide reference/status of this paper.

P15L6ff: "in the vicinity of the crest"; topographic feature (Crest), not ice-flow feature (divide), should be cited in terms of SMB's spatial pattern. I saw that "divide" is used at some other places as well; please correct them as well.

P15L7: Drews et al. did not exclude a possibility of recent crest migration, which is too young to deform the Raymond Arches found at depths greater than 50-100 m where Raymond Arches become more visible.

P15L21: here you say that the ice-core-derived results are compared with two climate models, but later you compare the results with ERA-Interim, RACMO2, and CESM.

P15 L25: replace R2 with correlation coefficient or such.

P15L30: please add more information to explain why a freely-evolving model output cannot be compared directly but still your discussion here can be valid.

P16L1: CESM output of the SMB mentioned here (0.295) is an average value for a certain period or the most recent SMB in 2011?

P16L11: How many days is this region covered with sea ice? This information is necessary to judge how 20-40 days fewer sea ice coverage is significant.

P17ff: Section 4.1 shows many numbers and it is very hard to keep tracking the main argument. Please carefully review this section and re-organize it so that the discussion can be presented more clearly.

P17L23: Drews's Figures 3b and 7 shows that anomalously low SMB is persistent at the current position to the age of ice at 60 m depth. This is I think support evidence of author's argument that the observed trend of SMB in the ice core presents the temporal changes, not migrating spatial patterns.

P20L6: change accumulation to SMB.

P20L9: indicate the name of these two coastal sites that show significant increase of SMB in the last 20 years compared to the last 200 years.

P20L13: change "less important" to "insignificant" or "less visible".

P21L28: remove "2009 and 2011" so it will be "than average SMB years (Table 2)."

P22L15: It is said that detrended dataset is not shown, but the authors presented 11-year running mean SMB (Figure 6). Is this running mean record good enough to identify anomalous events in 2-4 years (1991-95 and 1940-42)?

P24L11: "A 120-m-long", change "divide" to "summit" or "ridge (or crest)".

P24L14: "Therefore we counted annual layers to develop oldest and youngest estimates of the ice. The annual layer thickness, density, and thinning functions are applied to derive time series of SMB from annual layer thicknesses."

P24L20: do you mean that "wind re-distribution is significant near the ice-core site but it is likely that this effect is persistent over time so that ice-core records represent SMB time series rather than migrating spatial patterns of SMB"?

P24L25-27: I cannot agree. Probably you want to say "Neither currently available climate models and re-analysis data cannot resolve ice-rise topography so their predictions are hard to match with the ice-core-derived SMB. Nevertheless, their temporal trends can be compared, and …."

P25L10: I believe that the authors can be more confident about their results. Clear seasonal cycles (not only thin ice layers!) found in this ice core clearly demonstrated the potential of a deep core from this site as excellent paleoclimate proxies.

P25L21: Change "uncorrected SMB" to "annual layer thickness in ice"? "uncorrected SMB" sounds quite confusing.

Table 1

- I am not really sure how these average values for different periods are important. You said that it is for comparison with other studies and if so please consider adding an extra column showing the SMB values obtained from previous studies and compared with the average values that you are reporting.

Figure 1

- Change "accumulation" in the figure to "SMB".

Figure 3

- Add something like "d18O profiles are shown multiple times to better illustrate correlations between d18O and major ion profiles".

Figure 5

- It's very hard to see thin gray bands. Use more distinct color (red, blue, or such, not gray).

Figure 6

- Please align all four panels vertically so each panel can be a bit wider for full one-column width, and it is easier to compare time series. When I saw these panels first time, I had an impression that panels a and b are paired and c and d are paired.
- What do error bars in panels b and c show?
- How is the uncertainty range (panel d) derived? Please explain more clearly in the main text.

Figure 7

- Distinguish curve and line in the caption. Pink and blue are curves, while black one is a line.
- Please rewrite this caption; it is quite confusing. I believe that three datasets are normalized to their average values for the 1084-2000 period and their temporal variations are shown relative to those average values. I believe that "1979-1989" and "1850-2011" are typos.

Figure 8

- The caption is confusing. I believe that you want to say "Large-scale atmospheric and sea ice anomalies observed in CESM historical time series (1850-2005) for the years when ice-core-derived SMB is within highest 10% of the all SMB values in the past ~240 years....."

Figure 9

- Change "accumulation" in the figure to "SMB".

- Change the caption so that it is clearer that this figure shows SMB reconstructed with ice cores over the continent.

Figure S1

- Change to "….sections of sampling for major ions at 10 cm and 5 cm intervals. Isotope samples were taken at 5 cm intervals for the entire core."
- 2 sigma is used in Figure 5 to identify volcanic signals, whereas 4 sigma is used in this figure. I don't really see the merit to see this line; remove this line or justify why not 2 but 4 sigma is used here as a reference.
- It is impossible to distinguish light and dark gray colors in the ECM plot. And it is more important to show the 301-point (30 cm?) –smoothed ECM values in the figure because the smoothed ECM was used to facilitate annual layer counting.
- I believe that only ECM data are shown in terms of the standard deviation, but not water stable isotopes and major ions (change the caption).

Figure S2

- Add unit "sigma" to the first ECM panel.

Appendix

- Again, use "accumulation" and "SMB" consistently.
- Are latitude/longitude given in decimal degrees?

---

## Author Response (AR2)

**Author's response**

*Reviewer's and editor's comments are in italic*, author's response in plain characters.

**Editor's comment**
*Dear Ms. Philippe,*

*I now received two reviews for your revised manuscript, which are available for you at the journal web site. The reviewer #1 recommended accepting the manuscript with minor revisions. The reviewer #2 acknowledges that estimating past SMB using the timescale is largely improved, but argued that the timescale presented in the revised manuscript is not convincing enough.*

*I am unable to accept this paper in the current form but happy to see another revised version as a possible contribution to the journal The Cryosphere. Please submit a revised manuscript together with point-to-point responses to all comments brought by the reviewers. Some additional guidance can be found below.*

*1. I agree with the reviewer #1 that the isotope/chemistry sampling was made not at high enough resolution (Figure R1 in the response letter), and that uncertainty in age control using volcanic events and its consequences should be more clearly shown in the manuscript.*

We believe that the uncertainty was already well acknowledged in the manuscript, as it was stated p.8, lines 21-23: "it is very likely that our oldest estimate is closer to the real age-depth relationship than the youngest estimate. However, we will keep both of them as an evaluation of the influence of the dating uncertainty on our accumulation rates reconstruction."

All numbers and trends in the rest of the manuscript take the largest uncertainty into account, since we agreed already with the reviewers who said that the volcanic identifications were ambiguous. We only left Tambora in order to give the reader a good reason to believe that the correct timescale lies within the range identified between the oldest and the youngest estimates.
This does not affect our conclusions because:
- the increased accumulation trend begins in the end of the $2^{nd}$ half of the $20^{th}$ century, where the dating is literally "flawless" (max. 2-3 years difference between oldest and youngest estimates).
- Both the youngest and oldest estimates show an increasing trend.

However, we understand the reviewer's argument regarding the Tambora matching and decided to be even more cautious in our interpretation, in accordance with the editor's recommendations.

The example between 103 and 104 meters did not lead to an ambiguity in the annual layer counting since at least four parameters agree in showing three annual cycles (ECM, $\delta^{18}O$, nssSO$_4$, MSA). Even if SO$_4$/Na is less clear in this case, it is not a sufficient argument to find this layer ambiguous.
This is also a very isolated case. The full range from 101 to 108 m for example shows 22 years with, in our opinion, no ambiguity at all between the $\delta^{18}O$ and the Na/SO$_4$ ratio datasets. More interestingly, very close to the bottom of the core, the section between 112.3 m and 115 m shows 13 isotopically defined annual layers in the oldest estimate and 10 in the youngest estimate. However, the Na/SO$_4$ signal unambiguously reproduces 13 annual layers.

We actually use this excellent match between $\delta^{18}O$ and the Na/SO$_4$ ratio throughout the whole core including the deepest sections to confirm our range of estimate. We also believe this clearly shows that using a technique such as subtracting 1 or 2 more years every 20 or 10 years down from 40 m depth, as suggested by the reviewer would clearly overestimate the potential error on dating. It would be at the most 1/22 years below 100 m, a depth below which there is no obvious trend in SMB anyway.

*2. This reviewer informed me after the review was submitted that the reviewer misunderstood the uncertainty mentioned in the manuscript (0.6%, though it is said 6%). So, the authors do not need to address the comments directly related to this misunderstanding, but respond to the age control and uncertainty issues brought by the reviewer.*
Agreed

*3. I echo reviewer's view on the Tambora identification. Main concerns are: (1) the unknown 1809 event is not found, (2) Tambora and 1809 events do not sustain for several years as typically seen in many cores, and (3) low sulfate and high ECM peaks (or any other data properties) are not convincingly presented to support author's identification of the Tambora event, and (4) isotope data are unavailable from 100-101 m (see the review for the full argument by the reviewer). If the authors maintain the current argument on the Tambora identification and age control in general, please fully respond to the comments and provide argument that is more rigorous in the manuscript.*
We removed the sentence about Tambora in the abstract: "and the identified Tambora 1815 volcanic horizons confirms the oldest age-depth estimate" and reset Figure 5 as the original suggestion of all the volcanic events with a potential match. However, we do not use it to refine our depth-age scale anymore and instead use it to conclude that the background noise on ECM due to the coastal location is too strong. We consider the oldest and youngest estimates as representing our full range of uncertainty (see above).

*(1) the unknown 1809 event is not found.*
This event was found in version 1 at 104 m (New Fig. 5). Note that (i) the relative level of difference is from simple to double, as in the WAIS divide ice core mentioned by the referee (ii) unknown is just above 2σ, a threshold recognized in previous literature.

*(2) Tambora and 1809 events do not sustain for several years as typically seen in many cores*
First, the WAIS divide ice core shown by the reviewer is from 1766 m a.s.l. and 500 km away from the coast, while IC12 is typically coastal. Volcanic peaks are therefore less prominent compared to the background signal in our core. However, we agree that there are other peaks, especially in the nssSO$_4$ record that do not allow us to use Tambora as a strict tie point for our depth scale. This is why we chose to refrain from this interpretation in the revised manuscript.

*(3) low sulfate and high ECM peaks (or any other data properties) are not convincingly presented to support author's identification of the Tambora event*
We compare the raw ECM signal to the normalized signal in Fig. A1 below. Both are very similar, so we are convinced that our treatment of the ECM data did not affect our conclusions.
Density correction was not based on optical televiewer measurements but on a best fit through gravimetric measurements. We made that point clearer in the text. We now provide the manuscript explaining that density correction. We hope that it is now

clearer why we made this correction, and that it did not bring more noise in the dataset.

*(4) isotope data are unavailable from 100-101 m.*
This is unfortunately due to a missing ice core section (fell back in the borehole) and this drilling default is now mentioned in the text: "The ice core is complete, except for the 100-101 m section which fell back in the borehole and was recovered in broken pieces."

*4. I found that the data presented in this manuscript is of high interest and can be publishable even if the Tambora event isn't convincingly identifiable. If the authors pursue such publication, please fully address uncertainties in the timescale and provide a realistic range of uncertainty or possible alternative interpretations of the past SMB (does the revised timescale really support the increasing SMB?).*
Agreed and amended

*5. Many of reviewer #2's comments are associated with inaccurate or incomprehensive information on previous work in DML. For example, the current manuscript only mentions that Altnau found positive SMB trend in the interior DML. However, Altnau also found a negative trend in the ice shelves. Both should be mentioned to give the full picture of previous findings. Similarly, there are many firn cores collected in DML coast (not very few, see reviewer's comment on P3L20 of the revised manuscript), but most of them cover only up to several decades and the DIR ice core is one of only a few that covers nearly a century. Please revise the manuscript so that the manuscript articulates the current understanding in DML coastal region and emphasizes what's exactly new in this manuscript.*
Agreed and amended.

*6. I am afraid that some of Reviewer #2's comments refer wrong page/line numbers (e.g. a comment on P2L20 of the revised manuscript is probably on P3L4 of the revised manuscript). However, the authors are probably able to identify the statements that the reviewer concerned in most cases. If it is unfeasible for some specific comments, please say so in the response letter.*
Agreed. We believe we found and addressed all comments of referee #2

*7. Your manuscript cites Fig. 10 (P13L6ff), but the manuscript includes only 9 figures.*
Amended

*Editorial comments:*
*- I think that the authors use "accumulation" and "surface mass balance" interchangeably. Please distinguish them and use these terms consistently through the manuscript.*
Amended

*- Does the depth axis show physical depth or depth in water/ice equivalent? Table 2 shows that 2011 SMB is 0.98 m w.e./a, and Figure S1 shows that the 2011 layer is to about 1 m depth (I assume that the surface density is below 500 kg/m3).*
All figures show physical depth except Fig. 2, which is in ice equivalent, as we have now clearly indicated in Fig. 2.
The 2011 layer is the one between 1.07 m and 3.14 m so it is 2.07 m thick. With snow density of 473 kg/m at that depth, this gives 0.98 m w.e.
The 2012 layer is incomplete since the ice core was drilled in November 2012.

*- Figure 2: all depths should be positive.*
Amended

*- Figure 6: it's hard to distinguish curves in different colors. Please use more distinct colors.*
Amended

*- Figure 7: is the ERA-Interim record shown relative to the 1980s record, while CESM and ice core data are relative to the 1860s record? Is it better to show all of these data relative to the latest years?*
Agreed and amended

*Thank you for submitting the manuscript to TC/TCD. Both reviewers and I found significant improvements in your manuscript through the review process, and I encourage you to submit a revised manuscript for further consideration.*

*Sincerely,*

*Kenny Matsuoka*
*TC/TCD editor*

**Report 1 and author's response**

*Philippe et al. TCD*
*The authors put a lot of work into the revised version and the manuscript has been greatly improved. However, some points remain, which I will address in the following.*
*First a remark: the font size of the manuscript pdf file is a bit of an imposition, it was completely unnecessary to keep the formatting info on the right side. Especially with all the marked changes it was very hard to read. I suggest to avoid that in the future.*
*Comments to reply to specific comments (line numbers refer to the original manuscript:*
*P2, l23: you still do not mention that Altnau et al. found a negative trend for SMB at the coast at this point. Why?*
We mentioned it in section 4.3 but forgot to do so in the introduction. It is now included in both sections.

*P6, L3: your reply should be discussed in the text.*
We added to the text: "The CESM simulated sea-ice extent in the observational period is very realistic compared to observations (Lenaerts et al., 2016) and does not show any trend in the Atlantic sector, which gives us confidence that the sea ice is treated realistically."

*P10, L5: I still think that an SMB of 0.3 is difficult to use as a threshold since many sites at the coast have SMBs just around 0.3. so slightly above or below 0.3 would not mean a systematic difference here. (I am not sure why Massimo Frezzotti chose it in the first place.)*
Agreed. Although we already mentioned in the manuscript that these high accumulation sites were not all coastal, we now show the difference between coastal and inland sites in Fig. 9, using a distance of less than 100 km from the coast and below 1500 m a.s.l. to define coastal sites.

*P12, L20: Please, discuss this in the text, too. The high SMB in 2009 and 2011 found by Lenaerts was mainly due to the atmospheric circulation patterns during those years. Those patterns have a much stronger influence on SMB than a couple of days longer or shorter sea*

*ice coverage. (and keep in mind that sea ice extent refers to 15% sea ice concentration, so plenty of open water available for evaporation.)*
We agree that the atmospheric circulation largely determines the variability, and sea-ice and sea-surface temperature conditions play a secondary role. We have included this before the last paragraph before the conclusions:
"Atmospheric circulation exhibits a primary role in determining temporal and spatial SMB variability. Sea-ice and ocean surface conditions play a secondary role, and could contribute to a higher SMB in a warmer climate. A more recent study…."

*From now on, the line numbers refer to the revised version*
*P2*
*Ll9: what does it mean that this is the only record that supports model results?? Are all the other cores not representative and the DIR core is the only representative one? This is not self-evident.*
We think that the line numbers are wrong here but if you refer to the abstract, we removed "thereby supporting model predictions".

*L20: this is not consistent evidence: e.g. Fudge et al. found that SMB and temperature are not always positively correlated.*
*Fudge, T. J., B. R. Markle, K. M. Cuffey, C. Buizert, K. C. Taylor, E. J. Steig, E. D. Waddington, H. Conway, and M. Koutnik (2016), Variable relationship between accumulation and temperature in West Antarctica for the past 31,000 years, Geophys. Res. Lett., 43, 3795–3803, doi:10.1002/2016GL068356.*
Amended

*P3*
*L 18: see above, negative trend in SMB in coastal cores in Altnau et al.*
Amended

*L20: I would not call this "very few", there are quite a few investigations of DML cores from German, Scandinavian and Indian expeditions.*
We modified the sentence and other instances where such confusion was made: "few studies focused on ice cores spanning more than 100 years"

*L21: higher than the interior (a comparative needs something to compare to)*
Amended

*P4*
*L15: ice rises are too small to "block" atmospheric circulation. (this would mean that the air flows AROUND the ice rise rather than over it. Blocking is a clearly defined term in meteorology.*
We changed it to 'disrupt'

*L18-22: good!*
*P9*
*L3: this is not correct, there is ERA20C (ECMWF) meanwhile, which covers the entire 20th century.*
Correct, but ERA-20C suffers from severe data gaps in the early 20th century over the Southern Ocean (Titchner et al., 2014), which makes it useless for this purpose.

Titchner, H. A. & Rayner, N. A. The Met Office Hadley Centre sea ice and sea surface temperature data set, version 2: 1. Sea ice concentrations. J. Geophys. Res. Atmos. 119, 2864–2889 (2014).

*P12*
*L7: delete "is"*
Amended

*P17*
*L4. Better: source region of atmospheric moisture for DIR.*
Amended
*Fig. 8 should be described as part of the results section.*
Amended

*L2-8: sea ice is only one factor. The same factor that causes high accumulation might influence the sea ice extent without changes in sea ice being the reason for the accumulation deviations. I still do not find this paragraph very convincing. E.g. high sea ice extent related to lower air temperatures might be caused by a generally more zonal atmospheric circulation pattern, which at the same time could be the reason for low accumulation due to lack of meridional moisture transport (as Lenaerts et al. showed for 2009 and 2011. These things should be discussed in the text.*
We agree with the reviewer, which is why added to the text (see above): "Atmospheric circulation exhibits a primary role in determining temporal and spatial SMB variability. Sea-ice and ocean surface conditions play a secondary role, and could contribute to a higher SMB in a warmer climate."

*L18: see above. Does it not make you think that no other coastal core in DML shows an increase in SMB?*
Yes, even if we consider shallow cores, all of them show a decrease or no significant change.

*L27: see above, please explain the physical reason for the choice of the threshold.*
We don't use anymore SMB threshold but rather define coastal sites as those situated less than 100 km away from the coast and below 1500 m a.s.l.

*L31: if we compared*
Amended

*P20*
*L1-7: see above, maybe quote Fudge et al. here, too.*
The sentence was changed to: "However, both Altnau et al. (2015) and Fudge et al. (2016) found that SMB and changes in ice $\delta^{18}$O are not always correlated. They hypothesized that changes in synoptic circulation (cyclonic activity) have more influence than thermodynamics, especially at the coast."

*L8ff: this is still not clear. Atmospheric rivers don't occur for the whole year, just for certain events. Precipitation at the coast is usually event-type, but the events occur during the whole year, whereas in the interior those events happen not very often, but are related to amplified Rossby waves.*
This paragraph was changed by "In the presence of a blocking anticyclone at subpolar latitudes, an amplified Rossby wave invokes the advection of moist air (Schlosser et al., 2010;

Frezzotti et al., 2013). On these rare occasions, meridional moisture transport towards the interior in DML is concentrated into atmospheric rivers. Two recent manifestations of these short-lived events, in 2009 and 2011, have led to a recent positive mass balance of the East Antarctic ice sheet (Shepherd et al., 2012; Boening et al., 2012). It was also observed in situ, at a local scale, next to the Belgian Princess Elisabeth base (72 °S, 21 °E) (Gorodetskaya et al., 2013; 2014). Several of these precipitation events in a single year can represent up to 50 % of the annual SMB away from the coast (Schlosser et al., 2010; Lenaerts et al., 2013). At the coast, precipitation is usually event-type, but the events occur during the whole year. However, the 2009 and 2011 events are also observed in our data as two notably higher than average SMB years (2009 and 2011, Table 2)."

*L25: 2009 and 2011*
Amended

*P21*
*L8: wind is certainly a very important factor, but e.g. the interannual variability in the years 2009-2011 was definitely not mainly caused by the wind. Be careful with general statements like this. Of course, in years with fairly "average" flow patterns, the wind is the main factor, that is correct.*
Amended

*L21: why should there be a decrease at the coast then?*
Because the moisture is transferred inland.

*L24ff: see comment on P1*
Already answered above

*Parts of the discussion actually belong into the results section.*
Amended

*p23*
*l16ff: see above.*

We changed the sentence « Our analysis based on CESM output suggests that accumulation variability is also potentially explained by changes in sea ice cover combined with regional atmospheric changes. » to "Our analysis suggests that atmospheric circulation to a great extent determines SMB variability, with a potential secondary role of changes in sea ice cover".

**Report 2**
*Review of Revised Phillippe et al., 2016 manuscript:*
*The revised manuscript by Phillippe et al. has notable improvements, particularly in regards to the ice-flow modeling and reconstruction of the accumulation rate history. The collaboration with other researchers doing ice-dynamics work has led to a robust and convincing correction of strain thinning.*
*Unfortunately, the development of the timescale remains flawed. The attempt to identify Tambora is unconvincing and undermines the entire development of the timescale. A critique of the timescale development is below.*
*Overall, the authors need to admit that they:*

*- cannot reliably identify any volcanic events and thus have no age control beyond annual layer counting*

*- did not sample the core at high enough resolution and may thus be overcounting, predominantly in the deep core*

*- and thus the interpretation of an increase in recent accumulation is tentative and more work is needed to answer this interesting question definitively.*

*This work can become published, but only after giving up on identifying volcanic peaks and giving an honest and detailed assessment of the annual layer interpretation and the likelihood of overcounting in the deeper part of the core. The type of questions that needs to be answer is: What would the record look like if you subtracted 1 year in every 20 below 40 m depth? Or 1 year in every 10? And then discuss whether your annual layer count is reliable at that level. One approach could be to use automated techniques (such as Mai Winstrup's straticounter: https://github.com/maiwinstrup/StratiCounter) on the different data sets to get a sense of the uncertainty. My guess is that the real uncertainty is closer to 6% than 0.6%.*

**The Timescale:**

*The paper hinges upon accurate interpretation of the depth-age relationship. The main conclusion is that the accumulation rate has increased in recent decades compared to the previous couple of centuries. This interpretation relies both the timescale and the corrections for density and ice-flow induced thinning. The authors have greatly improved the corrections, which can now be both understood and trusted. However, the timescale remains both poorly described and untrustworthy.*

*The magnitude of the inferred accumulation rate depends directly upon the thickness of the identified annual layers. Thus, supporting the interpretation of increased recent accumulation requires showing that the annual cycles are properly identified. The authors attempt to do this in two ways: 1) the presentation of data with distinct seasonal cycles to convince the reader that the seasonal cycles at all depths of the core are unambiguous (or at least nearly unambiguous) and 2) identify horizons (such as volcanic events) that can be tied to other cores (or other paleoclimate records) that confirm the annual cycle interpretation.*

**-Tambora**

*I recommended focusing on Tambora in my previous review but am deeply frustrated by the revised manuscript. It is worth discussing here why Tambora is so distinctive in Antarctic ice core records. Tambora is indeed the largest event in the past few hundred years. But part of the reason it is so distinctive is that is in preceded by the second largest event of the past few hundred years, yielding distinctive double peak. I've attached a figure of Tambora and the unknown events as shown by Sigl et al. (2013) from the WAIS Divide ice core. It is worth noting that both events have durations of 3 years.*

*The purported Tambora in IC12 lacks all resemblance to this event found elsewhere.*

*1) the authors make no attempt to identify the preceding event (commonly know as the unknown 1809 event).*

*2) The sulfate peak associated with Tambora is not even the largest in the figure. The authors do not explain why the ECM is so anomalously high while the source of acidity, H2SO4, does not result in high SO4 levels.*

*3) The figure starts at 101 m, conveniently hiding the fact there is no data from 100 to 101 m – something that I do not believe is discussed in the text.*

*4) The lack of chemistry measurements of the full core means there is no ability to reliably identify and compare SO4 peaks along the core*

*5) The ECM data is heavily filtered, indicating it has major quality-control issues and reducing any confidence that it can reliably detect volcanic events. Further, the filtering methods are unclear with the techniques used reference (Karlof et al. 2000) being more*

*complicated than what is described in this text (just the Savitsky-Golay filter and normalization) – I'm still not sure what was done to the ECM data presented here. Further again, the ECM data is corrected for density based on borehole optical televiewing that is likely not that well depth-referenced (the methods are not described except for an unprovided in review manuscript) and regardless, is not appropriate for ~10 cm scale variations anyway, as explicitly stated in the given reference, Hubbard et al., 2013. This technique likely just introduced a bunch of noise at annual-to-volcanic frequencies into an already noisy record.*
*6) The authors claim that the ECM peak occurs in wintertime, despite being in an Na trough with a clear So4/Na peak (and hence SO4). Though the water isotopes are indeed unusual, there is a shoulder which may indicate a lack resolution to identify peaks. This seems more likely to be a thin year than volcano, let alone Tambora. Further, the logic of a wintertime peak is faulty. Tambora is a multi-year event such that the peak should be highest in summertime when there is both volcanic deposition and ocean-derived deposition. I guess the authors could argue that the coastal characteristics of deposition could truncate the duration of Tambora - but they would need to do that and be convincing with climate model output and observational data.*

*My biggest issue with the "identification" of Tambora is that it is so far from convincing I simply have no trust in the authors development of any part of the timescale. This is particularly important because the timescale was clearly developed iteratively; the chemistry measurements were only made in areas of uncertain annual layering indicating that the annual layers (of d18O and ECM) were interpreted prior to the chemistry measurements clarifying the annual layer iterpretation. It is likely that the volcanic matching was also done before, and thus the annual layer interpretation may have (consciously or subconsciously) been interpreted to get the right age at Tambora. While this sort of issue is not uncommon producing annual timescales, the lack of awareness of this issue in the manuscript is troubling especially given the propensity to pick too many years in undersampled data.*

See response to editor's question 3

*- **Annual layer interpretation***
*Evaluating annual interpretation in publications has to be based largely on trust since it is difficult to present that data and interpretation in a manageable way. The authors do a good job of presenting the data with the addition the supplementary figures. However, the authors fail to convince me of their uncertainty, and hence the underlying timescale. This is in part because of the unconvincingly attempt to identify Tambora, but also because the measurements are just not of sufficient quality and continuity to get 0.6% accuracy. Some of the issues:*
*- The stable isotopes, the primary parameter used to identify annual layers, are of insufficient resolution much of the time. The histogram in Figure R1 shows that the mode of the number of data points per annual layer is 4, with a significant number of layers identified with 3, 2, and even 1 data point. 4 data points is not enough for annual layer interpretation, less alone fewer. The authors may say that the other data sets define these layers, so I address that below:*
*- The ECM data is heavily filtered with a 301-point window and unclear other techniques (see above). The authors do not describe how this impacts the annual interpretation, which is a major shortcoming. But what I see of the ECM data suggests to me that it is of questionable reliability for interpreting annual layers, with an uncertainty of 10% not less than 1%.*

*- The chemistry data is sporadic with samples sizes that appear larger than for the stable isotopes (it's hard to see in Figure S2), such that the same criticisms of the stable isotope record apply to the chemistry records.*

*For instance, between 103 and 104 meters, there is a sequence of 3 thin years which is interpreted exactly the same in the oldest and youngest scenarios (1837-1835 or 1813-11). Yet the Na/So4 looks like there are only two years. In both cases, the sampling frequency is too low to be sure peaks and troughs are being resolved. This type of interpretation may be finding an extra year, biasing the annual layer thickness low, and underestimating the accumulation rate. While individual interpretations can always be nit-picked, the real concern here is that the authors do not even acknowledge the uncertainty.*

See response to editor's question 1.

Globally, we have now abandoned the use of volcanic markers to refine our relative dating uncertainty. We still believe that our oldest estimate is the best one, since it shows coherent synchronicity of $\delta^{18}O$ and $Na/SO_4$ signatures all the way down to the deepest part of the core. However, we now make it clearer that we treat our data set as an "uncertainty" range (see e.g. Fig. 7).

In our opinion, using the technique of ignoring 1 out of 10 or 20 layers below 40 m, as suggested by the reviewer, would bring us away from the one-to-one $\delta^{18}O$ - $Na/SO_4$ synchronicity observed for the oldest estimate even in the deep layers, therefore overestimating the errors.

Also, clearly, using our "uncertainty range" does not jeopardize the main conclusion of the paper that there exists a trend of increasing SMB, at least from the mid-20[th] century.

[Figure]

Fig. A1. Raw ECM data (10 mm running average, left axis) and normalized ECM data (right axis), respectively before and after "density correction, normalizing and Savitsky-Golay filtering".

[revised manuscript text omitted]
 SMBaAAccumulation rates for three different periods (chosen mainly for easier comparison to with previous studies) starting from theare calculated on the basis of deformation corrections (Nye and D J) and averaged over various periods framedaveraged over three different periods between by Tambora eruptionand Cerro Azul volcanic horizons and the surface (1816–2011):, the last 111 years compared to the full period of time (i.e. 1900–2011 cf. 1816–1900), the last 50 years compared to the previous full period of timeprevious (i.e., 1962–2011 cfvs. 18166–1961), and and the last 20 years compared to the previous full period of time previous (i.e. 1992–2011 cfvs 1816–1992), for the youngest and oldest estimates and average between both (Table 1e.g. Kaczmarska et al., 2004, Sigl et al. 2012; bold years in Table 1) are shown in Figure 6b and summarized in Table 2). The long term annual accumulation, starting from the oldest volcanic layer identified:From 1768 1816 to 201112, the average accumulationSMB rate is between 0.39 and 0.460.49 ± 0.02 m w.e. a$^{-1}$ depending on the correction applied. For the last 111 years, the SMB is 0.55 ± 0.02 m w.e. a$^{-1}$, representing a 26 ± 1 % increase compared to the previous period. For the last 50 years (Table 2). The recent (19551962–20111)2), the accumulation rateSMB is between 0.60 61and 0.63 ± 0.01 m w.e. a$^{-1}$ with, as expected, less impact from the different deformation corrections, representing a. The sharpest increase occurs between the periods 1902-1955 and 1955-1992 (36% to 45% increase). With a 31 years running mean, the rate of accumulation change between 1902 and 1992 is 0.21 m w.e. a$^{-1}$ (data not shown). 32 ± 4 % increase compared to the previous period. For the last 20 years (1992–2011), the accumulation rateSMB is 0.64 ± 0.01 m w.e. a$^{-1}$ and the increase compared to the previous period is 32 ± 3 %.

Table 3 shows the detailed annual accumulation rates for the last 10 years for both corrections. The highest accumulation of the last 10 years occurred in 2009 and 2011, which belong to the 3% and 1% highest

accumulation years of the whole record, respectively. Table 2 shows the detailed annual accumulation ratesSMB for the last 10 years for our oldest and youngest estimatescorrections. In boththe oldest estimates, tThe highest accumulationSMB during the last 10 years occurred in 2011 and 2009, which belong to the 1 % and 3 % highest accumulationSMB years of the whole record, respectively. In the youngest estimate, 2002 is higher than 2009.

**3.3 Sources of uncertainties**

AccumulationSurface Mass Balance rates reconstructed from ice cores can be characterized by substantial uncertainty show(Rupper et al., 2015). The accuracy of reconstructed snow accumulationSMB rates depends on the dating accuracy, which, in our corase, is determined byAs discussed before, v the oldest and youngest estimates. Also, given our vertical sampling resolution of $\delta^{18}O$, the location of summer peaks is only identifiable to a precision of 0.05 m where no other data are available, but this error only affects inter annual SMBaccumulation rates at an annual resolution, as shown by error-bars in Fig. 6. Note also that it is very unlikely that we have overestimated the number of years due to the $\delta^{18}O$ sampling resolution, since a one-to-one correspondence subsists, in the deepest part of the core, between the $\delta^{18}O$ and the $Na^+/SO_4^=$ ratio. time

SMB reconstructions are also influenced by density measurement error (2 % error) and small-scale variability in densification. The influence on SMB is very small. Callens et al. (2016) for example, used a semi-empirical model of firn compaction (Arthern et al., 2010) adjusting its parameters to fit the discrete measurements instead of using the best fit from Hubbard et al. (2013). Using the first model changes our reconstructed SMB values by less than 2 %.

Average SMB on longer time periods are therefore in all cases more robust than reconstructed annual SMB because they are less affected by uncertainties. These average estimates are also useful to reduce the influence of inter-annual variability.

Vertical strain rates also represent a potential source of error. A companion paper will be dedicated to a more precise assessment of this factor using repeated borehole optical televiewer stratigraphy. However, the present study uses a field-validated strain rate models which is as close as possible to reality, and shows that the using sithe simpler modified Dansgaard–Johnsen model changes the reconstructed SMBaccumulation rates by maximum 0.001 m w.e. a$^{-1}$. Therefore, we are confident that refiningknowing the exact strain rate profile will not change our main conclusions.

Average accumulation rates on longer time periods are therefore more robust than reconstructed annual accumulation rates because they are less affected by uncertainties. These average estimates are also useful to reduce the influence of inter annual variability.

Uncertainties are also influenced by density measurement error 2 %and small scale variability in densification. The influence on accumulation rates is very small. Callens et al. (2016) for example, used a semi-empirical model of firn compaction (Arthern et al., 2010) adjusting its parameters to fit the discrete measurements instead of using the best fit fromin Hubbard et al. (2013). Using the first model changes our reconstructed accumulation values by less than 2 %. Another possible source of possible error is the potential migration of the ice divide. Indeed, radar layers show accumulationSMB asymmetry next to the DIR divide. therefore, had induced non-climatic ratesHowever, rRhHoweverowever, two recent analysesDrews et al (2015) found that the ice divide of the DIR must have remained laterally stable for thousands of years to explain the comparatively large Raymond arches in the ice stratigraphy. Callens et al. (2016) find a similar argument by using the radar stratigraphy in the ice-rise flanks. The possibility for an ice-divide migration is therefore small indicate that there is a very low probability that such a migration occurred as the DIR has been stable for at least the last thousands of years (Drews et al., 2015; Callens et al., 2016). Temporal variability of accumulation ratesSMB at certain locations can also be due to the presence of surface undulations upstream (e.g. Kaspari et al, 2004), but this effect is minimised at ice divides.

Average accumulation rates on longer time periods are therefore more robust than reconstructed annual accumulation rates because they are less affected by uncertainties. These average estimates are also useful to reduce the influence of inter-annual variability.

**3.43 Relation to atmospheric and sea ice patternsComparison with climate models**

Figure 8Figure 7 compares the trend in our IC12 SMB record with outputs from two atmospheric models: ERA-Interim reanalysis (Dee et al., 2009) and the CESM model. ERA-Interim shows no trend in the Interestingly, in the relatively short overlapping period (1979–2012) it covers, which is not surprising since it is too short to be of climatic significance. Tthe ice core derived SMB correlates rather poorly moderately to ERA-Interim reanalysis (Dee et al., 2009) and RACMO2 (Lenaerts et al., 2014), yielding (both correlation coefficient Rr² = 0.36 and 0.5 0.4, respectively not shown). This much poorer correlation than that compared to ice cores collected on West Antarctica (Medley et al., 2013 (GRL),; Morris et al., 2015 (Nat Geo)) is presumably explained by the strong impact of local wind induced snow redistribution and sublimation on the SMB on the wind exposed ridge of the Derwael ice rise (Lenaerts et al., 2014). For a longer overlapping period, we used the output of the CESM model, although it is a freely evolving model that does not allow a direct comparison with

measured data. The average SMB at Derwael in CESM (closest grid point) is too low ($0.295 \pm 0.061$ m a$^{-1}$) because the orographic precipitation effect is not well simulated with the low model resolution. Figure 8 shows the relative trends in CESM output and ERA Interim compared to the IC12 record. However, ERA Interim shows no trend in the short period 1979-2015 but the period is too short to explore the mechanisms. Instead, CESM does reproduce (much of) the observed trend. Subtle small-scale variations in wind speed and direction, typically not resolved by reanalyses or regional climate models, might disrupt the inter-annual variability of SMB, although we assume that it does not impacinfluencet the positive SMB trend found in the ice core record. Unfortunately, our method does not allow for an explicit partitioning of the SMB explained by precipitation as opposed tovs. wind processes. Instead, we focus on the drivers of precipitation at the ice core site using the output of CESM (Fig. 8), and we discuss it in Sect. 4.1..

In anomalously high accumulationSMB years, sea ice coverage is substantially lower than average (20–40 fewer days with sea-ice cover, fig. 8) in the Southern Ocean northeast of the ice core location, which is the prevalent source region of atmospheric moisture for DIR (Lenaerts et al., 2013). This is associated with considerably higher near-surface temperatures (1–3 K). In low-accumulationSMB years (not shown), we see a reverse, but less pronounced signal, with higher sea ice fraction (10–20 days), and slightly lower temperatures at the oceanic source region of precipitation. 9). In anomalously high accumulation years (top panel), , the sea ice coverage is significantly lower than average (20-40 fewer days with sea-ice cover) in the Southern Ocean northeast of the ice core location, which is the prevalent source region of the atmospheric flow (Lenaerts et al., 2013). This is associated with significantly higher near surface temperatures (1-3 K). In low accumulation years (not shown), we see a reverse, but less pronounced (not significant) signal, with higher sea ice fraction (10-20 days), and slightly lower temperatures and the oceanic source region of precipitation.

Figure 9. Large-scale atmospheric, ocean and sea-ice anomalies in high-accumulation (10% highest) years in the CESM historical time series (1850-2005). The colours show the annual mean near-surface temperature anomaly (in °C), and the hatched areas show the anomaly in sea-ice coverage (>20 days less sea-ice cover than the mean). The green area shows the location of the ice core.

Figure 7 shows a summary of the output from the CESM as described in Section 2.5. In anomalously high accumulation years (top panel), the sea ice coverage is very low (20-40 fewer days with sea-ice cover) in the Southern Ocean northeast of the ice core location, which is the prevalent source region of the atmospheric flow (Lenaerts et al., 2013). This is associated with higher near surface temperatures (1-3 K), and a strengthening of

the low climatological low pressure system (>1 hPa lower surface pressure), located offshore the ice core location (Lenaerts et al., 2013). In low accumulation years (bottom panel), we see a reverse, albeit less strong, signal, with higher sea ice fraction, lower temperatures and higher core pressure of the low pressure system.

**4 Discussion**

**4.1 Regional-scaleSmall-scale variability**

Output of the CESM show that, along with atmospheric circulation, sea-icSea ice cover and near-surface temperatures at the ice core location also have an influence on precipitation at a regional scale, as shown by the output of the CESM (Fig. 8).

Figure 9. Large-scale atmospheric, ocean and sea-ice anomalies in high accumulation (10% highest) years in the CESM historical time series (1850-2005). The colours show the annual mean near-surface temperature anomaly (in °C), and the hatched areas show the anomaly in sea-ice coverage ( >20 days less sea-ice cover than the mean). The green area shows the location of the ice core.

Small-scale spatial variability in cyclonic activity and atmospheric rivers could both explain why our results are different from others in the same region, and why they correlate only moderately to the climate reanalyses (ERA-Interim and RACMO2). Orography can also greatly affect spatial variability in SMBsnow accumulation variability (Lenaerts et al., 20143). Local wind phenomena are important factors of interannual and spatial variability. Indeed, the lowpoorer correlation with ERA-Interim and RACMO2 in our study, as compared to ice cores collected on West Antarctica (Medley et al., 2013; MorrisThomas et al., 2015) is presumably explained by the strong influencempact of local wind-induced snow redistribution and sublimation on the SMB on the wind-exposed ridge of the Derwael ice riseDIR (Lenaerts et al., 2014).

However, Callens et al. (2016) showed that thise spatial pattern has been constant for the last thousands of years. Therefore, ourthe observed trend of increasing annual accumulationSMB is highly unlikely to be explained by a different orographic precipitation pattern caused by a change in local wind direction or strength, which would cause a different orographic precipitation pattern.

This argument, along with the existing correlations with ERA-Interim and RACMO2, suggests that the observedisese trends isare not only representativelimited of the climate on to the DIR the Roi Baudouin ice shelf

[revised manuscript text omitted]

**Figurese caption**

~1960_present _vs ~1816_present

[Figure]

~1990_present _vs ~1816_present

[Figure]

Change in accumulation (%)

[Figure]

~~Figure S1. Full vertical profile of water stable isotopes with a grey and black band on the left indicating sections of 10 cm and 5 cm resolution, respectively (a); major ion (b–f), normalized ECM conductivity expressed as multiple of standard deviation (σ) (light grey: 1 mm resolution, dark grey: 0.05 m running mean). The 4σ threshold is shown as a dotted vertical line, and identified volcanic peaks as dashed grey horizontal lines (g);~~

Figure 1: Location of IC12 and other ice cores referred to herein. (a-b) Difference in mean annual SMB between the period ~1960–present and the period ~1816–present (see Table A1 for exact periods); (a-b). (c-d) Same as (a-b) for the period ~1990–present compared to ~1816–present. (c-d). Panels (b) and (d) are expansions of the framed areas in panels (a) and (c).

Figure 1: Location of IC12 and other ice cores referred to in the discussion. Difference in mean annual SMB for the periods ~1960–present (a-b), and ~1990–present (c-d), compared to the period ~1816–present (see Table A1 for exact periods). Panels (b) and (d) are zooms of the framed areas in panels (a) and (c).

[revised manuscript text omitted]

running mean). The 4σ threshold is shown as a dotted vertical line, and identified volcanic peaks as dashed grey horizontal lines; annual layer boundaries in the youngest (Green) and the oldest (Blue) estimates. Each colour transition indicates a boundary.

Figure S1. Full vertical profile of water stable isotopes with a grey and black band on the left indicating sections of 10 cm and 5 cm resolution, respectively (a); major ion (b–f), normalized ECM conductivity expressed as multiple of standard deviation (σ) (light grey: 1 mm resolution, dark grey: 0.05 m running mean). The 4σ threshold is shown as a dotted vertical line, and identified volcanic peaks as dashed grey horizontal lines (g); annual layer boundaries in the youngest (Green) and the oldest (Blue) estimates. Each colour transition indicates a boundary (h).

---

## Author Response (AR3)

**Editor's comments**
Author's response in red

P1L20: SMN -> SMB. Also, sensitivity of what? Done (P1L19)

P1L22: "120-m-long" Done, although the website http://www.the-cryosphere.net/for_authors/manuscript_preparation.html indicates: "It is our house standard not to hyphenate modifiers containing abbreviated units (e.g. "3-m stick" should be "3m stick"). This also applies to the other side of the hyphenated term (e.g. "3m long rope", not "3-m-long rope")."

P1L23: DML is not used in the abstract. Done (P1L22)

P1L26: The bottom of the ice core is dated as 1759 +/- 16 AD, so that the ice core includes the climate proxies in the past 240 years, not only in the 20th century as it is said in the manuscript title and at the end of the abstract (20th and 21st centuries).
With our uncertainty range, we can say that the increase likely starts in the early 20$^{th}$ century and very likely in the mid-20$^{th}$ century, and is continuing in the 21$^{st}$ century. The title and the abstract summarize this main finding of the paper.

P1L32: What does "in at least the last 50 years" mean? I think that the authors want to say "Reconstructed SMB increases with time in the last 50 years by 30%" or "Reconstructed SMB increases with time, and this trend becomes even clearer in the past 50 years". Done (P1L28)

P2L11: see my comment above about "20th and 21st centuries." Same answer as above

P2L18: remove "coastal". Ice discharge always happens at the coast. Done (P2L11)

P2L21: balanced -> compensated? Done (P2L14)

P3L20 -24: Please rewrite these new sentences. In particular, I have no idea what "with high SMB" means at the line 21. Done (P3L27)

P4L2: "120-m-long" Done (P4L27)

P4L8: remove ", including DML," Done (P5L3)

P4L14-15: "preliminary ice core analysis" refers an earlier work of what is reported in this manuscript, I believe. So, it is not appropriate to cite that result in this way. Is it possible to show stake-measured SMB instead (as it is independent of the ice core work reported here)?
This sentence has been removed. It was indeed the value derived from early ice core analysis, which was not published at that time but has been used in Drews et al. (2015) and Callens et al. (2016). (P5L8-9)

P4L26: Change to "Radar stratigraphy shows that locally maximum SMB happens about 4 km upwind of the ice core site"? And consider adding a figure showing the ice core site together with layer-depth SMB and surface elevations in contours (similar to Fig. 4 of Drews et al., 2015).
Done the former (P5L20). Both the ice core position and layer-depth SMB and surface elevations are available in Figs. 1 and 4 of Drews et al. (2015) so we are not convinced that a figure is necessary.

P7L5-10: Because Kjær et al. is still under review, please include sentences that describe the magnitude of this correction. I assume that this method removes long-term trends but not short-term variations so that determining annual cycles in the ECM record is not sensitively affected by this correction. Also, it is hard to match depths of the ice core and borehole (Hubbard et al., 2013) precisely (Reviewer pointed out this issue but the authors did not respond clearly).
Kjær et al., 2016 is now published online. These sentences have been reworded (P7L20-22). However, note that we actually do not use the televiewer log depth scale because we only use the best fit on gravimetric density.

P7L28: change to "and rheological anisotropy of the ice. The strain rates are insensitive to the surface thinning and the strain rates remain the same even if the surface elevation is kept uniform in the model" or such. Also, be more specific which Drews's model result is used here. I think you used "A(n=3), dH = 100, chi = 0.03 m/a, layer-depth SMB, (anisotropic rheology)" in Fig. 11 of Drews et al.
Reworded to: "The magnitude of applied surface lowering is small and does not alter the strain rates significantly compared to a steady-state scenario" (P8L9-11)

P7L29: change to "Separately, we used GPS data to derive the horizontal strain on the surface."
Reworded to add more precision. (P8L11)

P7L30: change "0.002 a-1" to "2 x 10-3 a-1" Done (P8L14)

P8L1: What does "scaled" mean here? Do you mean "The vertical strain rates derived by Drews et al. (2015) is xx so we increased (or decreased?) Drews's vertical strain rate by xx uniformly"?? Even with this change, it is unclear what "best fit" means (if Drews's profile is simply shifted, not shape of the depth function is changed).
This sentence has now been reworded (P8L14-17). Scaled means that the shape of the vertical velocity profile was used and scaled to the long-term accumulation rate, since a lower acc. rate was used by Drews. Since the shape of the profile determines $\varepsilon_{zz}$ (and not the absolute values), this can be safely done.

P8L7: Change "alternatively" to "The other method we used to derive the vertical thinning rates is …." Done (P8L18)

P8L12-13: Figure 2b shows that Drews et al. and DJ model show distinct e_zz over the ice-core depths. They are different by ~13%. Is it significant for your discussion, i.e. do you need to develop the historical SMB records each for Drews's strain rate and for DJ strain rate?
It is shown in Fig. 6 that this difference in strain rate does not affect the historical SMB records, as the curves corrected with both corrections are overlaying. It was hard to see the green line in Fig 6a because the black line appears on top, so we made the green line thicker.

P8L15: do you mean "We used Drews's and DJ's strain rates to compensate dynamic thinning in the annual layer thickness in order to estimate past SMB."? Done (P8L26-29).

P9L11: Please add a sentence to describe how this model is used in this paper. Done (P9L2-3).

P9L26: What do you mean by "trend"? Is it east-west trend, temporal trend?? Done (P9L17).

P10L7: change to "These properties change smoothly over a few very thin ice layers (white dots in Fig. 3) so we assume that they are not disturbed by surface melting". Rewritten close to suggested (P9L25).

P10L22: change to "the reference surface (November 2012 AD)" Done (P10L10).

P10L23: change to "correspondingly dated to 1775 AD and 1743, respectively, or 1758 +/- 16 AD." (the mean of 1775 and 1743 is 1759, not 1758, but I assume that this difference is related to the timing of the drilling in 2012). Done (P10L11-12)

P10L28: Do you want to say "Hereafter, we examine volcanic signals in ECM signals as possible age controls to more precisely develop the depth-age scale bounded by the oldest and youngest cases." Done (P10L17-18).

P11L9: Be careful to say "threshold". If I understand correctly, the authors want to say "the preliminary depth-age scale developed with layer counting shows that the largest ECM peak beyond 4 sigma presents at 1815 so we interpreted it as the Tambora eruption. The secondary peak associated with the Tambora peak is found as well (unknown source, 1809) but its ECM peak reaches only 2 sigma. This ECM peak is lower than those found in most ice cores [ref] but still in a range of previously reported values [ref]. We found 13 other ECM peaks beyond 2 sigma, which can be potentially matched with known volcanic events. Nevertheless, there are many other ECM peaks beyond 2 sigma as well. So, we conclude ….". Rewritten close to suggested (P10L25-P11-L6)

P11L17: "absolute", not "relative"? I believe that the authors say that the absolute dating using volcanic eruptions remain uncertain. Reworded (P11L4).

P11L19: The response letter says that the authors prefer the oldest estimate. If it is the case, develop the argument here further and say something like "We believe that the oldest depth-age scale is more realistic than the youngest estimate because of matching with the Tambora eruption, though it is not really convincing. Therefore, we use …" Done (P10L27)

P12L6: Add thinning rate corrections. Done (P11L19)

P12L8: The authors said that they cannot conclude whether the young or old depth-age scales are better, but the age of the ice-core bottom shown here (1744) is probably tied to the oldest estimate (but if so, it would be 1743 not 1744). Also, Why is the youngest age in the core changed to 201 from November 2012? Changed to: from 1759 ± 16 years to November 2012 (P11L20)

P12L9: move the sentence "without correction for layer thinning" above so that you report the layer thickness first, and then derived SMB. Also, show the range of annual layer thickness (max, min, mean), instead of just reporting the mean value. Done (P12L8-9)

P12L12: I got confused. The paragraph immediately above reports the derived SMB, so I assume that the thinning corrections are already made. Please reorganize paragraphs in Section 3.2 to clearly demonstrate the logical flow. Done

P12L14: Is it really Section 4.2 (Discussion)? If so, it's better to say something like "we discuss this point further in Section 4.2." No, it was 2.4 (corrected, P11L28)

P12L28-30: Remove the sentence about dynamic thinning; it is obvious and rather confusing. You just say here that the layer has been thinned, not thickened. Done

P13: see comments to Table 1. Consider moving these paragraphs about averaged SMB values to a discussion session where you compare these values to previous studies.

The comparison to previous studies is discussed with Figure 9. In the discussion, it is not our goal to compare the absolute values but only the trends. However, we think it is important to provide the values we use to calculate our trends and that this is part of the results section.

P14L4: add "~240 years" after "the whole period" Done (P12L28)

P14L6: change to SMB. Done (P13L2)

P14L8: "bounded", instead of "determined"? The real SMB is expected to be somewhere between the oldest and youngest estimates. Done (P13L6)

P14L11: Here you explain the error bars in Figure 6, but the explanation is too brief to give a comprehensive idea what they are. If I understand correctly, the authors assume that the summer peak can be shifted up to 5 cm to both sides. In other words, if the annual layer is A cm thick, the thickest possible layer can be A + 10 cm (5 cm widen to both sides), and the thinnest possible layer can be A - 10 cm. Then you applied the thinning factor to estimate the uncertainty of the estimated SMB value. Do I understand correctly? However, if this is the case, the error bar is shorter when the SMB value is small, and it is longer otherwise. I cannot such feature in Figure 6.

There was indeed a mistake in the way individual error bars were calculated. They are now calculated the way the Editor describes (except the error is reduced to 5 cm below 85 m because the resolution of water stable isotopes measurements increases). The explanation is now given in the text (P12L5-7).

P14L11-13: I cannot understand this argument. Consistent features between isotopes and ions support an hypothesis that both represent seasonal changes. However, because both were sampled by 5 cm or 10 cm, both depth profiles may overlook an annual cycle that appears less than 5 cm thick. Uncorrected layer thickness (orange curve in Fig. 6a) shows that it is unlikely to miss such thin annual layers, but similarity of isotope and ion profiles cannot be the evidence for this argument.

We thank the Editor for pointing this out and we have reworded (P13L8-14).

P14L22: Provide reference/status of this paper. Done (P13L25).

P15L6ff: "in the vicinity of the crest"; topographic feature (Crest), not ice-flow feature (divide), should be cited in terms of SMB's spatial pattern. I saw that "divide" is used at some other places as well; please correct them as well. Replaced, unless we specifically refer to the ice flow feature, when mentioning Raymond effect.

P15L7: Drews et al. did not exclude a possibility of recent crest migration, which is too young to deform the Raymond Arches found at depths greater than 50-100 m where Raymond Arches become more visible. Reworded (P14L9-19).

P15L21: here you say that the ice-core-derived results are compared with two climate models, but later you compare the results with ERA-Interim, RACMO2, and CESM. Amended (P14L21).

P15 L25: replace R2 with correlation coefficient or such. Done (P14L24-25).

P15L30: please add more information to explain why a freely-evoluting model output cannot be compared directly but still your discussion here can be valid. We added to Section 2.5: Because CESM is not bound by

observations, and is a freely evolving model that generates its own climate, the simulated SMB time series cannot be directly compared to the observed one. Instead we use a statistical approach: we use the historical time series of CESM…."

P16L1: CESM output of the SMB mentioned here (0.295) is an average value for a certain period or the most recent SMB in 2011?
This is the average of the historical period, i.e. 1850-2005. We added this to the text (P14L28).

P16L11: How many days is this region covered with sea ice? This information is necessary to judge how 20-40 days fewer sea ice coverage is significant.
That varies widely, from ~120 days in the northern part to > 200 days in the southern part. We changed this line to:
"sea ice coverage is substantially lower than average (20–40 fewer days with sea-ice cover, **i.e. about 10-30% reduction compared to the average of 120-200 days**, Fig 8)" (P15L7).

P17ff: Section 4.1 shows many numbers and it is very hard to keep tracking the main argument. Please carefully review this section and re-organize it so that the discussion can be presented more clearly. Done.

P17L23: Drews's Figures 3b and 7 shows that anomalously low SMB is persistent at the current position to the age of ice at 60 m depth. This is I think support evidence of author's argument that the observed trend of SMB in the ice core presents the temporal changes, not migrating spatial patterns. Done (P14L17-19).

P20L6: change accumulation to SMB. Done (P17L14).

P20L9: indicate the name of these two coastal sites that show significant increase of SMB in the last 20 years compared to the last 200 years. Done (P17L17).

P20L13: change "less important" to "insignificant" or "less visible". Done (P17L18).

P21L28: remove "2009 and 2011" so it will be "than average SMB years (Table 2)." Done (P18L8)

P22L15: It is said that detrended dataset is not shown, but the authors presented 11-year running mean SMB (Figure 6). Is this running mean record good enough to identify anomalous events in 2-4 years (1991-95 and 1940-42)?
No, the running mean is not sufficient so we reworded to: "our record does not support this observation." (P18L21).

P24L11: "A 120-m-long", change "divide" to "summit" or "ridge (or crest)". Done (P19L17)

P24L14: "Therefore we counted annual layers to develop oldest and youngest estimates of the ice. The annual layer thickness, density, and thinning functions are applied to derive time series of SMB from annual layer thicknesses." Reworded closely (P19L19-22).

P24L20: do you mean that "wind re-distribution is significant near the ice -core site but it is likely that this effect is persistent over time so that ice-core records represent SMB time series rather than migrating spatial patterns of SMB"? Reworded closely (P19L26-P20L1).

P24L25-27: I cannot agree. Probably you want to say "Neither currently available climate models and re-analysis data cannot resolve ice-rise topography so their predictions are hard to match with the ice-core-derived SMB. Nevertheless, their temporal trends can be compared, and …." Done (P20L9-14).

P25L10: I believe that the authors can be more confident about their results. Clear seasonal cycles (not only thin ice layers!) found in this ice core clearly demonstrated the potential of a deep core from this site as excellent paleoclimate proxies. Done (P20L18).

P25L21: Change "uncorrected SMB" to "annual layer thickness in ice"? "uncorrected SMB" sounds quite confusing. Done (P20L24).

Table 1
- I am not really sure how these average values for different periods are important. You said that it is for comparison with other studies and if so please consider adding an extra column showing the SMB values obtained from previous studies and compared with the average values that you are reporting.
These values from previous studies are shown in Table A1. It is now mentioned in Table 1 caption: 'These values may be compared with those of several published studies, summarized in Table A1'.

Figure 1
- Change "accumulation" in the figure to "SMB". Done.

Figure 3
- Add something like "d18O profiles are shown multiple times to better illustrate correlations between d18O and major ion profiles". Done.

Figure 5
- It's very hard to see thin gray bands. Use more distinct color (red, blue, or such, not gray). Done.

Figure 6
- Please align all four panels vertically so each panel can be a bit wider for full one-column width, and it is easier to compare time series. When I saw these panels first time, I had an impression that panels a and b are paired and c and d are paired. Done.
- What do error bars in panels b and c show?
They now show error on individual annual layer thickness calculated as 5-10 cm error (depending on water stable isotopes resolution) converted to m w.e.. This is now mentioned in the caption and in the text (P12L5-7)
- How is the uncertainty range (panel d) derived? Please explain more clearly in the main text.
It is the uncertainty range bounded by the youngest and the oldest estimates for 11 year running means. It is now stated in the text (P12L7-8).

Figure 7
- Distinguish curve and line in the caption. Pink and blue are curves, while black one is a line.
Done
- Please rewrite this caption; it is quite confusing. I believe that three datasets are normalized to their average values for the 1084-2000 period and their temporal variations are shown relative to those average values. I believe that "1979-1989" and "1850-2011" are typos.
The temporal coverage being different for the three datasets, as well as their absolute SMB values, we chose the most recent 11 years period common to the three datasets (1979-1989) to see how these 11 year running

means compare with each other in terms of trends. 1850-2011 is the period of overlap between CESM and our ice core data.

Figure 8
- The caption is confusing. I believe that you want to say "Large-scale atmospheric and sea ice anomalies observed in CESM historical time series (1850-2005) for the years when ice-core-derived SMB is within highest 10% of the all SMB values in the past ~240 years….."
As explained in the text, CESM is a global climate model, not bound by reanalyses (unlike RACMO2, for example), which generates its own internal climate. This means that the observed SMB time series cannot be directly compared to CESM SMB; instead, we use a statistical approach and select the ten years in the CESM time series with the highest SMB in the 1850-2005.

Figure 9
- Change "accumulation" in the figure to "SMB".
Done
- Change the caption so that it is clearer that this figure shows SMB reconstructed with ice cores over the continent.
Done

Figure S1
- Change to "….sections of sampling for major ions at 10 cm and 5 cm intervals. Isotope samples were taken at 5 cm intervals for the entire core."
Amended, except intervals is replaced by resolution.

- 2 sigma is used in Figure 5 to identify volcanic signals, whereas 4 sigma is used in this figure. I don't really see the merit to see this line; remove this line or justify why not 2 but 4 sigma is used here as a reference.
Removed

- It is impossible to distinguish light and dark gray colors in the ECM plot. And it is more important to show the 301-point (30 cm?) –smoothed ECM values in the figure because the smoothed ECM was used to facilitate annual layer counting.
The light grey line has been removed. The running mean shown here uses a 0.05 m smoothing window on top of the Savitsky Golay 301-point filter.

- I believe that only ECM data are shown in terms of the standard deviation, but not water stable isotopes and major ions (change the caption).
Done

Figure S2
- Add unit "sigma" to the first ECM panel.
Done

Appendix
- Again, use "accumulation" and "SMB" consistently.
Done
- Are latitude/longitude given in decimal degrees?
Yes

[revised manuscript text omitted]

Fig. S1. Full vertical profile of water stable isotopes with, from left to right: a grey and black band indicating sections of sampling for major ions at 10 cm and 5 cm resolution, respectively; water stable isotopes,  taken at 5 cm resolution for the entire core; major ions, taken at 5 cm resolution for discrete sections; normalized ECM conductivity (0.05 m running mean, expressed as multiple of standard deviation σ)   ); annual layer boundaries in the youngest (Green) and the oldest (Blue) estimates.  (each colour transition indicates a boundary ).

Fig. S2. Full vertical profile, as in Fig. S1 but split in 17 sections for more visibility.

[Figure]

[Figure]

[Figure]

[Figure]

[Figure]

[Figure]

[Figure]

[Figure]

[Figure]

[Figure]

[Figure]

[Figure]

[Figure]

[Figure]

[Figure]

[Figure]

[Figure]

**Appendix A**

[revised manuscript text omitted]

---

## Author Response (AR4)

Editor's comments (back) and author's response (red)

P1L26: Understood, but then the last sentence of Abstract in the latest manuscript should be revised.

Done (P2L1-3). The last sentence of the abstract has been modified to: This is the first record from a coastal ice core in East Antarctica to show an increase in SMB, beginning in the early 20th century and particularly marked during the last 50 years.

P4L26: Cite Figs. 1 and 4 of Drews et al. (2015) to show the core site location relative to radar-derived SMB (P5L20 in the revised manuscript).

Done (P4L28).

P18L16 in the revised manuscript: Remove SOI (not used in the revised manuscript).

Done (P15L2). Also removed SAM for the same reason (P14L29)

[revised manuscript text omitted]